# M³-Impute: Mask-guided Representation Learning for Missing Value Imputation

## Abstract

Missing values are a common problem that poses significant challenges to data analysis and machine learning. This problem necessitates the development of an effective imputation method to fill in the missing values accurately, thereby enhancing the overall quality and utility of the datasets. Existing imputation methods, however, fall short of explicitly considering the 'missingness' information in the data during the embedding initialization stage and modeling the entangled feature and sample correlations during the learning process, thus leading to inferior performance. We propose M³-Impute, which aims to explicitly leverage the missingness information and such correlations with novel masking schemes. M³-Impute first models the data as a bipartite graph and uses a graph neural network to learn node embeddings, where the refined embedding initialization process directly incorporates the missingness information. They are then optimized through M³-Impute's novel feature correlation unit (**FCU**) and sample correlation unit (**SCU**) that effectively captures feature and sample correlations for imputation. Experiment results on 25 benchmark datasets under three different missingness settings show the effectiveness of M³-Impute by achieving 20 best and 4 second-best MAE scores on average.

## 1 Introduction

Missing values in a dataset are a pervasive issue in real-world data analysis. They arise for various reasons, ranging from the limitations of data collection methods to errors during data transmission and storage. Since many data analysis algorithms cannot directly handle missing values, the most common way to deal with them is to discard the corresponding samples or features with missing values, which would compromise the quality of data analysis. To tackle this problem, missing value imputation algorithms have been proposed to preserve all samples and features by imputing missing values with estimated ones based on the observed values in the dataset, so that the dataset can be analyzed as a complete one without losing any information.

The imputation of missing values usually requires modeling of correlations between different features and samples. Feature-wise correlations help predict missing values from other observed features in the same sample, while sample-wise correlations help predict them in one sample from other similar samples. It is thus important to jointly model the feature-wise and sample-wise correlations in the dataset. In addition, the prediction of missing values also largely depends on the 'missingness' of the data, i.e., whether a certain feature value is observed or not in the dataset. Specifically, the missingness information directly determines which observed feature values can be used for imputation. For example, even if two samples are closely related, it may be less effective to use them for imputation if they have missing values in exactly the same features. It still remains a challenging problem how to jointly model feature-wise and sample-wise correlations with such data missingness.

Among existing methods for missing value imputation, traditional methods (Burgette & Reiter, 2010; Hastie et al., 2015a; Mazumder et al., 2010a; García-Laencina et al., 2010; Honaker et al., 2011; Mazumder et al., 2010c; Hastie et al., 2015b) extract data correlations with statistical models, which are generally not flexible in handling mixed data types and struggle to scale up to large datasets. Recent learning-based imputation methods (Li et al., 2019; Mattei & Frellsen, 2019; Yoon et al., 2018; Kyono et al., 2021; Zheng & Charoenphakdee, 2022; Tashiro et al., 2021; Yoon & Sull, 2020; Muzellec et al., 2020; Du et al., 2024), instead, take advantage of the strong expressiveness and

scalability of machine/deep learning algorithms to model data correlations. However, most of them are still built upon the raw tabular data structure as is, which greatly restricts them from jointly modeling the feature-wise and sample-wise correlations. In light of this, graph-based methods (You et al., 2020; Spinelli et al., 2020) have been proposed to model the raw data as a bipartite graph, with samples and features being two different types of nodes. A sample node and a feature node are connected if the feature value is observed in that sample. The missing values are then predicted as the inner product between the embeddings of the corresponding sample and feature nodes. However, this simple prediction does not explicitly consider the specific missingness information as mentioned above.

In this work, we address these problems by proposing $M^3$-Impute, a mask-guided representation learning method for missing value imputation. The key idea behind $M^3$-Impute is to explicitly utilize the data-missingness information as model input with our proposed novel masking schemes so that it can accurately learn feature-wise and sample-wise correlations in the presence of different kinds of data missingness. $M^3$-Impute first builds a bipartite graph from the data as used in You et al. (2020). In the embedding initialization for graph representation learning, however, we not only use the the relationships between samples and their associated features but also the missingness information so as to initialize the embeddings of samples and features jointly and effectively. We then propose novel feature correlation unit (**FCU**) and sample correlation unit (**SCU**) in $M^3$-Impute to explicitly take feature-wise and sample-wise correlations into account for imputation. **FCU** learns the correlations between the target missing feature and observed features within each sample, which are then further updated via a soft mask on the sample missingness information. **SCU** then computes the sample-wise correlations with another soft mask on the missingness information for each pair of samples that have values to impute. We then integrate the output embeddings of **FCU** and **SCU** to estimate the missing values in a dataset. We carry out extensive experiments on 25 open datasets. The results show that $M^3$-Impute outperforms state-of-the-art methods in 20 of the 25 datasets on average under three different settings of missing value patterns, achieving up to 22.22% improvement in MAE compared to the second-best method.

## 2 RELATED WORK

**Traditional methods:** These imputation approaches include joint modeling with expectation-maximization (EM) (Dempster et al., 1977; Ghahramani & Jordan, 1993; Honaker et al., 2011), $k$-nearest neighbors (kNN) (García-Laencina et al., 2010; Troyanskaya et al., 2001), and matrix completion (Hastie et al., 2015a; Cai et al., 2010; Candes & Recht, 2012; Mazumder et al., 2010b). However, joint modeling with EM and matrix completion often lack the flexibility to handle data with mixed modalities, while kNN faces scalability issues due to its high computational complexity. In contrast, $M^3$-Impute is scalable and adaptive to different data distributions.

**Learning-based methods:** Iterative imputation frameworks (Jarrett et al., 2022; Azur et al., 2011; Kyono et al., 2021; van Buuren & Groothuis-Oodshoorn, 2011; Stekhoven & Bühlmann, 2012; Van Buuren et al., 2006), such as MICE (van Buuren & Groothuis-Oodshoorn, 2011) and HyperImpute (Jarrett et al., 2022), have been extensively studied. These iterative frameworks apply different imputation methods for each feature and iteratively estimate missing values until convergence. In addition, for deep neural network learners, both generative models (Yoon et al., 2018; Mattei & Frellsen, 2019; Yoon & Sull, 2020; Li et al., 2019; Rombach et al., 2022; Zheng & Charoenphakdee, 2022) and discriminative models (Kyono et al., 2021; Du et al., 2024; Wu et al., 2020) have also been proposed. However, these methods are built upon raw tabular data structures, which may fall short of capturing the complex correlations in features, samples, and their combination. In contrast, $M^3$-Impute is based on the bipartite graph modeling of the data, which is more suitable for learning the data correlations for imputation.

**Graph neural network-based methods:** GNN-based methods (You et al., 2020; Spinelli et al., 2020) are proposed to address the drawbacks mentioned above due to their effectiveness in modeling complex relations between entities. Among them, GRAPE (You et al., 2020) transforms tabular data into a bipartite graph where features are one type of nodes and samples are the other. A sample node is connected to a feature node only if the corresponding feature value is present. This transformation allows the imputation task to be framed as a link prediction problem, where the inner product of the learned node embeddings is computed as the predicted values. However, these methods do not explicitly encode the missingness information of different samples and features

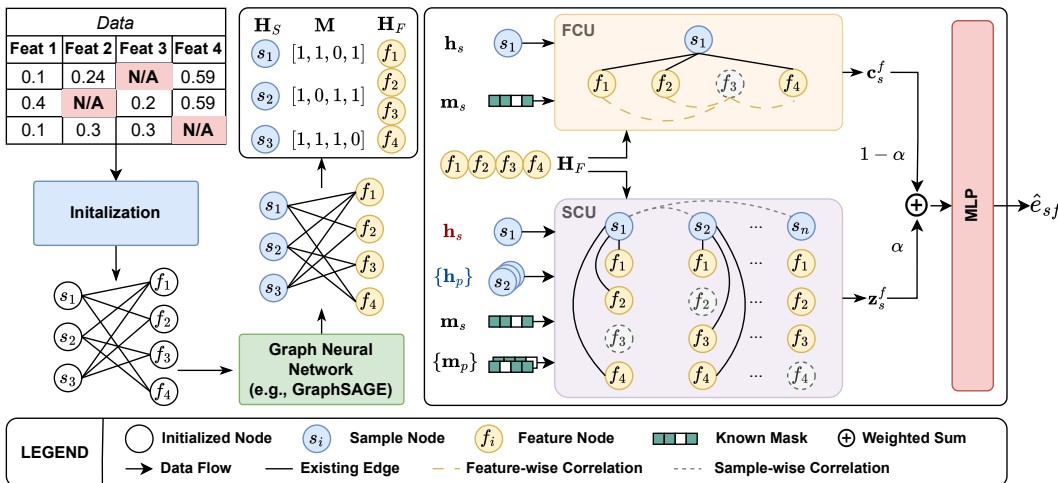

Figure 1: Overview of the M³-Impute model. The tabular data with missing values is first modeled as a bipartite graph with our refined initialization unit, which incorporates the missingness information in node embedding initialization. The graph is then processed with a GNN to update node embeddings. After that, we apply our novel soft masking schemes on these node embeddings to further encode correlation and missingness information in the learning process, using our novel components of feature correlation unit (**FCU**) and sample correlation unit (**SCU**). Eventually, the missing value is predicted with an MLP on the weighted sum of the outputs from **FCU** and **SCU**.

into the imputation process, which can impair their imputation accuracy. In contrast, M³-Impute enables explicit modeling of missingness information through **FCU** and **SCU** as well as our novel initialization unit so that feature-wise and sample-wise correlations can be accurately captured in the imputation process.

## 3 M³-IMPUTE

### 3.1 OVERVIEW

We here provide an overview of M³-Impute to impute the missing value of feature $f$ for a given sample $s$, as depicted in Figure 1. Initially, the data matrix with missing values is modeled as an undirected bipartite graph, and the missing value is imputed by predicting the edge weight $\hat{e}_{sf}$ of its corresponding missing edge (Section 3.2). M³-Impute next employs a GNN model, such as GraphSAGE (Hamilton et al., 2017), on the bipartite graph to learn the embeddings of samples and features. These embeddings, along with the known masks of the data matrix (used to indicate which feature values are available in each sample), are then input into our novel feature correlation unit (**FCU**) and sample correlation unit (**SCU**), which shall be explained in Section 3.3 and Section 3.4, to obtain feature-wise and sample-wise correlations, respectively. Finally, M³-Impute takes the feature-wise and sample-wise correlations into a multi-layer perceptron (MLP) to predict the missing feature value $\hat{e}_{sf}$ (Section 3.5). The whole process, including the embedding generation, is trained in an end-to-end manner.

### 3.2 INITIALIZATION UNIT

Let $\mathbf{A} \in \mathbb{R}^{n \times m}$ be an $n \times m$ matrix that consists of $n$ data samples and $m$ features, where $\mathbf{A}_{ij}$ denotes the $j$-th feature value of the $i$-th data sample. We introduce an $n \times m$ mask matrix $\mathbf{M} \in \{0, 1\}^{n \times m}$ for $\mathbf{A}$ to indicate that the value of $\mathbf{A}_{ij}$ is *observed* when $\mathbf{M}_{ij} = 1$. In other words, the goal of imputation here is to predict the missing feature values $\mathbf{A}_{ij}$ for $i$ and $j$ such that $\mathbf{M}_{ij} = 0$. We define the *masked* data matrix $\mathbf{D}$ to be $\mathbf{D} = \mathbf{A} \odot \mathbf{M}$, where $\odot$ is the Hadamard product, i.e., the element-wise multiplication of two matrices.

As used in recent studies (You et al., 2020), we model the masked data matrix $\mathbf{D}$ as a bipartite graph and tackle the missing value imputation problem as a link prediction task on the bipartite graph. Specifically, $\mathbf{D}$ is modeled as an undirected bipartite graph $\mathcal{G} = (\mathcal{S} \cup \mathcal{F}, \mathcal{E})$, where $\mathcal{S} =$

$\{s_1, s_2, \ldots, s_n\}$ is the set of 'sample' nodes and $\mathcal{F} = \{f_1, f_2, \ldots, f_m\}$ is the set of 'feature' nodes. Also, $\mathcal{E}$ is the set of edges that only exist between sample node $s$ and feature node $f$ when $\mathbf{D}_{sf} \neq 0$, and each edge $(s, f) \in \mathcal{E}$ is associated with edge weight $e_{sf}$, which is given by $e_{sf} = \mathbf{D}_{sf}$. Then, the missing value imputation problem becomes, for any missing entries in $\mathbf{D}$ (where $\mathbf{D}_{sf} = 0$), to predict their corresponding edge weights by developing a learnable mapping $F(\cdot)$, i.e.,

$$\hat{e}_{sf} = F(\mathcal{G}, (s, f) \notin \mathcal{E}). \tag{1}$$

The recent studies that use the bipartite graph modeling initialize all sample node embeddings as all-one vectors and feature node embeddings as one-hot vectors, which have a value 1 in the positions representing their respective features and 0's elsewhere. We observe, however, that such an initialization does not effectively utilize the information from the masked data matrix, which leads to inferior imputation accuracy, as shall be demonstrated in Section 4.3. Thus, in $M^3$-Impute, we propose to initialize each sample node embedding based on its associated (initial) feature embeddings instead of initializing them separately. While the feature embeddings are randomly initialized, the sample node embeddings are initialized in a way that reflects the embeddings of the features whose values are available in their corresponding samples.

Let $\mathbf{h}_f^0$ be the initial embedding of feature $f$, which is a randomly initialized $d$-dimensional vector, and define $\mathbf{H}_F^0 = [\mathbf{h}_{f_1}^0 \mathbf{h}_{f_2}^0 \ldots \mathbf{h}_{f_m}^0] \in \mathbb{R}^{d \times m}$. Also, let $\mathbf{d}_s \in \mathbb{R}^m$ be the $s$-th column vector of $\mathbf{D}^\top$, which is a vector of the feature values of sample $s$, and let $\mathbf{m}_s \in \mathbb{R}^m$ be its corresponding mask vector, i.e., $\mathbf{m}_s = \mathrm{col}_s(\mathbf{M}^\top)$, where $\mathrm{col}_s(\cdot)$ denotes the $s$-th column vector of the matrix. We then initialize the embedding $\mathbf{h}_s^0$ of each sample node $s$ as follows:

$$\mathbf{h}_s^0 = \phi\Big(\mathbf{H}_F^0\big[\mathbf{d}_s + \epsilon(\mathbb{1} - \mathbf{m}_s)\big]\Big), \tag{2}$$

where $\mathbb{1} \in \mathbb{R}^m$ is an all-one vector, and $\phi(\cdot)$ is an MLP. Note that the term $\mathbf{d}_s + \epsilon(\mathbb{1} - \mathbf{m}_s)$ indicates a vector that consists of observable feature values of $s$ and some small positive values $\epsilon$ in the places where the feature values are unavailable (masked out).

### 3.3 FEATURE CORRELATION UNIT

To improve the accuracy of missing value imputation, we aim to fully exploit feature correlations which often appear in the datasets. While the feature correlations are naturally captured by GNNs, we observe that there is still room for improvement. We propose **FCU** as an integral component of $M^3$-Impute to fully exploit the feature correlations.

To impute the missing value of feature $f$ for a given sample $s$, **FCU** begins by computing the feature 'context' vector of sample $s$ in the embedding space that reflects the correlations between the target missing feature $f$ and observed features. Let $\mathbf{h}_f \in \mathbb{R}^d$ be the learned embedding vector of feature $f$ from the GNN, and let $\mathbf{H}_F$ be the $d \times m$ matrix that consists of all the learned feature embedding vectors. We first obtain dot-product similarities between feature $f$ and all the features in the embedding space, i.e., $\mathbf{H}_F^\top \mathbf{h}_f$. We then mask out the similarity values with respect to *non-observed* features in sample $s$. Here, instead of applying the mask vector $\mathbf{m}_s$ of sample $s$ directly, we use a learnable 'soft' mask vector, denoted by $\mathbf{m}_s'$, which is defined to be $\mathbf{m}_s' = \sigma_1(\mathbf{m}_s) \in \mathbb{R}^m$, where $\sigma_1(\cdot)$ is an MLP with the GELU activation function (Hendrycks & Gimpel, 2016). In other words, we obtain feature-wise similarities with respect to sample $s$, denoted by $\mathbf{r}_s^f$, as follows:

$$\mathbf{r}_s^f = \sigma_2\left((\mathbf{H}_F^\top \mathbf{h}_f) \odot \mathbf{m}_s'\right) \in \mathbb{R}^d, \tag{3}$$

where $\sigma_2(\cdot)$ denotes another MLP with the GELU activation function. **FCU** next obtains the Hadamard product between the learned embedding vector of sample $s$, $\mathbf{h}_s$, and the feature-wise similarities with respect to sample $s$, $\mathbf{r}_s^f$, to learn their joint representations in a multiplicative manner. Specifically, **FCU** obtains the feature context vector of sample $s$, denoted by $\mathbf{c}_s^f$, as follows:

$$\mathbf{c}_s^f = \sigma_3\left(\mathbf{h}_s \odot \mathbf{r}_s^f\right) \in \mathbb{R}^d, \tag{4}$$

where $\sigma_3(\cdot)$ is also an MLP with the GELU activation function. That is, **FCU** fuses the representation vector of $s$ and the vector that has embedding similarity values between the target feature $f$ and the available features in $s$ through the effective use of the soft mask $\mathbf{m}_s'$. From (3) and (4), the operations of **FCU** can be written as

$$\mathbf{c}_s^f = \mathbf{FCU}(\mathbf{h}_s, \mathbf{m}_s, \mathbf{H}_F) = \sigma_3\left(\mathbf{h}_s \odot \sigma_2\left((\mathbf{H}_F^\top \mathbf{h}_f) \odot \sigma_1(\mathbf{m}_s)\right)\right). \tag{5}$$

## 3.4 SAMPLE CORRELATION UNIT

To measure similarities between $s$ and other samples, a common approach would be to use the dot product or cosine similarity between their embedding vectors. This approach, however, fails to take into account the observability or availability of each feature in a sample. It also does not capture the fact that different observed features are of different importance to the target feature to impute when it comes to measuring the similarities. We introduce **SCU** as another integral component of M³-Impute to compute the sample 'context' vector of sample $s$ by incorporating the embedding vectors of its similar samples as well as different weights of observed features. **SCU** works based on the two novel masking schemes, which shall be explained shortly.

Suppose we are to impute the missing value of feature $f$ for a given sample $s$. **SCU** aims to leverage the information from the samples that are similar to $s$. As a first step to this end, we create a subset of samples $\mathcal{P} \subset \mathcal{S}$ that are similar to $s$. Specifically, we randomly choose and put a sample into $\mathcal{P}$ with probability that is proportional to the cosine similarity between $s$ and the sample. This operation is repeated without replacement until $\mathcal{P}$ reaches a given size.

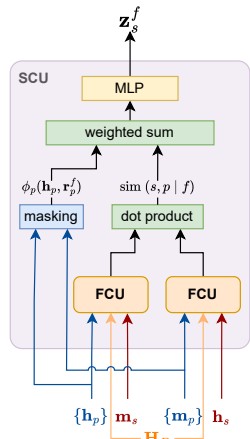

Figure 2: **SCU**.

**Mutual Sample Masking:** Given a subset of samples $\mathcal{P}$ that include $s$, we first compute the pairwise similarities between $s$ and other samples in the subset $\mathcal{P}$. While they are computed in a similar way to **FCU**, we only consider the commonly observed features (or the common ones that have feature values) in both $s$ and its peer $p \in \mathcal{P} \setminus \{s\}$, to calculate their pairwise similarity in the sense that the missing value of feature $f$ is inferred. Specifically, we compute the pairwise similarity between $s$ and $p \in \mathcal{P} \setminus \{s\}$, which is denoted by $\text{sim}(s, p \mid f)$, as follows:

$$\text{sim}(s, p \mid f) = \textbf{FCU}(\mathbf{h}_s, \mathbf{m}_p, \mathbf{H}_F) \cdot \textbf{FCU}(\mathbf{h}_p, \mathbf{m}_s, \mathbf{H}_F) \in \mathbb{R}, \quad (6)$$

where $\mathbf{h}_s$ and $\mathbf{h}_p$ are the learned embedding vectors of samples $s$ and $p$ from the GNN, respectively, and $\mathbf{m}_s$ and $\mathbf{m}_p$ are their respective mask vectors. Note that the multiplication in the RHS of (6) is the dot product.

**Irrelevant Feature Masking:** After we obtain the pairwise similarities between $s$ and other samples in $\mathcal{P}$, it would be natural to consider a weighted sum of their corresponding embedding vectors, i.e., $\sum_{p \in \mathcal{P} \setminus \{s\}} \text{sim}(s, p \mid f) \, \mathbf{h}_p$, in imputing the value of the target feature $f$. However, we observe that $\mathbf{h}_p$ contains the information from the features whose values are available in $p$ as well as possibly other features as it is learned via the so-called neighborhood aggregation mechanism that is central to GNNs, but some of the features may be irrelevant in inferring the value of feature $f$. Thus, instead of using $\{\mathbf{h}_p\}$ directly, we introduce a $d$-dimensional mask vector $\mathbf{r}_p^f$ for $\mathbf{h}_p$, which is to mask out potentially irrelevant feature information in $\mathbf{h}_p$, when it comes to imputing the value of feature $f$. Specifically, it is defined by

$$\mathbf{r}_p^f = \sigma_4 \left( [\mathbf{m}_p; \overline{\mathbf{m}}_f] \right) \in \mathbb{R}^d, \quad (7)$$

where $\overline{\mathbf{m}}_f$ is an $m$-dimensional one-hot vector that has a value 1 in the place of feature $f$ and 0's elsewhere, $[\cdot \, ; \cdot]$ denotes the vector concatenation operation, and $\sigma_4(\cdot)$ is an MLP with the GELU activation function. Note that the rationale behind the design of $\mathbf{r}_p^f$ is to embed the information on the features whose values are present in $p$ as well as the information on the target feature $f$ to impute. The mask $\mathbf{r}_p^f$ is then applied to $\mathbf{h}_p$ to obtain the masked embedding vector of $p$ as follows:

$$\phi_p(\mathbf{h}_p, \mathbf{r}_p^f) = \sigma_5 \left( \mathbf{h}_p \odot \mathbf{r}_p^f \right) \in \mathbb{R}^d, \quad (8)$$

where $\sigma_5(\cdot)$ is also an MLP with the GELU activation function. Once we have the masked embedding vectors of samples (excluding $s$) in $\mathcal{P}$, we finally compute the sample context vector of sample $s$, denoted by $\mathbf{z}_s^f$, which is a weighted sum of the masked embedding vectors with weights being the pairwise similarity values, i.e.,

$$\mathbf{z}_s^f = \sigma_6 \left( \sum_{p \in \mathcal{P} \setminus \{s\}} \text{sim}(s, p \mid f) \, \phi_p(\mathbf{h}_p, \mathbf{r}_p^f) \right) \in \mathbb{R}^d, \quad (9)$$

---

**Algorithm 1** Forward computation of $M^3$-Impute to impute the value of feature $f$ for sample $s$.

---

1: **Input:** Bipartite graph $\mathcal{G}$, initial feature node embeddings $\mathbf{H}_F^0$, GNN model (e.g., GraphSAGE) $\mathbf{GNN}(\cdot)$, known mask matrix $\mathbf{M}$, and a subset of samples $\mathcal{P} \subset \mathcal{S}$.
2: **Output:** Predicted missing feature value $\hat{e}_{sf}$.
3: Obtain initial sample node embeddings $\mathbf{H}_S^0$ according to Equation (2).
4: $\mathbf{H}_S, \mathbf{H}_F = \mathbf{GNN}(\mathbf{H}_S^0, \mathbf{H}_F^0, \mathcal{G})$.            ▷ Perform graph representation learning
5: $\mathbf{c}_s^f = \mathbf{FCU}(\mathbf{h}_s, \mathbf{m}_s, \mathbf{H}_F)$.
6: $\mathbf{z}_s^f = \mathbf{SCU}(\mathbf{H}_{\mathcal{P}}, \mathbf{M}_{\mathcal{P}}, \mathbf{H}_F)$.
7: Predict the missing feature value $\hat{e}_{sf}$ using Equation (11).

---

where $\sigma_6(\cdot)$ is again an MLP with the GELU activation function. From (6)–(9), the operations of **SCU** can be written as

$$\mathbf{z}_s^f = \mathbf{SCU}(\mathbf{H}_{\mathcal{P}}, \mathbf{M}_{\mathcal{P}}, \mathbf{H}_F) = \sigma_6\left(\sum_{p \in \mathcal{P} \setminus \{s\}} \mathrm{sim}(s, p \mid f)\, \sigma_5\left(\mathbf{h}_p \odot \sigma_4\left([\mathbf{m}_p; \overline{\mathbf{m}}_f]\right)\right)\right), \quad (10)$$

where $\mathbf{H}_{\mathcal{P}} = \{\mathbf{h}_p, p \in \mathcal{P}\}$ and $\mathbf{M}_{\mathcal{P}} = \{\mathbf{m}_p, p \in \mathcal{P}\}$.

### 3.5 IMPUTATION

For a given sample $s$, to impute the missing value of feature $f$, $M^3$-Impute obtains its feature context vector $\mathbf{c}_s^f$ and sample context vector $\mathbf{z}_s^f$ through **FCU** and **SCU**, respectively, which are then used for imputation. Specifically, it is done by predicting the corresponding edge weight $\hat{e}_{sf}$ as follows:

$$\hat{e}_{sf} = \phi_\alpha\left((1 - \alpha)\mathbf{c}_s^f + \alpha \mathbf{z}_s^f\right), \quad (11)$$

where $\phi_\alpha(\cdot)$ denotes an MLP with a non-linear activation function (i.e., ReLU for continuous values and softmax for discrete ones), and $\alpha$ is a learnable scalar parameter. This scalar parameter $\alpha$ is introduced to strike a balance between leveraging feature-wise correlation and sample-wise correlation. It is necessary because the quality of $\mathbf{z}_s^f$ relies on the quality of the samples chosen in $\mathcal{P}$, so overly relying on $\mathbf{z}_s^f$ would backfire if their quality is not as desired. To address this problem, instead of employing a fixed weight $\alpha$, we make $\alpha$ learnable and adaptive in determining the weights for $\mathbf{c}_s^f$ and $\mathbf{z}_s^f$ . Note that this kind of learnable parameter approach has been widely adopted in natural language processing (See et al., 2017; Wang et al., 2019; Li et al., 2020; Paulus et al., 2018) and computer vision (Dai et al., 2017; Zhu et al., 2019a;b), showing superior performance to its fixed counterpart. In $M^3$-Impute, the scalar parameter $\alpha$ is learned based on the similarity values between $s$ and its peer samples $p \in \mathcal{P} \setminus \{s\}$ as follows:

$$\alpha = \phi_\gamma\left(\left\|_{p \in \mathcal{P} \setminus \{s\}} \mathrm{sim}\left(s, p \mid f\right)\right.\right), \quad (12)$$

where $\|$ represents the concatenation operation, and $\phi_\gamma(\cdot)$ is an MLP with the activation function $\gamma(x) = 1 - 1/e^{|x|}$. The overall operation of $M^3$-Impute is summarized in Algorithm 1. To learn network parameters, we use cross-entropy loss and mean square error loss for imputing discrete and continuous feature values, respectively.

## 4 EXPERIMENTS

### 4.1 EXPERIMENT SETUP

**Datasets:** We conduct experiments on 25 open datasets. These real-world datasets consist of mixed data types with both continuous and discrete values and cover different domains including civil engineering (CONCRETE, ENERGY), physics and chemistry (YACHT), thermal dynamics (NAVAL), etc. Since the datasets are fully observed, we introduce missing values by applying a randomly generated mask to the data matrix. Specifically, as used in prior studies (Jarrett et al., 2022; Kyono et al., 2021), we apply three masking generation schemes, namely missing completely at random (MCAR), missing at random (MAR), and missing not at random (MNAR).[1] We use MCAR with

---

[1]More details about the datasets and mask generation for missing values can be found in Appendix.

Table 1: Imputation accuracy in MAE. MAE scores are enlarged by 10 times.

| | Yacht | Wine | Concrete | Housing | Energy | Naval | Kin8nm | Power |
|---|---|---|---|---|---|---|---|---|
| Mean | $2.09 \pm .04$ | $0.98 \pm .01$ | $1.79 \pm .01$ | $1.85 \pm .00$ | $3.10 \pm .04$ | $2.31 \pm .00$ | $\underline{2.50} \pm .00$ | $1.68 \pm .00$ |
| Svd | $2.46 \pm .16$ | $0.92 \pm .01$ | $1.94 \pm .02$ | $1.53 \pm .03$ | $2.24 \pm .06$ | $0.50 \pm .00$ | $3.67 \pm .06$ | $2.33 \pm .01$ |
| Spectral | $2.64 \pm .11$ | $0.91 \pm .01$ | $1.98 \pm .04$ | $1.46 \pm .03$ | $2.26 \pm .09$ | $0.41 \pm .00$ | $2.80 \pm .01$ | $2.13 \pm .01$ |
| Mice | $1.68 \pm .05$ | $0.77 \pm .00$ | $1.34 \pm .01$ | $1.16 \pm .03$ | $1.53 \pm .04$ | $0.20 \pm .01$ | $\underline{2.50} \pm .00$ | $1.16 \pm .01$ |
| Knn | $1.67 \pm .02$ | $0.72 \pm .00$ | $1.16 \pm .03$ | $0.95 \pm .01$ | $1.81 \pm .03$ | $0.10 \pm .00$ | $2.77 \pm .01$ | $1.38 \pm .01$ |
| Gain | $2.26 \pm .11$ | $0.86 \pm .00$ | $1.67 \pm .03$ | $1.23 \pm .02$ | $1.99 \pm .03$ | $0.46 \pm .02$ | $2.70 \pm .00$ | $1.31 \pm .05$ |
| Miwae | $2.37 \pm .01$ | $1.00 \pm .00$ | $1.81 \pm .01$ | $1.74 \pm .04$ | $2.79 \pm .04$ | $2.37 \pm .00$ | $2.57 \pm .01$ | $1.72 \pm .00$ |
| Grape | $\underline{1.46} \pm .01$ | $\mathbf{0.60} \pm .00$ | $\underline{0.75} \pm .01$ | $\underline{0.64} \pm .01$ | $1.36 \pm .01$ | $0.07 \pm .00$ | $\underline{2.50} \pm .00$ | $\underline{1.00} \pm .00$ |
| Miracle | $3.84 \pm .00$ | $0.70 \pm .00$ | $1.71 \pm .05$ | $3.12 \pm .00$ | $3.94 \pm .01$ | $0.18 \pm .00$ | $\mathbf{2.49} \pm .00$ | $1.13 \pm .01$ |
| HyperImpute | $1.76 \pm .03$ | $\underline{0.67} \pm .01$ | $0.84 \pm .02$ | $0.82 \pm .01$ | $\underline{1.32} \pm .02$ | $\mathbf{0.04} \pm .00$ | $2.58 \pm .05$ | $1.06 \pm .01$ |
| $M^3$-Impute | $\mathbf{1.33} \pm .04$ | $\mathbf{0.60} \pm .00$ | $\mathbf{0.71} \pm .01$ | $\mathbf{0.59} \pm .00$ | $\mathbf{1.31} \pm .01$ | $\underline{0.06} \pm .00$ | $\underline{2.50} \pm .00$ | $\mathbf{0.99} \pm .00$ |

a missing ratio of 30%, unless otherwise specified. We follow the preprocessing steps adopted in Grape (You et al., 2020) to scale feature values to [0, 1] with a MinMax scaler (Leskovec et al., 2014). Due to the space limit, we below present the results of eight datasets that are used in Grape and report other results in Appendix.

**Baseline models:** $M^3$-Impute is compared against popular and state-of-the-art imputation methods, including statistical methods, deep generative methods, and graph-based methods listed as follows: **MEAN**: It imputes the missing value $\hat{e}_{sf}$ as the mean of observed values in feature $f$ from all the samples. K-nearest neighbors (**kNN**) (Troyanskaya et al., 2001): It imputes the missing value $\hat{e}_{sf}$ using the kNNs that have observed values in feature $f$ with weights that are based on the Euclidean distance to sample $s$. Multivariate imputation by chained equations (**Mice**) (van Buuren & Groothuis-Oudshoorn, 2011): This method runs multiple regressions where each missing value is modeled upon the observed non-missing values. Iterative SVD (**Svd**) (Hastie et al., 2015a): It imputes missing values by solving a matrix completion problem with iterative low-rank singular value decomposition. Spectral regularization algorithm (**Spectral**) (Mazumder et al., 2010a): This matrix completion algorithm uses the nuclear norm as a regularizer and imputes missing values with iterative soft-thresholded SVD. **Miwae** (Mattei & Frellsen, 2019): It works based on an autoencoder generative model trained to maximize a potentially tight lower bound of the log-likelihood of the observed data and Monte Carlo techniques for imputation. **Miracle** (Kyono et al., 2021): It uses the imputation results from naive methods such as **MEAN** and refines them iteratively by learning a missingness graph (m-graph) and regularizing an imputation function. **Gain** (Yoon et al., 2018): This method trains a data imputation generator with a generalized generative adversarial network in which the discriminator aims to distinguish between real and imputed values. **Grape** (You et al., 2020): It models the data as a bipartite graph and imputes missing values by predicting the weights of the missing edges, each of which is done based on the inner product between the embeddings of its corresponding sample and feature nodes. **HyperImpute** (Jarrett et al., 2022): HyperImpute is a framework that conducts an extensive search among a set of imputation methods, selecting the optimal imputation method with fine-tuned parameters for each feature in the dataset. We follow the official implementations of all the baseline models and report their hyperparameters in Appendix.

**$M^3$-Impute configurations:** Parameters of $M^3$-Impute are updated by the Adam optimizer with a learning rate of 0.001 for 40,000 epochs. For graph representation learning, we use a three-layer GNN model, which is a variant of GraphSAGE (Hamilton et al., 2017) that not only learns node embeddings but also edge embeddings via the neighborhood aggregation mechanism, as similarly used in Grape (You et al., 2020). We employ mean-pooling as the aggregation function and use ReLU as the activation function for the GNN layers. We set the embedding dimension $d$ to 128. We randomly drop 50% of observable edges during training to improve the model's generalization ability. For each experiment, we conduct five runs with different random seeds and report the average results.

## 4.2 OVERALL PERFORMANCE

We first compare the feature imputation performance of $M^3$-Impute with popular and state-of-the-art imputation methods. As shown in Table 1, $M^3$-Impute achieves the lowest imputation MAE for six out of the eight examined datasets and the second-best MAE scores in the other two, which validates the effectiveness of $M^3$-Impute. For KIN8NM dataset, $M^3$-Impute underperforms Miracle. It is mainly because each feature in KIN8NM is independent of the others, so none of the observed features can help impute missing feature values. For NAVAL dataset, the only model that outperforms $M^3$-Impute is HyperImpute (Jarrett et al., 2022). In the NAVAL dataset, nearly every feature exhibits a

Table 2: Ablation study. $M^3$-Uniform stands for $M^3$-Impute with the uniform sampling strategy.

| | Yacht | Wine | Concrete | Housing | Energy | Naval | Kin8nm | Power |
|---|---|---|---|---|---|---|---|---|
| HyperImpute | $1.76 \pm .03$ | $0.67 \pm .01$ | $0.84 \pm .02$ | $0.82 \pm .01$ | $1.32 \pm .02$ | $\mathbf{0.04} \pm .00$ | $\underline{2.58} \pm .05$ | $1.06 \pm .01$ |
| Grape | $1.46 \pm .01$ | $\mathbf{0.60} \pm .00$ | $0.75 \pm .01$ | $0.64 \pm .01$ | $1.36 \pm .01$ | $0.07 \pm .00$ | $\mathbf{2.50} \pm .00$ | $\underline{1.00} \pm .00$ |
| **Architecture** | | | | | | | | |
| Init Only | $1.43 \pm .01$ | $\mathbf{0.60} \pm .00$ | $0.74 \pm .00$ | $0.63 \pm .01$ | $1.35 \pm .01$ | $\underline{0.06} \pm .00$ | $\mathbf{2.50} \pm .00$ | $\mathbf{0.99} \pm .00$ |
| Init+**FCU** | $1.35 \pm .01$ | $\underline{0.61} \pm .00$ | $\underline{0.72} \pm .03$ | $\underline{0.61} \pm .02$ | $1.32 \pm .00$ | $\underline{0.07} \pm .01$ | $\mathbf{2.50} \pm .00$ | $\mathbf{0.99} \pm .00$ |
| Init+**SCU** | $1.37 \pm .01$ | $\mathbf{0.60} \pm .00$ | $0.73 \pm .00$ | $0.63 \pm .01$ | $\mathbf{1.30} \pm .00$ | $0.09 \pm .01$ | $\mathbf{2.50} \pm .00$ | $\underline{1.00} \pm .00$ |
| $M^3$-Impute | $\mathbf{1.33} \pm .04$ | $\mathbf{0.60} \pm .00$ | $\mathbf{0.71} \pm .01$ | $\mathbf{0.59} \pm .00$ | $\underline{1.31} \pm .01$ | $\underline{0.06} \pm .00$ | $\mathbf{2.50} \pm .00$ | $\mathbf{0.99} \pm .00$ |
| **Sampling Strategy** | | | | | | | | |
| $M^3$-Uniform | $\underline{1.34} \pm .01$ | $\mathbf{0.60} \pm .00$ | $0.73 \pm .01$ | $\underline{0.61} \pm .00$ | $\underline{1.31} \pm .00$ | $\underline{0.06} \pm .00$ | $\mathbf{2.50} \pm .00$ | $\mathbf{0.99} \pm .00$ |

strong linear correlation with the other features, i.e., every pair of features has correlation coefficient close to one. This allows HyperImpute to readily select a linear model from its model pool for each feature to impute. Nonetheless, $M^3$-Impute exhibits overall superior performance to the baselines as it can be well adapted to datasets with different levels of correlations over features and samples. In other words, $M^3$-Impute benefits from explicitly incorporating the missingness information with our carefully designed masking schemes to better capture feature-wise and sample-wise correlations.

Furthermore, we evaluate the performance of $M^3$-Impute under MAR and MNAR settings. We observe that $M^3$-Impute consistently outperforms all the baselines under all the eight datasets and achieves an even larger margin in the improvement compared to the case with MCAR setting. This implies that our explicit modeling of the missingness information through our novel soft masking schemes in **FCU** and **SCU** as well as the initialization unit is effective in handling different patterns of missing values in the input data. It is worth noting that some baseline models may perform worse than simple imputers, such as Mean, on certain datasets, as similarly observed in recent studies (You et al., 2020; Jarrett et al., 2022). It may be because these datasets are of relative small size, and the number of samples and features is not sufficient to train the corresponding models. Comprehensive results on all the datasets across different missing ratios are provided in Appendix.

## 4.3 ABLATION STUDY

To study the effectiveness of three integral components of $M^3$-Impute, we consider three variants of $M^3$-Impute, each with a subset of the components, namely initialization only (Init Only), initialization + **FCU** (Init + **FCU**), and initialization + **SCU** (Init + **SCU**). The performance of these variants are evaluated against the top-performing imputation baselines such as Grape and HyperImpute. As shown in Table 2, the three variants derived from $M^3$-Impute achieve lower MAE values than both baselines in most datasets, demonstrating the effectiveness of our novel components in $M^3$-Impute.

Specifically, for initialization only, the key difference between $M^3$-Impute and Grape lies in our refined initialization process to explicitly leverage missingness information in node embeddings. The reduced MAE values observed by the Init Only variant demonstrate that our proposed initialization process is more effective in utilizing information between samples and their associated features, including missing ones, as compared to the basic initialization used in Grape (You et al., 2020). In addition, we observe that when **FCU** or **SCU** is incorporated, MAE values are further reduced for most datasets. This validates that explicitly modeling of missingness information through our novel masking schemes in **FCU** and **SCU** indeed improves imputation accuracy. When all the three components are combined together as in $M^3$-Impute, they work synergistically to lower MAE values, validating the efficacy of incorporating the missingness information when capturing sample-wise and feature-wise correlations for missing data imputation.

## 4.4 ROBUSTNESS

**Missing ratio:** In practice, datasets may possess different missing ratios. To validate the model's robustness under such circumstances, we evaluate the performance of $M^3$-Impute and other baseline models with varying missing ratios, i.e., 0.1, 0.3, 0.5, and 0.7. Figure 3 shows their performance. We use the MAE of HyperImpute ($HI$) as the reference performance and offset the performance of each model by $\text{MAE}_x - \text{MAE}_{HI}$, where $x$ represents the considered model. For clarity, we here only report the results of four top-performing models. As shown in Figure 3, $M^3$-Impute outperforms other baseline models for almost all the cases, especially under YACHT, CONCRETE, ENERGY,

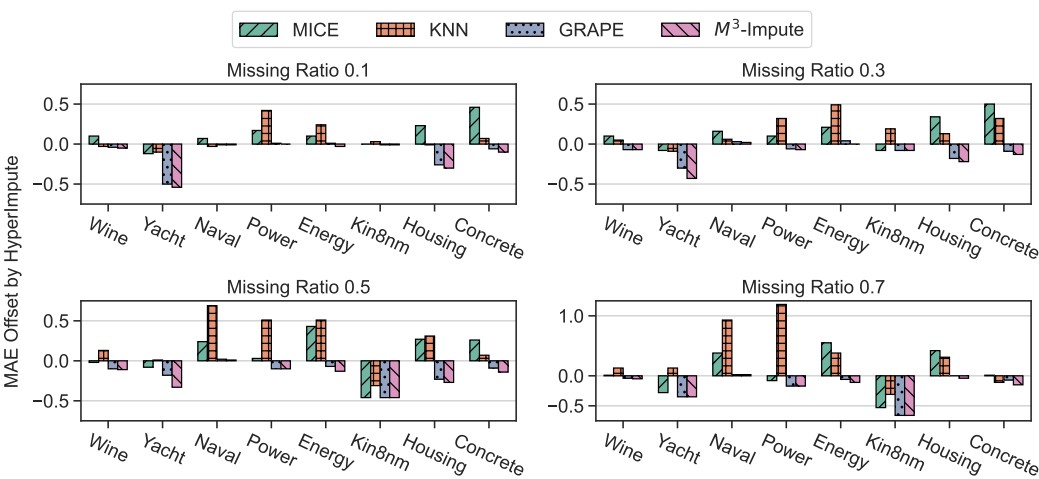

Figure 3: Model performance vs. missing ratios. MAE scores are offset by HyperImpute.

and HOUSING datasets. It is worth noting that modeling feature correlations in these datasets is particularly challenging due to the presence of considerable amounts of weakly correlated features, along with a few strongly correlated ones. Nonetheless, **FCU** and **SCU** in M³-Impute are able to better capture such correlations with our efficient masking schemes, thereby resulting in a large improvement in imputation accuracy. In addition, for KIN8NM dataset, M³-Impute ties with the second-best model, Grape. As mentioned in Section 4.2, each feature in KIN8NM is independent of the others, so none of the observed features can help impute missing feature values. For NAVAL dataset, where each feature strongly correlates with the others, M³-Impute surpasses Grape but falls short of HyperImpute, due to the same reason as discussed in Section 4.2. Overall, M³-Impute is robust to various missing ratios. Comprehensive results can be found in Appendix.

**Sampling strategy in SCU:** While **SCU** uses a sampling strategy based on pairwise cosine similarities to construct a subset of samples $\mathcal{P}$, the simplest sampling strategy to build $\mathcal{P}$ would be to choose samples uniformly at random without replacement (M³-Uniform). Intuitively, this approach cannot identify similar peer samples accurately and thus would lead to inferior performance. Nonetheless, as shown in Table 2, even with this naive uniform sampling strategy, M³-Uniform still outperforms the two leading imputation baselines.

Table 3: MAE scores for varying peer-sample size ($|\mathcal{P}|-1$) and different values of $\epsilon$.

|  | Yacht | Wine | Concrete | Housing | Energy | Naval | Kin8nm | Power |
|---|---|---|---|---|---|---|---|---|
| Peer = 1 | $1.34 \pm .00$ | $\mathbf{0.60} \pm .00$ | $0.73 \pm .00$ | $0.61 \pm .01$ | $1.32 \pm .00$ | $\mathbf{0.06} \pm .00$ | $\mathbf{2.5} \pm .00$ | $\mathbf{0.99} \pm .00$ |
| Peer = 2 | $1.35 \pm .01$ | $\underline{0.61} \pm .01$ | $\underline{0.72} \pm .01$ | $\underline{0.59} \pm .00$ | $\underline{1.32} \pm .00$ | $\mathbf{0.06} \pm .00$ | $2.5 \pm .00$ | $1.00 \pm .00$ |
| Peer = 5 | $\mathbf{1.33} \pm .04$ | $\mathbf{0.60} \pm .00$ | $0.71 \pm .01$ | $\underline{0.60} \pm .00$ | $1.32 \pm .01$ | $\mathbf{0.06} \pm .00$ | $2.5 \pm .00$ | $0.99 \pm .00$ |
| Peer = 10 | $\underline{1.33} \pm .01$ | $\underline{0.61} \pm .00$ | $\mathbf{0.71} \pm .01$ | $\underline{0.60} \pm .01$ | $1.31 \pm .01$ | $\underline{0.07} \pm .00$ | $2.5 \pm .00$ | $1.00 \pm .00$ |
| Peer = 15 | $\underline{1.34} \pm .00$ | $\underline{0.61} \pm .00$ | $\underline{0.72} \pm .01$ | $\underline{0.60} \pm .00$ | $1.31 \pm .00$ | $\underline{0.07} \pm .00$ | $2.5 \pm .00$ | $0.99 \pm .00$ |
| Peer = 20 | $\underline{1.34} \pm .04$ | $\underline{0.61} \pm .00$ | $\underline{0.72} \pm .01$ | $\underline{0.60} \pm .01$ | $1.31 \pm .00$ | $\underline{0.07} \pm .00$ | $2.5 \pm .00$ | $1.00 \pm .00$ |
| $\epsilon = 0$ | $1.34 \pm .01$ | $\underline{0.61} \pm .00$ | $\mathbf{0.71} \pm .01$ | $\mathbf{0.60} \pm .01$ | $1.30 \pm .00$ | $\mathbf{0.06} \pm .00$ | $2.50 \pm .00$ | $0.99 \pm .00$ |
| $\epsilon = 10^{-5}$ | $\mathbf{1.31} \pm .01$ | $\underline{0.61} \pm .00$ | $0.71 \pm .00$ | $\underline{0.60} \pm .01$ | $1.30 \pm .00$ | $\underline{0.07} \pm .00$ | $2.50 \pm .00$ | $1.00 \pm .00$ |
| $\epsilon = 10^{-4}$ | $\underline{1.33} \pm .04$ | $\mathbf{0.60} \pm .00$ | $0.71 \pm .01$ | $0.60 \pm .00$ | $1.30 \pm .00$ | $\mathbf{0.06} \pm .00$ | $2.50 \pm .00$ | $0.99 \pm .00$ |
| $\epsilon = 10^{-3}$ | $\underline{1.33} \pm .04$ | $\mathbf{0.60} \pm .00$ | $\underline{0.72} \pm .01$ | $0.60 \pm .01$ | $1.30 \pm .00$ | $\underline{0.07} \pm .01$ | $2.50 \pm .00$ | $0.99 \pm .00$ |

**Size of $\mathcal{P}$ in SCU:** Intuitively, a proper peer size ($|\mathcal{P}|-1$) should balance high-similarity peers and potential peers that serve for regularization and generalization purposes. In general, the trend across different datasets shows that a too-small peer size may only include high-similarity peers, while a too-large peer size may include too many noisy nodes and incur higher computational overhead. As shown in Table 3, the variation in performance is small, indicating that our method is relatively robust to this parameter. From the extensive experiments on 25 datasets, we recommend a peer size of 5–10 for practical use.

**Initialization parameter $\epsilon$:** We also evaluate whether a non-zero value of $\epsilon$ in the initialization unit of M³-Impute indeed leads to an improvement in imputation accuracy. As shown in Table 3, for YACHT and WINE datasets, the introduction of a non-zero value of $\epsilon$ results in lower MAE scores.

Another insight that we have from Table 3 is that $\epsilon$ should not be set too large, as a large value of $\epsilon$ might impose incorrect weights to the features with missing values. We observe that it is an overall good choice to set $\epsilon$ to $1 \times 10^{-5}$ or $1 \times 10^{-4}$.

## 4.5 RUNNING TIME ANALYSIS

We present a running time comparison in Table 4. The results show that our method is both accurate and time-efficient. For example, for inference with GPU, the time taken to impute *all* the missing values for any dataset we tested is less than one second under the setting of MCAR with 30% missingness. More results can be found in Appendix.

Table 4: Running time (in seconds) for feature imputation using different methods at test time. (C) represents CPU running time and (G) indicates GPU running time.

| Model | Yacht | Wine | Concrete | Housing | Energy | Naval | Kin8nm | Power |
|---|---|---|---|---|---|---|---|---|
| Mean (C) | 0.0006 | 0.0018 | 0.0013 | 0.0018 | 0.0011 | 0.0105 | 0.0037 | 0.0020 |
| kNN (C) | 0.03 | 0.75 | 0.27 | 0.10 | 0.17 | 47.97 | 16.36 | 17.92 |
| Mice (C) | 0.03 | 0.13 | 0.04 | 0.07 | 0.05 | 1.21 | 0.16 | 0.05 |
| Gain (C) | 3.25 | 3.95 | 3.21 | 4.24 | 3.38 | 4.01 | 3.05 | 3.21 |
| HyperImpute (C) | 21.68 | 23.98 | 36.00 | 131.83 | 42.12 | 56.88 | 28.67 | 22.61 |
| Miracle (C) | 3.94 | 12.28 | 8.65 | 6.23 | 5.23 | 75.19 | 35.03 | 32.77 |
| Miwae (C) | 7.63 | 37.14 | 23.44 | 12.13 | 17.95 | 283.64 | 164.71 | 206.09 |
| Grape (C) | 0.05 | 0.26 | 0.12 | 0.10 | 0.09 | 3.32 | 1.14 | 0.63 |
| Grape (G) | 0.02 | 0.02 | 0.02 | 0.02 | 0.02 | 0.19 | 0.07 | 0.05 |
| $M^3$-Impute (C) | 0.05 | 0.43 | 0.18 | 0.14 | 0.13 | 5.09 | 1.55 | 0.78 |
| $M^3$-Impute (G) | 0.02 | 0.02 | 0.04 | 0.04 | 0.04 | 0.56 | 0.19 | 0.12 |

## 4.6 DIFFERENT GNN VARIANTS

We also conduct experiments using different GNN variants such as GraphSAGE, GAT, and GCN. The results in Table 5 indicate that different aggregation mechanisms may introduce varying errors, but our method consistently outperforms its Grape counterpart, demonstrating its effectiveness.

Table 5: MAE for different GNN variants under MCAR setting with 30% missingness.

| | Yacht | Wine | Concrete | Housing | Energy | Naval | Kin8nm | Power |
|---|---|---|---|---|---|---|---|---|
| $M^3$-**Impute** w/ different GNN variants: | | | | | | | | |
| E-GraphSage | **1.33** | **0.60** | **0.71** | **0.60** | **1.32** | **0.06** | **2.50** | **0.99** |
| GCN | 2.04 | 0.97 | 1.76 | 1.62 | 2.37 | 2.10 | 2.50 | 1.66 |
| GAT | 1.56 | 0.95 | 1.01 | 0.78 | 1.39 | 0.31 | 2.50 | 1.02 |
| GraphSage | 1.48 | 0.68 | 1.02 | 0.78 | 1.44 | 0.37 | 2.50 | 1.05 |
| **Grape** w/ different GNN variants: | | | | | | | | |
| E-GraphSage | **1.46** | **0.60** | **0.75** | **0.64** | **1.36** | **0.07** | **2.50** | **1.00** |
| GCN | 2.04 | 1.44 | 2.24 | 2.90 | 3.27 | 2.71 | 2.50 | 1.73 |
| GAT | 2.02 | 0.99 | 1.90 | 2.13 | 3.22 | 2.45 | 2.50 | 1.67 |
| GraphSage | 2.02 | 0.98 | 1.81 | 1.74 | 3.17 | 2.30 | 2.50 | 1.66 |

## 5 CONCLUSION

In this paper, we highlighted the importance of missingness information and presented $M^3$-Impute, a mask-guided representation learning method for missing value imputation. $M^3$-Impute improved the embedding initialization process by considering the relationships between samples and their associated features (including missing ones). In addition, for more effective representation learning, we introduced two novel components in $M^3$-Impute – **FCU** and **SCU**, which explicitly model the missingness information with our novel soft masking schemes to better capture data correlations for imputation. Extensive experiment results on 25 open datasets demonstrate the effectiveness of $M^3$-Impute, where it achieves overall superior performance to popular and state-of-the-art methods, with 20 best and 4 second-best MAE scores on average under three different settings of missing value patterns. For reproducibility purpose, we have included the implementations of $M^3$-Impute and all the baseline models with detailed running instructions in the supplementary material.

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

# A APPENDIX

Table 6: Overview of datasets, which contains continuous features (Cont. F.) and discrete features (Disc. F.).

|  | Concrete | Housing | Wine | Yacht | Energy | Kin8nm | Naval | Power |
|---|---|---|---|---|---|---|---|---|
| # Samples | 1030 | 506 | 1599 | 308 | 768 | 8192 | 11934 | 9568 |
| # Cont. F. | 8 | 12 | 11 | 6 | 8 | 8 | 16 | 4 |
| # Disc. F. | 0 | 1 | 0 | 0 | 0 | 0 | 0 | 0 |

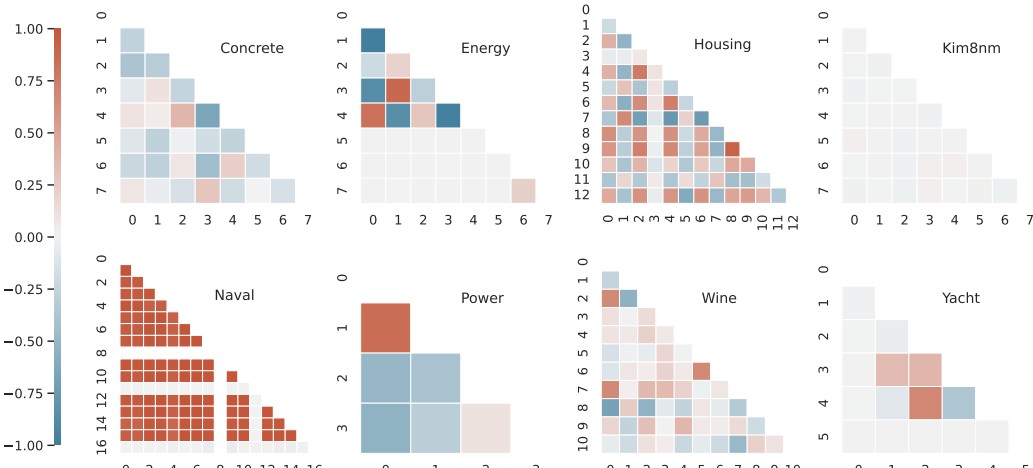

Figure 4: Pearson correlation coefficients of UCI datasets.

In this section, we elaborate on extensive and comprehensive experiment results. We first provide an overview of the dataset details in Section A.1, and present the performance of the imputation methods under different missingness settings, namely MAR and MNAR, in Section A.2. We then provide the comprehensive results across different missingness ratios in Section A.3. For more thorough analysis, we extend our evaluation of M$^3$-Impute on 17 additional datasets, totaling 25 datasets, in Section A.4, and elaborate on the computational resources used in Section A.5. We further assess the quality of imputed values generated by M$^3$-Impute by leveraging them in downstream tasks in Section A.6. Finally, we perform a sensitivity analysis on the hyperparameters of M$^3$-Impute in Section A.7, and provide the implementation details of baselines, including their hyperparameter choices, in Section A.8.

## A.1 DATASET DETAILS

Table 6 presents the statistics of the eight UCI datasets (Dua & Graff, 2017) used throughout Section 4. Figure 4 illustrates the Pearson correlation coefficients among the features. In the Kin8nm dataset, all features are linearly independent, whereas the Naval dataset exhibits strong correlations among its features. Under the MCAR setting, M$^3$-Impute performs comparably to the baseline imputation methods on these two datasets (shown in Table 1). However, in real-world scenarios, features are not always entirely independent or strongly correlated. In the other six datasets, we observe a mix of weakly correlated features along with a few that are strongly correlated. In these cases, M$^3$-Impute consistently outperforms all baseline methods.

## A.2 DETAILED RESULTS OF DIFFERENT MISSINGNESS SETTINGS

We adopt the same procedure outlined in Grape (You et al., 2020) to generate missing values under different settings.

- **MCAR**: An $n \times m$ matrix is sampled from a uniform distribution. Positions with values no greater than the ratio of missingness are viewed as missing and the remaining positions are observable.

Table 7: MAE scores under MAR setting with 30% missingness.

| | Yacht | Wine | Concrete | Housing | Energy | Naval | Kin8nm | Power |
|---|---|---|---|---|---|---|---|---|
| Mean | $2.20 \pm .13$ | $1.09 \pm .05$ | $1.79 \pm .21$ | $2.02 \pm .20$ | $3.26 \pm .36$ | $2.75 \pm .11$ | $\mathbf{2.49} \pm .01$ | $1.81 \pm .25$ |
| Svd | $2.64 \pm .22$ | $1.04 \pm .14$ | $2.32 \pm .06$ | $1.71 \pm .15$ | $3.68 \pm .16$ | $0.52 \pm .11$ | $2.69 \pm .02$ | $2.37 \pm .62$ |
| Spectral | $3.06 \pm .11$ | $0.91 \pm .13$ | $2.12 \pm .17$ | $1.84 \pm .28$ | $2.88 \pm .35$ | $1.29 \pm .47$ | $3.56 \pm .01$ | $3.37 \pm .04$ |
| Mice | $1.79 \pm .10$ | $0.79 \pm .01$ | $1.27 \pm .08$ | $1.22 \pm .05$ | $1.12 \pm .07$ | $\underline{0.27} \pm .01$ | $\underline{2.51} \pm .03$ | $1.16 \pm .11$ |
| Knn | $1.69 \pm .07$ | $\underline{0.66} \pm .07$ | $\underline{0.89} \pm .30$ | $0.89 \pm .12$ | $1.61 \pm .35$ | $\mathbf{0.07} \pm .00$ | $2.94 \pm .01$ | $1.11 \pm .04$ |
| Gain | $2.07 \pm .02$ | $1.13 \pm .20$ | $1.87 \pm .16$ | $0.92 \pm .05$ | $2.26 \pm .14$ | $0.91 \pm .07$ | $2.93 \pm .02$ | $1.42 \pm .01$ |
| Miwae | $2.17 \pm .02$ | $0.98 \pm .02$ | $1.80 \pm .01$ | $1.53 \pm .05$ | $3.91 \pm .04$ | $2.91 \pm .07$ | $2.58 \pm .02$ | $2.05 \pm .01$ |
| Grape | $\underline{1.20} \pm .03$ | $\mathbf{0.60} \pm .00$ | $\mathbf{0.77} \pm .02$ | $\underline{0.66} \pm .01$ | $\underline{1.05} \pm .02$ | $\mathbf{0.07} \pm .01$ | $\mathbf{2.49} \pm .00$ | $1.06 \pm .04$ |
| Miracle | $3.75 \pm .00$ | $0.70 \pm .00$ | $1.94 \pm .00$ | $2.24 \pm .00$ | $3.89 \pm .00$ | $0.36 \pm .00$ | $2.82 \pm .10$ | $\mathbf{0.86} \pm .01$ |
| Hyperimpute | $2.06 \pm .12$ | $0.78 \pm .06$ | $1.30 \pm .15$ | $1.05 \pm .21$ | $1.11 \pm .38$ | $1.01 \pm .18$ | $3.07 \pm .06$ | $1.07 \pm .14$ |
| $M^3$-Impute | $\mathbf{1.09} \pm .03$ | $\mathbf{0.60} \pm .00$ | $\mathbf{0.77} \pm .02$ | $\mathbf{0.60} \pm .00$ | $\mathbf{0.98} \pm .02$ | $\mathbf{0.07} \pm .00$ | $\mathbf{2.49} \pm .00$ | $\underline{1.01} \pm .00$ |

Table 8: MAE scores under MNAR setting with 30% missingness.

| | Yacht | Wine | Concrete | Housing | Energy | Naval | Kin8nm | Power |
|---|---|---|---|---|---|---|---|---|
| Mean | $2.18 \pm .09$ | $1.04 \pm .02$ | $1.80 \pm .09$ | $1.95 \pm .13$ | $3.17 \pm .22$ | $2.60 \pm .07$ | $\underline{2.49} \pm .01$ | $1.76 \pm .14$ |
| Svd | $2.61 \pm .13$ | $1.06 \pm .07$ | $2.24 \pm .05$ | $1.58 \pm .06$ | $3.55 \pm .09$ | $0.53 \pm .05$ | $2.69 \pm .02$ | $2.27 \pm .25$ |
| Spectral | $2.75 \pm .14$ | $1.01 \pm .08$ | $1.86 \pm .03$ | $1.60 \pm .22$ | $2.50 \pm .15$ | $1.35 \pm .21$ | $3.34 \pm .05$ | $3.14 \pm .41$ |
| Mice | $1.91 \pm .10$ | $0.77 \pm .07$ | $1.37 \pm .05$ | $1.22 \pm .06$ | $1.57 \pm .03$ | $\underline{0.21} \pm .07$ | $2.50 \pm .00$ | $1.08 \pm .02$ |
| Knn | $1.92 \pm .10$ | $0.75 \pm .05$ | $1.15 \pm .32$ | $0.95 \pm .11$ | $1.96 \pm .11$ | $\mathbf{0.08} \pm .02$ | $3.06 \pm .02$ | $1.65 \pm .07$ |
| Gain | $2.34 \pm .12$ | $0.92 \pm .05$ | $1.80 \pm .05$ | $1.08 \pm .05$ | $1.92 \pm .06$ | $1.12 \pm .03$ | $2.78 \pm .03$ | $1.22 \pm .03$ |
| Miwae | $2.17 \pm .00$ | $0.99 \pm .01$ | $1.81 \pm .03$ | $1.60 \pm .02$ | $3.63 \pm .06$ | $2.63 \pm .03$ | $2.55 \pm .02$ | $1.95 \pm .03$ |
| Grape | $\underline{1.23} \pm .03$ | $\underline{0.61} \pm .00$ | $\underline{0.73} \pm .01$ | $\underline{0.61} \pm .01$ | $\underline{1.16} \pm .01$ | $\mathbf{0.08} \pm .01$ | $\mathbf{2.46} \pm .01$ | $\underline{1.02} \pm .01$ |
| Miracle | $3.85 \pm .00$ | $0.70 \pm .00$ | $1.87 \pm .00$ | $2.51 \pm .00$ | $3.86 \pm .00$ | $0.30 \pm .00$ | $2.64 \pm .00$ | $1.06 \pm .00$ |
| Hyperimpute | $1.95 \pm .10$ | $0.72 \pm .03$ | $0.88 \pm .02$ | $0.85 \pm .03$ | $1.19 \pm .24$ | $0.85 \pm .04$ | $2.71 \pm .06$ | $1.09 \pm .06$ |
| $M^3$-Impute | $\mathbf{1.15} \pm .02$ | $\mathbf{0.60} \pm .00$ | $\mathbf{0.68} \pm .02$ | $\mathbf{0.54} \pm .01$ | $\mathbf{1.09} \pm .01$ | $\mathbf{0.08} \pm .00$ | $\mathbf{2.46} \pm .00$ | $\mathbf{1.00} \pm .00$ |

- **MAR**: A subset of features is randomly selected to be fully observed. The values for the remaining features are removed according to a logistic model with random weights, using the fully observed feature values as input. The desired rate of missingness is achieved by adjusting the bias term.

- **MNAR**: This is done by first applying the MAR mechanism above. Then, the remaining feature values are masked out using the MCAR mechanism.

In addition to the results for MCAR setting presented in Table 4.2, Tables 7 and 8 present the MAE scores under MAR and MNAR settings, respectively. $M^3$-Impute consistently outperforms all the baseline methods in both scenarios.

A.3 ROBUSTNESS AGAINST VARIOUS MISSINGNESS SETTINGS

Tables 16, 17, and 18 present the performance of various imputation methods under different levels of missingness across MCAR, MAR, and MNAR settings, respectively. $M^3$-Impute achieves the lowest MAE scores in most cases and the second-best MAE scores in the remaining ones.

A.4 FURTHER EVALUATION ON 17 ADDITIONAL DATASETS

Table 9: Overview of 17 additional datasets, which contains continuous features (Cont. F.) and discrete features (Disc. F.).

| | **AIr**foil | **BL**ood | **Wine-White** | **IO**nosphere | **BR**east | **IR**is | **DI**abetes | **PR**otein | **SP**am |
|---|---|---|---|---|---|---|---|---|---|
| # Samples | 1503 | 748 | 4899 | 351 | 569 | 150 | 442 | 45730 | 4601 |
| # Cont. F. | 5 | 4 | 12 | 34 | 30 | 4 | 10 | 9 | 57 |
| # Disc. F. | 1 | 0 | 0 | 0 | 0 | 0 | 0 | 0 | 0 |

| | **LE**tter | **AB**alone | **Ai4i** | **CMC** | **GE**rman | **ST**eel | **LI**bras | **California-H**ousing |
|---|---|---|---|---|---|---|---|---|
| # Samples | 20000 | 4177 | 10000 | 1473 | 1000 | 1941 | 360 | 20640 |
| # Cont. F. | 16 | 7 | 7 | 8 | 13 | 33 | 91 | 9 |
| # Disc. F. | 0 | 1 | 5 | 1 | 7 | 0 | 0 | 0 |

In this experiment, we further evaluate $M^3$-Impute on 17 datasets: Airfoil (Brooks et al., 2014), Blood (Yeh, 2008), Wine-White (Cortez et al., 2009), Ionosphere (Sigillito et al., 1988), Breast

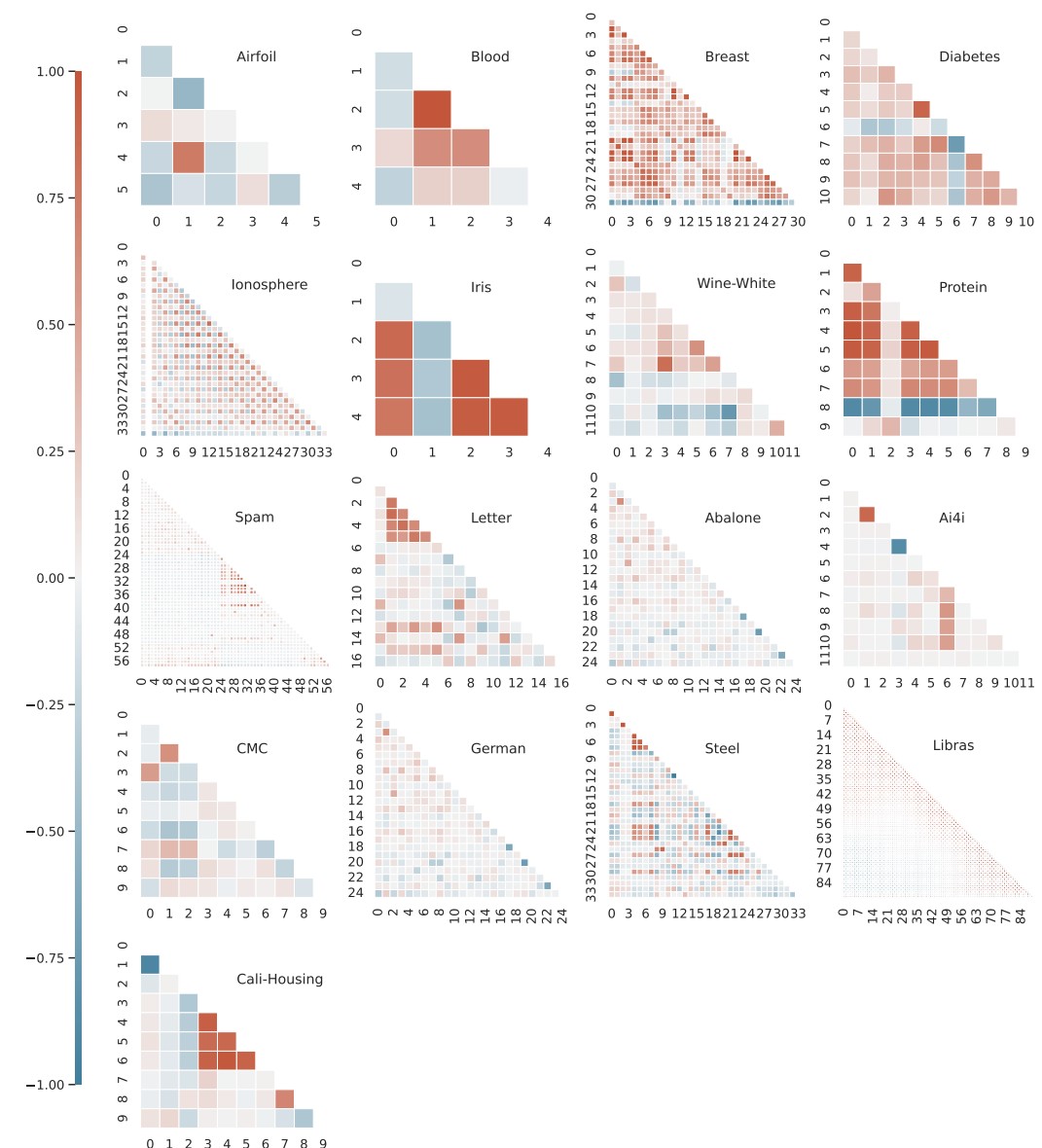

Figure 5: Pearson correlation coefficient of 17 additional datasets.

Cancer (Wolberg et al., 1995), Iris (Fisher, 1936), Diabetes (Efron et al., 2004), Protein, Spam, Letter (Slate, 1991), Abalone, Ai4i, CMC (Lim, 1999), German (Hofmann, 1994), Steel, Libras, and California-housing. An overview of the dataset details is provided in Table 9, and feature correlations are illustrated in Figure 5. We conduct experiments with missingness in data under MCAR, MAR, and MNAR settings, each with missing ratios of {0.1, 0.3, 0.5, 0.7}. Results are presented in Tables 19, 20, and 21 for MCAR, MAR, and MNAR settings, respectively. Across all three types of missingness for the 17 datasets, $M^3$-Impute achieves 13 best and 3 second-best MAE scores on average.

## A.5 COMPUTATIONAL RESOURCES

All our experiments are conducted on a GPU server running Ubuntu 22.04, with PyTorch 2.1.0 and CUDA 12.1. We train and test $M^3$-Impute using a single NVIDIA A100 80G GPU. With the experimental setup described in Section 4.1, the total running time (including both training and testing) for each of the five repeated runs ranged from 1 to 5 hours, depending on the scale of the datasets. In Section 4.5, we report the running time of $M^3$-Impute on eight datasets. Here we extend

Table 10: Running time (in seconds) for feature imputation using different methods at test time. (C) represents CPU running time and (G) indicates GPU running time.

| Model | Protein | Spam | Letter | Libras | California-housing |
|---|---|---|---|---|---|
| Mean (C) | 0.0187 | 0.0151 | 0.0162 | 0.0073 | 0.0090 |
| Knn (C) | 523.68 | 18.50 | 126.43 | 0.63 | 103.14 |
| Svd (C) | 0.61 | 5.29 | 0.61 | 0.16 | 0.48 |
| Mice (C) | 1.36 | 8.99 | 2.21 | 14.45 | 0.35 |
| Spectral (C) | 3.13 | 1.61 | 3.24 | 0.55 | 1.62 |
| Gain (C) | 5.07 | 5.71 | 4.28 | 8.94 | 3.73 |
| HyperImpute (C) | 37.89 | 183.68 | 26.25 | 26.67 | 35.03 |
| Miracle (C) | 173.69 | 91.53 | 110.45 | 34.24 | 81.48 |
| Miwae (C) | 982.18 | 147.75 | 443.49 | 17.80 | 454.16 |
| Grape (C) | 6.97 | 4.33 | 5.25 | 0.84 | 3.13 |
| Grape (G) | 0.37 | 0.24 | 0.28 | 0.05 | 0.18 |
| $M^3$-Impute (C) | 10.27 | 7.53 | 7.80 | 1.51 | 4.61 |
| $M^3$-Impute (G) | 0.99 | 0.69 | 0.76 | 0.19 | 0.51 |

our investigation to larger datasets with more samples and features. As shown in Table 10. $M^3$-Impute also demonstrates efficiency on these datasets. When using a GPU, the time required to impute *all* missing values for any dataset is less than one second under the setting of MCAR with 30% missingness.

## A.6 DOWNSTREAM TASK PERFORMANCE

We further evaluate the quality of imputed values from different imputation methods by performing a downstream label prediction task. In particular, each sample in the eight examined datasets contains a continuous label, and the task is to predict the label using the feature values. Starting with an input data matrix with 30% missingness, we first impute the data using the corresponding imputation methods, and then do linear regression on the completed data matrix to predict labels. As shown in Table 11, $M^3$-Impute consistently achieves good performance across different datasets, with six best performance and two second best, indicating its effectiveness in the missing value imputation.

Table 11: MAE scores of label prediction under the MCAR setting with 30% missingness.

| | Yacht | Wine | Concrete | Housing | Energy | Naval | Kin8nm | Power |
|---|---|---|---|---|---|---|---|---|
| Mean | 9.08 | 0.54 | 10.50 | 4.44 | 4.29 | 0.0065 | **0.18** | 6.31 |
| Svd | 9.12 | 0.55 | 10.90 | 4.26 | 3.40 | 0.0058 | 0.19 | 6.93 |
| Spectral | 8.98 | 0.54 | 10.50 | 4.33 | 3.42 | 0.0057 | 0.19 | 6.67 |
| Mice | 8.95 | 0.54 | 10.20 | 3.99 | 2.84 | 0.0044 | **0.18** | 4.99 |
| Knn | 8.91 | 0.53 | 9.95 | 4.17 | 3.04 | 0.0049 | **0.18** | 5.68 |
| Gain | 9.82 | 0.53 | 10.60 | 4.20 | 2.79 | 0.0060 | **0.18** | 5.08 |
| Miwae | 9.40 | 0.54 | 10.60 | 5.43 | 3.65 | 0.0065 | **0.18** | 5.50 |
| Grape | 8.96 | 0.52 | 9.17 | 3.66 | 2.61 | 0.0038 | **0.18** | 4.83 |
| Miracle | 9.70 | 0.55 | 10.50 | 5.01 | 4.37 | 0.0039 | **0.18** | 4.93 |
| HyperImpute | 9.58 | **0.51** | 9.94 | 3.87 | **2.49** | **0.0032** | **0.18** | **4.69** |
| $M^3$-Impute | **8.82** | **0.51** | 9.04 | 3.60 | 2.57 | 0.0036 | **0.18** | **4.69** |

## A.7 ANALYSIS OF HYPERPARAMETERS IN $M^3$-IMPUTE

In this experiment, we evaluate the sensitivity of $M^3$-Impute to various hyperparameter settings. Tables 12, 13, and 14 summarize the performance of $M^3$-Impute across different hidden dimensions, GNN layer counts, and edge dropout ratios. Overall, the results show that $M^3$-Impute is robust to different hyperparameter settings across the tested datasets. Based on these observations, we

Table 12: MAE of $M^3$-Impute with varying embedding dimensions under the MCAR setting with 30% missingness.

| | Yacht | Wine | Concrete | Housing | Energy | Naval | Kin8nm | Power |
|---|---|---|---|---|---|---|---|---|
| d=32 | 1.38 | 0.63 | 0.89 | 0.71 | 1.32 | 0.18 | **2.50** | 1.02 |
| d=64 | 1.34 | 0.62 | 0.78 | 0.63 | **1.31** | 0.11 | **2.50** | 1.01 |
| d=128 | **1.33** | **0.60** | 0.71 | **0.60** | 1.32 | **0.06** | **2.50** | **0.99** |
| d=256 | 1.37 | 0.61 | **0.68** | **0.60** | 1.33 | **0.06** | **2.50** | **0.99** |

Table 13: MAE of M$^3$-Impute with varying numbers of GNN layers (L) under the MCAR setting with 30% missingness.

| | Yacht | Wine | Concrete | Housing | Energy | Naval | Kin8nm | Power |
|---|---|---|---|---|---|---|---|---|
| L = 1 | 1.43 | 0.62 | 0.82 | 0.65 | **1.30** | 0.12 | **2.50** | 1.01 |
| L = 2 | 1.36 | 0.61 | 0.77 | 0.62 | 1.31 | 0.09 | **2.50** | 1.01 |
| L = 3 | **1.33** | **0.60** | **0.71** | **0.60** | 1.32 | **0.06** | 2.50 | **0.99** |
| L = 4 | 1.35 | 0.60 | 0.71 | 0.61 | 1.32 | **0.06** | 2.50 | **0.99** |

Table 14: MAE of M$^3$-Impute with varying edge dropout ratios under MCAR setting with 30% missingness.

| Drop % | Yacht | Wine | Concrete | Housing | Energy | Naval | Kin8nm | Power |
|---|---|---|---|---|---|---|---|---|
| 90% | 1.53 | 0.66 | 0.94 | 0.69 | 1.35 | 0.13 | **2.50** | 1.03 |
| 70% | 1.35 | 0.62 | 0.75 | 0.62 | 1.33 | 0.08 | **2.50** | 1.00 |
| 50% | **1.33** | **0.60** | **0.71** | **0.60** | 1.32 | **0.06** | 2.50 | **0.99** |
| 30% | 1.38 | 0.61 | 0.73 | 0.60 | **1.31** | **0.06** | 2.50 | 1.00 |

recommend setting the hidden dimension to 128, the number of GNN layers to 3, and the edge dropout ratio to 50% as a general guideline.

## A.8 BASELINE CONFIGURATION

For Mean, Svd, Spectral, and Knn, we follow the widely adopted implementation in Grape (You et al., 2020). For Gain (Yoon et al., 2018), Miwae (Mattei & Frellsen, 2019), Grape (You et al., 2020), Miracle (Kyono et al., 2021), and HyperImpute (Jarrett et al., 2022), we use their official implementations. By default, we follow the optimal parameter settings provided in the original papers. However, we observe that part of the baselines do not perform well with their default parameters on certain datasets. To ensure a fair comparison, we conduct a grid search over the hyperparameters and report the best results achieved across all our experiments. The search ranges for these hyperparameters are detailed in Table 15.

Table 15: Hyperparameter search space.

| Model | Hyperparameters |
|---|---|
| Svd | `rank` = {3, 5, 10, 20}
`max_iters` = {200, 1000, 2000} |
| Spectral | `max_iters` = {100, 200, 500} |
| Mice | `max_iter` = {10, 30, 50, 100} |
| Knn | `K` = {3, 5, 10, 20} |
| Gain | `n_epochs` = {1000, 2000, 3000} |
| Miwae | `n_epochs` = {1000, 2000, 3000}
`K` = {5, 10, 15, 20} |
| Grape | `hidden_dim` = {64, 128, 256}
`edge_dropout_ratio` = {0.1, 0.3, 0.5} |
| Miracle | `n_hidden` = {8, 16, 32, 64}
`reg_lambda` = range(0.1, 1, 0.1)
`reg_beta` = range(0.1, 1, 0.1)
`max_steps` = {500, 1000, 2000} |

Table 16: MAE scores under the **MCAR** setting across different levels of missingness.

|  | Yacht | Wine | Concrete | Housing | Energy | Naval | Kin8nm | Power |
|---|---|---|---|---|---|---|---|---|
| **Missing 10%** | | | | | | | | |
| Mean | 2.22 ± 0.05 | 0.96 ± 0.02 | 1.81 ± 0.02 | 1.84 ± 0.01 | 3.09 ± 0.07 | 2.30 ± 0.01 | 2.50 ± 0.01 | 1.68 ± 0.00 |
| Svd | 1.92 ± 0.16 | 0.88 ± 0.03 | 2.04 ± 0.04 | 1.69 ± 0.11 | 1.75 ± 0.10 | 0.34 ± 0.00 | 5.04 ± 0.06 | 2.26 ± 0.04 |
| Spectral | 2.24 ± 0.12 | 0.76 ± 0.02 | 1.84 ± 0.05 | 1.28 ± 0.04 | 1.76 ± 0.08 | 0.38 ± 0.01 | 2.71 ± 0.02 | 1.77 ± 0.02 |
| Mice | 1.38 ± 0.13 | 0.62 ± 0.01 | 0.97 ± 0.04 | 0.98 ± 0.04 | 1.28 ± 0.07 | 0.13 ± 0.00 | 2.50 ± 0.01 | 1.01 ± 0.01 |
| Knn | 1.40 ± 0.17 | 0.49 ± 0.01 | 0.58 ± 0.05 | 0.74 ± 0.04 | 1.42 ± 0.05 | **0.03 ± 0.00** | 2.53 ± 0.01 | 1.26 ± 0.00 |
| Gain | 2.30 ± 0.04 | 0.83 ± 0.04 | 1.62 ± 0.05 | 1.16 ± 0.05 | 1.95 ± 0.05 | 0.45 ± 0.01 | 2.74 ± 0.02 | 1.22 ± 0.00 |
| Miwae | 2.39 ± 0.00 | 1.01 ± 0.04 | 1.93 ± 0.02 | 1.73 ± 0.01 | 3.30 ± 0.00 | 2.37 ± 0.00 | 2.57 ± 0.00 | 1.72 ± 0.00 |
| Grape | 1.00 ± 0.00 | 0.48 ± 0.00 | 0.45 ± 0.01 | 0.49 ± 0.00 | 1.19 ± 0.00 | 0.05 ± 0.00 | 2.49 ± 0.00 | 0.85 ± 0.03 |
| Miracle | 3.87 ± 0.01 | 0.62 ± 0.00 | 1.63 ± 0.01 | 3.07 ± 0.00 | 4.04 ± 0.00 | 0.12 ± 0.00 | **2.48 ± 0.01** | 0.99 ± 0.00 |
| HyperImpute | 1.50 ± 0.11 | 0.52 ± 0.00 | 0.51 ± 0.04 | 0.75 ± 0.04 | 1.18 ± 0.05 | 0.06 ± 0.04 | 2.50 ± 0.00 | **0.84 ± 0.00** |
| M³-Impute | **0.96 ± 0.00** | **0.47 ± 0.01** | **0.41 ± 0.01** | **0.45 ± 0.00** | **1.15 ± 0.00** | 0.05 ± 0.00 | 2.49 ± 0.00 | 0.84 ± 0.01 |
|  | Yacht | Wine | Concrete | Housing | Energy | Naval | Kin8nm | Power |
| **Missing 30%** | | | | | | | | |
| Mean | 2.09 ± 0.04 | 0.98 ± 0.01 | 1.79 ± 0.01 | 1.85 ± 0.00 | 3.10 ± 0.04 | 2.31 ± 0.00 | 2.50 ± 0.00 | 1.68 ± 0.00 |
| Svd | 2.46 ± 0.16 | 0.92 ± 0.01 | 1.94 ± 0.02 | 1.53 ± 0.03 | 2.24 ± 0.06 | 0.50 ± 0.00 | 3.67 ± 0.06 | 2.33 ± 0.01 |
| Spectral | 2.64 ± 0.11 | 0.91 ± 0.01 | 1.98 ± 0.04 | 1.46 ± 0.03 | 2.26 ± 0.09 | 0.41 ± 0.00 | 2.80 ± 0.01 | 2.13 ± 0.01 |
| Mice | 1.68 ± 0.05 | 0.77 ± 0.00 | 1.34 ± 0.01 | 1.16 ± 0.03 | 1.53 ± 0.04 | 0.20 ± 0.01 | 2.50 ± 0.00 | 1.16 ± 0.01 |
| Knn | 1.67 ± 0.02 | 0.72 ± 0.00 | 1.16 ± 0.03 | 0.95 ± 0.01 | 1.81 ± 0.03 | 0.10 ± 0.00 | 2.77 ± 0.01 | 1.38 ± 0.01 |
| Gain | 2.26 ± 0.11 | 0.86 ± 0.00 | 1.67 ± 0.03 | 1.23 ± 0.02 | 1.99 ± 0.03 | 0.46 ± 0.02 | 2.70 ± 0.00 | 1.31 ± 0.05 |
| Miwae | 2.37 ± 0.01 | 1.00 ± 0.00 | 1.81 ± 0.01 | 1.74 ± 0.04 | 2.79 ± 0.04 | 2.37 ± 0.00 | 2.57 ± 0.00 | 1.72 ± 0.00 |
| Grape | 1.46 ± 0.01 | **0.60 ± 0.00** | 0.75 ± 0.01 | 0.64 ± 0.01 | 1.36 ± 0.01 | 0.07 ± 0.00 | 2.50 ± 0.00 | 1.00 ± 0.00 |
| Miracle | 3.84 ± 0.00 | 0.70 ± 0.00 | 1.71 ± 0.00 | 3.12 ± 0.00 | 3.94 ± 0.00 | 0.18 ± 0.00 | **2.49 ± 0.00** | 1.13 ± 0.00 |
| HyperImpute | 1.76 ± 0.03 | 0.67 ± 0.01 | 0.84 ± 0.02 | 0.82 ± 0.01 | 1.32 ± 0.02 | **0.04 ± 0.00** | 2.58 ± 0.05 | 1.06 ± 0.01 |
| M³-Impute | **1.33 ± 0.04** | **0.60 ± 0.00** | **0.71 ± 0.01** | **0.59 ± 0.00** | **1.31 ± 0.01** | 0.06 ± 0.00 | 2.50 ± 0.00 | **0.99 ± 0.00** |
|  | Yacht | Wine | Concrete | Housing | Energy | Naval | Kin8nm | Power |
| **Missing 50%** | | | | | | | | |
| Mean | 2.12 ± 0.02 | 0.98 ± 0.01 | 1.81 ± 0.01 | 1.84 ± 0.01 | 3.08 ± 0.02 | 2.31 ± 0.00 | **2.50 ± 0.00** | 1.67 ± 0.00 |
| Svd | 3.00 ± 0.11 | 1.18 ± 0.00 | 2.19 ± 0.01 | 1.88 ± 0.01 | 2.88 ± 0.04 | 0.87 ± 0.00 | 3.30 ± 0.01 | 2.92 ± 0.02 |
| Spectral | 3.17 ± 0.13 | 1.13 ± 0.00 | 2.31 ± 0.01 | 1.76 ± 0.03 | 3.03 ± 0.02 | 0.46 ± 0.00 | 3.02 ± 0.00 | 2.98 ± 0.02 |
| Mice | 1.99 ± 0.08 | 0.83 ± 0.00 | 1.59 ± 0.03 | 1.33 ± 0.02 | 2.13 ± 0.12 | 0.31 ± 0.01 | **2.50 ± 0.00** | 1.32 ± 0.01 |
| Knn | 2.08 ± 0.02 | 0.98 ± 0.01 | 1.40 ± 0.02 | 1.37 ± 0.01 | 2.21 ± 0.01 | 0.76 ± 0.01 | 2.65 ± 0.00 | 1.80 ± 0.01 |
| Gain | 2.33 ± 0.03 | 1.18 ± 0.15 | 2.20 ± 0.17 | 1.43 ± 0.09 | 2.58 ± 0.09 | 0.56 ± 0.03 | 2.86 ± 0.06 | 1.36 ± 0.00 |
| Miwae | 2.41 ± 0.01 | 1.02 ± 0.00 | 1.87 ± 0.04 | 1.76 ± 0.01 | 3.23 ± 0.00 | 2.39 ± 0.01 | 2.58 ± 0.00 | 1.73 ± 0.02 |
| Grape | 1.89 ± 0.02 | 0.75 ± 0.01 | 1.24 ± 0.00 | 0.83 ± 0.01 | 1.63 ± 0.01 | 0.09 ± 0.00 | **2.50 ± 0.00** | **1.19 ± 0.00** |
| Miracle | 3.84 ± 0.00 | 0.81 ± 0.00 | 1.80 ± 0.00 | 3.07 ± 0.00 | 3.94 ± 0.00 | 0.24 ± 0.00 | 2.76 ± 0.00 | 1.29 ± 0.00 |
| HyperImpute | 2.07 ± 0.11 | 0.85 ± 0.00 | 1.33 ± 0.08 | 1.06 ± 0.11 | 1.70 ± 0.05 | **0.07 ± 0.00** | 2.96 ± 0.04 | 1.29 ± 0.01 |
| M³-Impute | **1.74 ± 0.01** | **0.74 ± 0.00** | **1.19 ± 0.02** | **0.79 ± 0.01** | **1.57 ± 0.00** | 0.08 ± 0.00 | 2.50 ± 0.00 | **1.19 ± 0.00** |
|  | Yacht | Wine | Concrete | Housing | Energy | Naval | Kin8nm | Power |
| **Missing 70%** | | | | | | | | |
| Mean | 2.16 ± 0.06 | 0.99 ± 0.00 | 1.81 ± 0.01 | 1.83 ± 0.02 | 3.08 ± 0.01 | 2.31 ± 0.00 | 2.50 ± 0.00 | 1.67 ± 0.00 |
| Svd | 3.78 ± 0.06 | 1.63 ± 0.02 | 2.53 ± 0.03 | 2.58 ± 0.07 | 3.65 ± 0.09 | 1.56 ± 0.00 | 3.58 ± 0.00 | 3.88 ± 0.01 |
| Spectral | 4.17 ± 0.10 | 1.67 ± 0.02 | 2.75 ± 0.01 | 2.59 ± 0.05 | 4.00 ± 0.03 | 1.04 ± 0.00 | 3.73 ± 0.01 | 4.33 ± 0.01 |
| Mice | 2.21 ± 0.10 | 0.93 ± 0.01 | 1.72 ± 0.02 | 1.54 ± 0.04 | 2.71 ± 0.15 | 0.53 ± 0.00 | 2.62 ± 0.08 | 1.46 ± 0.01 |
| Knn | 2.62 ± 0.08 | 1.05 ± 0.00 | 1.60 ± 0.01 | 1.43 ± 0.02 | 2.54 ± 0.04 | 1.08 ± 0.00 | 2.84 ± 0.01 | 2.73 ± 0.00 |
| Gain | 3.07 ± 0.08 | 1.61 ± 0.15 | 2.84 ± 0.04 | 3.09 ± 0.04 | 3.83 ± 0.15 | 1.07 ± 0.02 | 3.31 ± 0.21 | 1.51 ± 0.05 |
| Miwae | 2.40 ± 0.01 | 1.02 ± 0.03 | 1.86 ± 0.01 | 1.75 ± 0.00 | 3.20 ± 0.01 | 2.39 ± 0.01 | 2.58 ± 0.00 | 1.73 ± 0.04 |
| Grape | **2.14 ± 0.01** | 0.88 ± 0.01 | 1.64 ± 0.02 | 1.12 ± 0.01 | 2.10 ± 0.01 | 0.17 ± 0.00 | **2.49 ± 0.00** | **1.37 ± 0.00** |
| Miracle | 3.88 ± 0.00 | 0.90 ± 0.00 | 2.23 ± 0.00 | 3.05 ± 0.00 | 3.95 ± 0.00 | 0.48 ± 0.00 | 2.88 ± 0.00 | 1.54 ± 0.00 |
| HyperImpute | 2.49 ± 0.08 | 0.92 ± 0.02 | 1.71 ± 0.01 | 1.12 ± 0.13 | 2.16 ± 0.06 | **0.15 ± 0.00** | 3.15 ± 0.03 | 1.54 ± 0.02 |
| M³-Impute | **2.14 ± 0.00** | **0.87 ± 0.00** | **1.56 ± 0.01** | **1.08 ± 0.00** | **2.05 ± 0.00** | 0.17 ± 0.00 | **2.49 ± 0.00** | **1.37 ± 0.00** |

Table 17: MAE scores under the **MAR** setting across different levels of missingness.

| | Yacht | Wine | Concrete | Housing | Energy | Naval | Kin8nm | Power |
|---|---|---|---|---|---|---|---|---|
| Missing 10% | | | | | | | | |
| Mean | 2.06 ± .16 | 0.97 ± .08 | 1.95 ± .15 | 1.46 ± .14 | 3.44 ± .23 | 2.62 ± .20 | 2.50 ± .02 | 1.82 ± .16 |
| Svd | 2.61 ± .16 | 1.18 ± .07 | 2.26 ± .20 | 1.34 ± .16 | 3.37 ± .28 | 0.65 ± .19 | 2.64 ± .02 | 2.27 ± .29 |
| Spectral | 2.36 ± .32 | 0.86 ± .11 | 1.93 ± .18 | 1.19 ± .10 | 2.24 ± .35 | 0.45 ± .22 | 3.12 ± .04 | 2.15 ± .26 |
| Mice | 1.26 ± .17 | 0.65 ± .02 | 1.03 ± .08 | 1.05 ± .08 | 0.89 ± .09 | 0.18 ± .01 | 2.48 ± .01 | 0.98 ± .09 |
| Knn | 1.62 ± .51 | 0.51 ± .02 | 0.47 ± .10 | 0.74 ± .02 | 1.50 ± .10 | 0.03 ± .01 | 2.92 ± .03 | 0.84 ± .10 |
| Gain | 2.00 ± .12 | 0.79 ± .02 | 1.72 ± .04 | 0.91 ± .13 | 1.51 ± .06 | 0.54 ± .16 | 2.77 ± .03 | 1.24 ± .08 |
| Miwae | 2.06 ± .01 | 1.02 ± .00 | 1.88 ± .00 | 1.56 ± .01 | 3.82 ± .01 | 2.92 ± .00 | 2.57 ± .01 | 2.06 ± .01 |
| Grape | 0.75 ± .00 | **0.49** ± .01 | 0.42 ± .01 | 0.51 ± .02 | 0.89 ± .01 | 0.07 ± .02 | **2.46** ± .00 | 0.70 ± .03 |
| Miracle | 3.78 ± .00 | 0.58 ± .04 | 1.65 ± .00 | 2.21 ± .00 | 3.45 ± .00 | 0.16 ± .01 | 2.52 ± .01 | 0.81 ± .00 |
| HyperImpute | 1.22 ± .13 | 0.51 ± .01 | 0.49 ± .03 | 0.82 ± .16 | **0.70** ± .14 | **0.02** ± .02 | 2.56 ± .04 | 0.77 ± .06 |
| M$^3$-Impute | **0.70** ± .01 | **0.49** ± .00 | **0.39** ± .01 | **0.44** ± .01 | 0.85 ± .00 | 0.07 ± .00 | **2.46** ± .00 | **0.69** ± .01 |
| | Yacht | Wine | Concrete | Housing | Energy | Naval | Kin8nm | Power |
| Missing 30% | | | | | | | | |
| Mean | 2.20 ± .13 | 1.09 ± .05 | 1.79 ± .21 | 2.02 ± .20 | 3.26 ± .36 | 2.75 ± .11 | **2.49** ± .01 | 1.81 ± .25 |
| Svd | 2.64 ± .22 | 1.04 ± .14 | 2.32 ± .06 | 1.71 ± .15 | 3.68 ± .16 | 0.52 ± .11 | 2.69 ± .02 | 2.37 ± .62 |
| Spectral | 3.06 ± .11 | 0.91 ± .13 | 2.12 ± .17 | 1.84 ± .28 | 2.88 ± .35 | 1.29 ± .47 | 3.56 ± .01 | 3.37 ± .04 |
| Mice | 1.79 ± .10 | 0.79 ± .01 | 1.27 ± .08 | 1.22 ± .05 | 1.12 ± .07 | 0.27 ± .01 | 2.51 ± .03 | 1.16 ± .11 |
| Knn | 1.69 ± .07 | 0.66 ± .07 | 0.89 ± .30 | 0.89 ± .12 | 1.61 ± .35 | **0.07** ± .00 | 2.94 ± .01 | 1.11 ± .04 |
| Gain | 2.07 ± .02 | 1.13 ± .20 | 1.87 ± .16 | 0.92 ± .05 | 2.26 ± .14 | 0.91 ± .05 | 2.93 ± .02 | 1.42 ± .01 |
| Miwae | 2.17 ± .01 | 0.98 ± .01 | 1.80 ± .04 | 1.54 ± .00 | 3.91 ± .01 | 2.80 ± .01 | 2.58 ± .01 | 2.05 ± .02 |
| Grape | 1.20 ± .03 | **0.60** ± .00 | **0.77** ± .02 | 0.66 ± .01 | 1.05 ± .02 | **0.07** ± .01 | **2.49** ± .00 | 1.06 ± .04 |
| Miracle | 3.75 ± .00 | 0.70 ± .00 | 1.94 ± .00 | 2.24 ± .00 | 3.89 ± .00 | 0.36 ± .00 | 2.82 ± .10 | **0.86** ± .01 |
| Hyperimpute | 2.06 ± .12 | 0.78 ± .06 | 1.30 ± .15 | 1.05 ± .21 | 1.11 ± .38 | 1.01 ± .18 | 3.07 ± .06 | 1.07 ± .14 |
| M$^3$-Impute | **1.09** ± .03 | **0.60** ± .00 | **0.77** ± .02 | **0.60** ± .00 | **0.98** ± .02 | **0.07** ± .00 | **2.49** ± .00 | 1.01 ± .00 |
| | Yacht | Wine | Concrete | Housing | Energy | Naval | Kin8nm | Power |
| Missing 50% | | | | | | | | |
| Mean | 2.20 ± .13 | 0.98 ± .06 | 1.74 ± .09 | 1.92 ± .05 | 3.30 ± .12 | 2.80 ± .05 | **2.49** ± .01 | 1.86 ± .03 |
| Svd | 2.95 ± .11 | 1.06 ± .04 | 2.63 ± .42 | 1.78 ± .14 | 3.69 ± .14 | 0.59 ± .14 | 2.83 ± .03 | 2.64 ± .27 |
| Spectral | 3.41 ± .09 | 1.35 ± .02 | 2.14 ± .02 | 2.10 ± .30 | 3.62 ± .16 | 1.98 ± .25 | 3.84 ± .01 | 4.02 ± .32 |
| Mice | 2.15 ± .09 | 0.87 ± .05 | 1.56 ± .08 | 1.45 ± .03 | 1.96 ± .04 | 0.25 ± .08 | 2.61 ± .08 | 1.35 ± .14 |
| Knn | 2.45 ± .19 | 0.90 ± .09 | **1.04** ± .11 | 1.14 ± .26 | 1.64 ± .39 | 0.07 ± .01 | 3.00 ± .01 | 1.44 ± .23 |
| Gain | 3.40 ± .08 | 1.60 ± .33 | 2.13 ± .24 | 1.95 ± .10 | 3.04 ± .35 | 1.02 ± .05 | 3.08 ± .08 | 1.69 ± .09 |
| Miwae | 2.31 ± .01 | 1.09 ± .03 | 1.74 ± .03 | 1.84 ± .03 | 3.45 ± .00 | 2.89 ± .01 | 2.57 ± .01 | 1.95 ± .01 |
| Grape | **2.07** ± .00 | **0.81** ± .00 | **1.17** ± .01 | 0.91 ± .01 | 1.73 ± .03 | 0.10 ± .01 | **2.49** ± .00 | **1.35** ± .00 |
| Miracle | 3.98 ± .00 | 0.87 ± .00 | 2.50 ± .00 | 2.41 ± .00 | 4.08 ± .00 | 0.63 ± .00 | 2.86 ± .00 | 1.55 ± .00 |
| HyperImpute | 2.47 ± .12 | **0.81** ± .06 | 1.60 ± .03 | 1.01 ± .09 | **1.50** ± .22 | **0.04** ± .02 | 3.19 ± .01 | **1.27** ± .15 |
| M$^3$-Impute | **2.07** ± .00 | **0.81** ± .00 | **1.17** ± .01 | **0.84** ± .00 | 1.64 ± .03 | 0.10 ± .00 | **2.49** ± .00 | **1.35** ± .00 |
| | Yacht | Wine | Concrete | Housing | Energy | Naval | Kin8nm | Power |
| Missing 70% | | | | | | | | |
| Mean | 2.13 ± .12 | 1.01 ± .03 | 1.85 ± .03 | 1.91 ± .06 | 3.19 ± .07 | 3.11 ± .48 | **2.49** ± .01 | 1.79 ± .07 |
| Svd | 3.10 ± .21 | 1.36 ± .03 | 2.47 ± .20 | 2.44 ± .21 | 4.33 ± .38 | 0.82 ± .03 | 3.13 ± .03 | 2.70 ± .22 |
| Spectral | 3.68 ± .27 | 1.49 ± .17 | 2.55 ± .22 | 2.49 ± .15 | 3.97 ± .01 | 2.82 ± .69 | 4.10 ± .01 | 4.23 ± .34 |
| Mice | 2.28 ± .12 | 0.97 ± .03 | 1.76 ± .08 | 1.81 ± .01 | 2.81 ± .18 | 0.50 ± .03 | 2.62 ± .02 | 1.38 ± .17 |
| Knn | **2.02** ± .34 | 1.15 ± .05 | **1.54** ± .06 | 1.63 ± .25 | **1.65** ± .28 | 0.19 ± .04 | 2.98 ± .01 | **1.23** ± .06 |
| Gain | 3.64 ± .27 | 1.94 ± .05 | 2.59 ± .11 | 2.74 ± .09 | 4.40 ± .04 | 0.69 ± .05 | 4.00 ± .04 | 2.41 ± .42 |
| Miwae | 2.31 ± .01 | 1.08 ± .01 | 1.89 ± .00 | 1.84 ± .02 | 3.18 ± .00 | 2.95 ± .00 | 2.58 ± .00 | 1.94 ± .01 |
| Grape | 2.06 ± .00 | 0.94 ± .01 | 1.69 ± .03 | 1.20 ± .01 | 2.23 ± .02 | 0.17 ± .00 | **2.49** ± .00 | 1.42 ± .00 |
| Miracle | 3.99 ± .00 | 0.96 ± .00 | 2.70 ± .04 | 2.83 ± .00 | 3.82 ± .00 | 0.24 ± .01 | 2.89 ± .00 | 1.61 ± .00 |
| HyperImpute | 2.56 ± .04 | 0.96 ± .02 | 1.93 ± .04 | 1.28 ± .06 | 2.43 ± .12 | **0.12** ± .09 | 3.22 ± .02 | 1.36 ± .18 |
| M$^3$-Impute | 2.06 ± .00 | **0.92** ± .00 | 1.68 ± .01 | **1.13** ± .01 | 2.16 ± .00 | 0.17 ± .00 | **2.49** ± .00 | 1.42 ± .00 |

Table 18: MAE scores under the **MNAR** setting across different levels of missingness.

| | Yacht | Wine | Concrete | Housing | Energy | Naval | Kin8nm | Power |
|---|---|---|---|---|---|---|---|---|
| **Missing 10%** | | | | | | | | |
| Mean | 2.20 ± .10 | 0.97 ± .02 | 1.88 ± .01 | 1.75 ± .12 | 3.12 ± .07 | 2.48 ± .05 | **2.51** ± .01 | 1.72 ± .05 |
| Svd | 2.64 ± .01 | 1.09 ± .06 | 2.14 ± .03 | 1.31 ± .03 | 3.45 ± .23 | 0.75 ± .03 | 2.63 ± .01 | 2.08 ± .05 |
| Spectral | 2.23 ± .03 | 0.83 ± .05 | 1.97 ± .02 | 1.18 ± .05 | 1.86 ± .05 | 0.30 ± .05 | 3.00 ± .01 | 2.36 ± .22 |
| Mice | 1.41 ± .05 | 0.65 ± .01 | 1.07 ± .03 | 0.93 ± .01 | 1.33 ± .14 | 0.12 ± .00 | 2.51 ± .02 | 1.04 ± .02 |
| Knn | 1.44 ± .14 | 0.53 ± .03 | 0.54 ± .04 | 0.60 ± .04 | 1.64 ± .11 | **0.03** ± .00 | 3.00 ± .03 | 1.55 ± .02 |
| Gain | 2.39 ± .03 | 0.86 ± .01 | 1.66 ± .04 | 1.05 ± .08 | 1.94 ± .04 | 0.42 ± .01 | 2.74 ± .01 | 1.23 ± .02 |
| Miwae | 2.23 ± .01 | 1.01 ± .02 | 1.92 ± .03 | 1.50 ± .01 | 3.25 ± .01 | 2.50 ± .01 | 2.59 ± .00 | 1.83 ± .01 |
| Grape | 1.13 ± .01 | **0.49** ± .00 | 0.46 ± .01 | 0.55 ± .01 | 1.14 ± .00 | 0.04 ± .00 | 2.51 ± .00 | 0.88 ± .00 |
| Miracle | 3.87 ± .00 | 0.62 ± .00 | 1.70 ± .04 | 2.51 ± .00 | 4.05 ± .00 | 0.11 ± .01 | 2.51 ± .02 | 1.01 ± .03 |
| HyperImpute | 1.51 ± .07 | 0.55 ± .01 | 0.58 ± .02 | 0.73 ± .04 | 1.11 ± .06 | **0.03** ± .00 | 2.51 ± .01 | 0.85 ± .00 |
| M³-Impute | **1.08** ± .00 | **0.49** ± .00 | **0.44** ± .01 | **0.50** ± .01 | **1.10** ± .00 | 0.04 ± .00 | 2.51 ± .00 | **0.84** ± .01 |
| | Yacht | Wine | Concrete | Housing | Energy | Naval | Kin8nm | Power |
| **Missing 30%** | | | | | | | | |
| Mean | 2.18 ± .09 | 1.04 ± .02 | 1.80 ± .09 | 1.95 ± .13 | 3.17 ± .22 | 2.60 ± .07 | 2.49 ± .01 | 1.76 ± .14 |
| Svd | 2.61 ± .13 | 1.06 ± .07 | 2.24 ± .05 | 1.58 ± .06 | 3.55 ± .09 | 0.53 ± .05 | 2.69 ± .02 | 2.27 ± .25 |
| Spectral | 2.75 ± .14 | 1.01 ± .08 | 1.86 ± .03 | 1.60 ± .22 | 2.50 ± .15 | 1.35 ± .21 | 3.34 ± .00 | 3.14 ± .41 |
| Mice | 1.91 ± .10 | 0.77 ± .07 | 1.37 ± .05 | 1.22 ± .06 | 1.57 ± .03 | 0.21 ± .07 | 2.50 ± .00 | 1.08 ± .02 |
| Knn | 1.92 ± .10 | 0.75 ± .05 | 1.15 ± .32 | 0.95 ± .11 | 1.96 ± .11 | **0.08** ± .02 | 3.06 ± .02 | 1.65 ± .07 |
| Gain | 2.34 ± .12 | 0.92 ± .05 | 1.80 ± .05 | 1.08 ± .05 | 1.92 ± .06 | 1.12 ± .03 | 2.78 ± .13 | 1.22 ± .03 |
| Miwae | 2.17 ± .00 | 0.99 ± .01 | 1.81 ± .03 | 1.60 ± .02 | 3.63 ± .00 | 2.63 ± .03 | 2.55 ± .02 | 1.95 ± .03 |
| Grape | 1.23 ± .03 | 0.61 ± .00 | 0.73 ± .01 | 0.61 ± .01 | 1.16 ± .01 | **0.08** ± .01 | 2.46 ± .01 | 1.02 ± .01 |
| Miracle | 3.85 ± .00 | 0.70 ± .00 | 1.87 ± .00 | 2.51 ± .00 | 3.86 ± .00 | 0.30 ± .00 | 2.64 ± .00 | 1.06 ± .00 |
| Hyperimpute | 1.95 ± .10 | 0.72 ± .03 | 0.88 ± .02 | 0.85 ± .03 | 1.19 ± .24 | 0.85 ± .04 | 2.71 ± .06 | 1.09 ± .06 |
| M³-Impute | **1.15** ± .02 | **0.60** ± .00 | **0.68** ± .02 | **0.54** ± .01 | **1.09** ± .01 | **0.08** ± .00 | 2.46 ± .00 | **1.00** ± .00 |
| | Yacht | Wine | Concrete | Housing | Energy | Naval | Kin8nm | Power |
| **Missing 50%** | | | | | | | | |
| Mean | 2.17 ± .08 | 1.01 ± .06 | 1.85 ± .09 | 1.89 ± .13 | 3.27 ± .10 | 2.66 ± .11 | 2.49 ± .01 | 1.77 ± .06 |
| Svd | 3.08 ± .10 | 1.15 ± .12 | 2.46 ± .15 | 1.90 ± .16 | 3.56 ± .25 | 0.58 ± .09 | 2.83 ± .02 | 2.62 ± .12 |
| Spectral | 3.38 ± .04 | 1.31 ± .02 | 2.36 ± .15 | 1.91 ± .11 | 3.49 ± .29 | 1.99 ± .08 | 3.82 ± .01 | 4.19 ± .35 |
| Mice | 2.17 ± .07 | 0.90 ± .07 | 1.59 ± .06 | 1.50 ± .14 | 2.39 ± .35 | 0.27 ± .02 | 2.62 ± .02 | 1.35 ± .09 |
| Knn | 2.31 ± .10 | 1.02 ± .07 | 1.35 ± .10 | 1.55 ± .33 | 2.51 ± .19 | 0.39 ± .18 | 3.03 ± .01 | 1.69 ± .26 |
| Gain | 3.22 ± .07 | 1.26 ± .12 | 1.96 ± .08 | 1.76 ± .07 | 3.16 ± .13 | 0.74 ± .09 | 3.12 ± .13 | 1.72 ± .24 |
| Miwae | 2.32 ± .00 | 1.08 ± .02 | 1.78 ± .02 | 1.87 ± .02 | 3.40 ± .00 | 2.80 ± .02 | 2.58 ± .00 | 1.91 ± .03 |
| Grape | **2.09** ± .00 | 0.82 ± .00 | 1.23 ± .00 | 0.90 ± .01 | 1.69 ± .01 | 0.10 ± .00 | 2.49 ± .00 | 1.33 ± .00 |
| Miracle | 3.97 ± .00 | 0.88 ± .00 | 2.43 ± .00 | 2.53 ± .00 | 3.81 ± .00 | 0.59 ± .00 | 2.84 ± .05 | 1.55 ± .00 |
| HyperImpute | 2.39 ± .10 | 0.83 ± .06 | 1.56 ± .04 | 0.99 ± .06 | **1.62** ± .18 | **0.05** ± .02 | 3.17 ± .05 | **1.29** ± .12 |
| M³-Impute | **2.09** ± .00 | **0.81** ± .00 | **1.21** ± .01 | **0.87** ± .01 | **1.62** ± .01 | 0.10 ± .00 | 2.48 ± .00 | 1.32 ± .00 |
| | Yacht | Wine | Concrete | Housing | Energy | Naval | Kin8nm | Power |
| **Missing 70%** | | | | | | | | |
| Mean | 2.25 ± .21 | 1.01 ± .02 | 1.83 ± .08 | 1.96 ± .10 | 3.22 ± .13 | 2.81 ± .08 | **2.49** ± .00 | 1.78 ± .05 |
| Svd | 3.11 ± .05 | 1.38 ± .03 | 2.68 ± .21 | 2.32 ± .15 | 4.19 ± .30 | 1.79 ± .39 | 3.16 ± .03 | 2.87 ± .22 |
| Spectral | 3.63 ± .10 | 1.68 ± .11 | 2.73 ± .04 | 2.29 ± .26 | 3.68 ± .18 | 2.83 ± .66 | 4.15 ± .01 | 4.35 ± .30 |
| Mice | 2.22 ± .17 | 0.93 ± .03 | 1.80 ± .03 | 1.75 ± .15 | 2.66 ± .29 | 0.56 ± .04 | 2.64 ± .02 | 1.41 ± .13 |
| Knn | 2.09 ± .31 | 1.15 ± .03 | 1.92 ± .13 | 1.80 ± .23 | 2.36 ± .39 | 0.83 ± .10 | 3.02 ± .00 | 1.60 ± .07 |
| Gain | 3.70 ± .31 | 1.66 ± .19 | 2.57 ± .09 | 2.77 ± .14 | 4.27 ± .08 | 0.42 ± .01 | 3.56 ± .27 | 1.89 ± .28 |
| Miwae | 2.32 ± .01 | 1.07 ± .02 | 1.90 ± .04 | 1.83 ± .02 | 3.17 ± .02 | 2.92 ± .03 | 2.57 ± .00 | 1.90 ± .00 |
| Grape | **2.08** ± .00 | 0.94 ± .01 | 1.73 ± .03 | 1.22 ± .01 | 2.27 ± .01 | 0.18 ± .00 | **2.49** ± .00 | 1.43 ± .00 |
| Miracle | 3.99 ± .00 | 0.95 ± .00 | 2.66 ± .02 | 2.88 ± .00 | 2.85 ± .00 | 0.91 ± .01 | 2.82 ± .00 | 1.61 ± .00 |
| HyperImpute | 2.42 ± .16 | 0.94 ± .03 | 1.86 ± .05 | 1.22 ± .02 | 2.41 ± .06 | **0.14** ± .00 | 3.19 ± .02 | **1.36** ± .13 |
| M³-Impute | **2.08** ± .00 | **0.92** ± .00 | **1.70** ± .02 | **1.15** ± .02 | **2.19** ± .02 | 0.18 ± .00 | **2.49** ± .00 | 1.41 ± .00 |

Table 19: MAE scores under the **MCAR** setting across different levels of missingness on the extra 17 datasets. **AI** is short for **AI**rfoil. Please refer to Table 9 for dataset names.

| Dataset | Mean | Knn | Svd | Mice | Spectral | HI | Gain | Miracle | Miwae | Grape | M³-Impute |
|---|---|---|---|---|---|---|---|---|---|---|---|
| **Missing 10%** | | | | | | | | | | | |
| AI | 2.32 ± .03 | 1.59 ± .04 | 2.67 ± .08 | 1.83 ± .01 | 2.16 ± .03 | 0.59 ± .03 | 2.18 ± .02 | 1.81 ± .01 | 2.30 ± .02 | 0.61 ± .00 | **0.55** ± .01 |
| BL | 1.16 ± .02 | 0.96 ± .03 | 0.90 ± .02 | 0.61 ± .02 | 0.90 ± .05 | 0.53 ± .02 | 1.26 ± .07 | 1.48 ± .05 | 1.98 ± .05 | **0.51** ± .00 | **0.51** ± .00 |
| WW | 0.76 ± .01 | **0.38** ± .00 | 0.86 ± .01 | 0.50 ± .00 | 0.69 ± .01 | 0.44 ± .01 | 0.70 ± .02 | 0.49 ± .00 | 0.78 ± .01 | 0.43 ± .00 | 0.42 ± .00 |
| IO | 2.00 ± .02 | 1.02 ± .03 | 1.21 ± .05 | 1.20 ± .06 | 1.30 ± .03 | 1.14 ± .04 | 1.39 ± .06 | 5.66 ± .00 | 5.12 ± .02 | 1.04 ± .01 | **0.96** ± .02 |
| BR | 1.08 ± .02 | 0.50 ± .01 | 0.57 ± .00 | **0.25** ± .01 | 0.31 ± .00 | 0.28 ± .00 | 0.47 ± .02 | 1.40 ± .00 | 1.90 ± .02 | 0.31 ± .00 | 0.30 ± .02 |
| IR | 2.18 ± .09 | 1.25 ± .22 | 1.67 ± .25 | 0.94 ± .13 | 1.43 ± .22 | 0.97 ± .2 | 1.27 ± .17 | 3.20 ± .00 | 4.80 ± .25 | 0.71 ± .01 | **0.69** ± .02 |
| DI | 1.80 ± .11 | 1.51 ± .07 | 1.68 ± .04 | 1.17 ± .08 | 1.38 ± .07 | 1.09 ± .04 | 1.47 ± .12 | 2.82 ± .00 | 5.23 ± .16 | 1.12 ± .02 | **1.08** ± .02 |
| PR | 0.91 ± .01 | 0.26 ± .00 | 0.98 ± .00 | 0.28 ± .00 | 0.80 ± .00 | **0.20** ± .00 | 0.51 ± .00 | 0.27 ± .00 | 0.94 ± .01 | 0.22 ± .02 | **0.20** ± .00 |
| SP | 0.23 ± .00 | **0.14** ± .01 | 0.30 ± .01 | 0.19 ± .00 | 0.16 ± .00 | 0.17 ± .02 | 0.22 ± .01 | 0.19 ± .01 | 0.16 ± .00 | 0.17 ± .00 | 0.16 ± .01 |
| LE | 1.28 ± .01 | **0.32** ± .00 | 1.25 ± .01 | 0.88 ± .00 | 1.26 ± .01 | 0.49 ± .00 | 1.07 ± .01 | 0.76 ± .01 | 1.37 ± .01 | 0.42 ± .00 | 0.41 ± .01 |
| AB | 2.50 ± .03 | 2.06 ± .06 | 2.41 ± .04 | 2.01 ± .02 | 2.29 ± .02 | **1.71** ± .14 | 2.23 ± .05 | 3.68 ± .00 | 2.57 ± .03 | 1.84 ± .03 | **1.71** ± .02 |
| A4 | 1.06 ± .02 | 1.08 ± .02 | 1.14 ± .02 | 0.78 ± .02 | 1.22 ± .02 | **0.67** ± .02 | 0.81 ± .04 | 1.09 ± .04 | 0.71 ± .01 | 0.70 ± .00 | 0.70 ± .02 |
| CM | 2.31 ± .07 | 2.19 ± .11 | 2.43 ± .10 | 1.97 ± .08 | 2.62 ± .05 | 1.98 ± .15 | 2.35 ± .17 | 1.86 ± .00 | 2.35 ± .08 | 1.75 ± .01 | **1.68** ± .02 |
| GE | 2.50 ± .03 | 2.06 ± .06 | 2.41 ± .04 | 2.01 ± .02 | 2.29 ± .02 | 1.71 ± .14 | 2.23 ± .05 | 3.68 ± .00 | 2.57 ± .03 | 1.85 ± .02 | **1.68** ± .02 |
| ST | 1.78 ± .02 | 0.55 ± .00 | 1.27 ± .01 | 0.70 ± .02 | 0.87 ± .01 | 0.59 ± .00 | 0.96 ± .02 | 0.74 ± .00 | 1.68 ± .02 | 0.34 ± .01 | **0.29** ± .01 |
| LI | 1.82 ± .01 | 0.19 ± .01 | 0.36 ± .00 | **0.05** ± .00 | 0.14 ± .00 | 0.06 ± .00 | 0.37 ± .00 | 6.05 ± .00 | 2.16 ± .01 | 0.10 ± .00 | 0.10 ± .00 |
| CH | 1.13 ± .00 | 0.78 ± .01 | 1.30 ± .00 | 0.56 ± .00 | 1.26 ± .01 | 0.42 ± .00 | 1.07 ± .03 | 0.53 ± .00 | 1.15 ± .01 | **0.40** ± .00 | **0.40** ± .00 |
| **Missing 30%** | | | | | | | | | | | |
| AI | 2.32 ± .05 | 2.18 ± .04 | 2.76 ± .05 | 1.97 ± .04 | 2.30 ± .07 | 1.09 ± .02 | 2.22 ± .06 | 1.97 ± .00 | 2.36 ± .06 | 1.16 ± .02 | **1.09** ± .03 |
| BL | 1.14 ± .01 | 0.93 ± .01 | 0.97 ± .04 | 0.69 ± .01 | 0.94 ± .03 | 0.63 ± .02 | 1.26 ± .04 | 1.50 ± .00 | 2.03 ± .05 | 0.68 ± .00 | 0.67 ± .00 |
| WW | 0.76 ± .00 | 0.64 ± .01 | 0.87 ± .00 | 0.61 ± .01 | 0.78 ± .01 | 0.55 ± .00 | 0.73 ± .01 | 0.58 ± .00 | 0.77 ± .00 | 0.52 ± .00 | 0.52 ± .00 |
| IO | 2.01 ± .03 | 1.07 ± .03 | 1.26 ± .03 | 1.37 ± .03 | 1.38 ± .02 | 1.18 ± .04 | 1.50 ± .01 | 5.56 ± .00 | 5.14 ± .06 | 1.08 ± .01 | **1.01** ± .01 |
| BR | 1.06 ± .00 | 0.53 ± .01 | 0.58 ± .00 | 0.34 ± .01 | 0.38 ± .00 | 0.33 ± .01 | 0.51 ± .01 | 1.42 ± .00 | 1.89 ± .02 | 0.37 ± .00 | 0.36 ± .01 |
| IR | 2.15 ± .09 | 1.54 ± .22 | 1.70 ± .07 | 1.07 ± .09 | 1.48 ± .13 | 1.04 ± .11 | 1.29 ± .07 | 3.22 ± .00 | 4.60 ± .17 | **0.82** ± .00 | **0.82** ± .00 |
| DI | 1.78 ± .03 | 1.71 ± .04 | 1.76 ± .02 | 1.29 ± .05 | 1.48 ± .03 | **1.17** ± .02 | 1.47 ± .06 | 2.69 ± .00 | 5.05 ± .04 | 1.31 ± .00 | 1.29 ± .01 |
| PR | 0.91 ± .00 | 0.58 ± .01 | 1.00 ± .00 | 0.33 ± .00 | 1.14 ± .00 | 0.25 ± .00 | 0.72 ± .06 | 0.32 ± .00 | 0.94 ± .00 | 0.25 ± .02 | **0.24** ± .00 |
| SP | 0.23 ± .00 | 0.17 ± .00 | 0.31 ± .00 | 0.22 ± .00 | 0.16 ± .00 | **0.16** ± .00 | 0.21 ± .00 | 0.19 ± .00 | 0.16 ± .00 | 0.17 ± .01 | 0.16 ± .00 |
| LE | 1.28 ± .00 | 0.89 ± .01 | 1.29 ± .00 | 1.00 ± .00 | 1.75 ± .01 | 0.61 ± .01 | 1.09 ± .01 | 1.06 ± .01 | 1.33 ± .04 | 0.53 ± .00 | 0.52 ± .00 |
| AB | 2.52 ± .02 | 2.34 ± .02 | 2.60 ± .04 | 2.26 ± .04 | 2.45 ± .02 | 2.05 ± .25 | 2.27 ± .09 | 3.67 ± .00 | 2.59 ± .01 | 2.01 ± .00 | **1.84** ± .02 |
| A4 | 1.07 ± .00 | 1.17 ± .00 | 1.18 ± .01 | 0.87 ± .01 | 1.58 ± .01 | **0.75** ± .02 | 1.03 ± .02 | 0.84 ± .02 | 1.12 ± .01 | 0.79 ± .00 | 0.76 ± .00 |
| CM | 2.35 ± .03 | 2.32 ± .04 | 2.52 ± .05 | 2.06 ± .02 | 2.96 ± .01 | 1.91 ± .03 | 2.33 ± .15 | 2.01 ± .00 | 2.37 ± .04 | 1.87 ± .00 | **1.81** ± .01 |
| GE | 2.52 ± .02 | 2.34 ± .02 | 2.60 ± .04 | 2.26 ± .04 | 2.45 ± .02 | 2.05 ± .25 | 2.27 ± .09 | 3.67 ± .00 | 2.59 ± .01 | 2.01 ± .01 | **1.87** ± .02 |
| ST | 1.80 ± .01 | 0.78 ± .02 | 1.37 ± .01 | 0.95 ± .02 | 1.10 ± .01 | 0.72 ± .01 | 1.03 ± .02 | 0.95 ± .03 | 1.70 ± .01 | 0.45 ± .00 | **0.39** ± .00 |
| LI | 1.82 ± .01 | 0.25 ± .00 | 0.37 ± .00 | **0.11** ± .00 | 0.18 ± .00 | 0.11 ± .00 | 0.46 ± .00 | 5.61 ± .00 | 2.13 ± .02 | 0.10 ± .00 | 0.10 ± .00 |
| CH | 1.13 ± .00 | 1.17 ± .00 | 1.35 ± .01 | 0.69 ± .00 | 1.50 ± .00 | 0.57 ± .01 | 1.07 ± .03 | 0.67 ± .00 | 1.16 ± .00 | **0.54** ± .00 | **0.54** ± .01 |
| **Missing 50%** | | | | | | | | | | | |
| AI | 2.32 ± .02 | 2.30 ± .04 | 2.93 ± .01 | 2.17 ± .04 | 2.45 ± .06 | **1.59** ± .04 | 2.30 ± .04 | 2.17 ± .05 | 2.34 ± .04 | 1.68 ± .01 | 1.66 ± .02 |
| BL | 1.15 ± .02 | 1.14 ± .06 | 1.12 ± .02 | 0.86 ± .04 | 1.11 ± .02 | 0.82 ± .02 | 1.31 ± .07 | 1.48 ± .09 | 1.98 ± .01 | **0.78** ± .00 | **0.78** ± .00 |
| WW | 0.76 ± .00 | 0.85 ± .00 | 0.91 ± .00 | 0.67 ± .00 | 0.96 ± .00 | 0.69 ± .01 | 0.99 ± .01 | 0.68 ± .01 | 0.76 ± .00 | **0.61** ± .00 | **0.61** ± .00 |
| IO | 2.02 ± .02 | 1.33 ± .03 | 1.40 ± .01 | 1.58 ± .01 | 1.54 ± .01 | 1.37 ± .02 | 2.44 ± .38 | 5.46 ± .00 | 5.10 ± .04 | 1.18 ± .00 | **1.11** ± .01 |
| BR | 1.07 ± .00 | 0.77 ± .00 | 0.62 ± .00 | 0.47 ± .01 | 0.48 ± .00 | **0.41** ± .00 | 0.87 ± .07 | 1.43 ± .00 | 1.89 ± .01 | 0.44 ± .00 | 0.44 ± .00 |
| IR | 2.19 ± .06 | 1.75 ± .07 | 2.19 ± .03 | 1.44 ± .12 | 2.13 ± .03 | 1.34 ± .12 | 1.49 ± .08 | 3.22 ± .00 | 4.78 ± .08 | 1.05 ± .01 | 1.05 ± .00 |
| DI | 1.78 ± .02 | 1.96 ± .02 | 1.86 ± .00 | 1.55 ± .03 | 1.69 ± .00 | **1.38** ± .01 | 1.58 ± .01 | 2.74 ± .00 | 5.01 ± .02 | 1.44 ± .01 | 1.44 ± .00 |
| PR | 0.91 ± .00 | 0.64 ± .00 | 1.04 ± .00 | 0.42 ± .00 | 1.56 ± .00 | 0.33 ± .00 | 1.00 ± .02 | 0.39 ± .01 | 0.94 ± .00 | **0.30** ± .00 | **0.30** ± .00 |
| SP | 0.23 ± .00 | 0.21 ± .00 | 0.32 ± .00 | 0.23 ± .00 | 0.17 ± .00 | 0.19 ± .00 | 0.21 ± .00 | 0.21 ± .00 | **0.16** ± .00 | 0.17 ± .01 | 0.16 ± .00 |
| LE | 1.28 ± .00 | 1.39 ± .00 | 1.37 ± .00 | 1.07 ± .00 | 2.35 ± .00 | 0.81 ± .00 | 2.20 ± .04 | 1.17 ± .01 | 1.32 ± .02 | **0.69** ± .00 | **0.69** ± .01 |
| AB | 2.52 ± .01 | 2.56 ± .03 | 2.90 ± .05 | 2.39 ± .03 | 2.64 ± .01 | 2.15 ± .20 | 2.83 ± .07 | 3.63 ± .00 | 2.65 ± .01 | 2.19 ± .00 | **2.00** ± .03 |
| A4 | 1.07 ± .00 | 1.20 ± .02 | 1.28 ± .01 | 0.96 ± .01 | 1.95 ± .00 | **0.86** ± .02 | 1.38 ± .04 | 1.00 ± .00 | 1.12 ± .01 | 0.87 ± .00 | **0.85** ± .00 |
| CM | 2.36 ± .01 | 2.43 ± .06 | 2.65 ± .01 | 2.33 ± .03 | 3.50 ± .04 | 2.18 ± .02 | 2.50 ± .16 | 2.24 ± .03 | 2.40 ± .02 | 2.01 ± .02 | **1.94** ± .02 |
| GE | 2.52 ± .01 | 2.56 ± .03 | 2.90 ± .05 | 2.38 ± .03 | 2.64 ± .01 | 2.15 ± .20 | 2.83 ± .07 | 3.63 ± .00 | 2.65 ± .01 | 2.17 ± .01 | **2.03** ± .01 |
| ST | 1.80 ± .00 | 1.41 ± .01 | 1.54 ± .01 | 1.19 ± .01 | 1.42 ± .00 | 0.89 ± .01 | 1.67 ± .01 | 1.22 ± .05 | 1.70 ± .01 | 0.61 ± .01 | **0.56** ± .01 |
| LI | 1.82 ± .00 | 0.35 ± .00 | 0.39 ± .00 | 0.26 ± .01 | 0.27 ± .00 | 0.17 ± .00 | 0.97 ± .12 | 5.30 ± .00 | 2.08 ± .00 | 0.14 ± .00 | 0.13 ± .01 |
| CH | 1.13 ± .00 | 1.21 ± .00 | 1.45 ± .00 | 0.84 ± .00 | 1.74 ± .00 | 0.74 ± .00 | 1.33 ± .07 | 0.83 ± .00 | 1.16 ± .00 | **0.70** ± .00 | **0.70** ± .00 |
| **Missing 70%** | | | | | | | | | | | |
| AI | 2.32 ± .01 | 2.45 ± .05 | 3.02 ± .03 | 2.26 ± .02 | 2.67 ± .03 | 2.26 ± .03 | 2.36 ± .03 | 2.31 ± .02 | 2.37 ± .01 | **2.06** ± .01 | 2.07 ± .01 |
| BL | 1.15 ± .01 | 1.18 ± .02 | 1.27 ± .02 | 0.99 ± .04 | 1.30 ± .01 | 0.99 ± .03 | 1.33 ± .05 | 1.47 ± .01 | 1.97 ± .01 | 0.93 ± .00 | **0.92** ± .00 |
| WW | 0.76 ± .00 | 0.92 ± .01 | 1.02 ± .00 | 0.72 ± .00 | 1.31 ± .00 | 0.82 ± .02 | 1.49 ± .11 | 0.72 ± .05 | 0.79 ± .01 | **0.68** ± .00 | **0.68** ± .00 |
| IO | 2.02 ± .01 | 2.21 ± .04 | 1.90 ± .05 | 1.83 ± .04 | 1.94 ± .04 | 1.54 ± .01 | 3.47 ± .47 | 5.31 ± .00 | 5.12 ± .02 | 1.29 ± .00 | **1.27** ± .01 |
| BR | 1.07 ± .00 | 1.07 ± .01 | 0.73 ± .00 | 0.79 ± .07 | 0.65 ± .01 | **0.52** ± .00 | 1.10 ± .07 | 1.45 ± .00 | 1.88 ± .01 | 0.55 ± .00 | **0.55** ± .00 |
| IR | 2.22 ± .03 | 2.48 ± .14 | 2.87 ± .03 | 1.79 ± .07 | 2.91 ± .05 | 1.57 ± .01 | 1.66 ± .07 | 3.20 ± .00 | 4.71 ± .10 | 1.34 ± .00 | **1.33** ± .02 |
| DI | 1.81 ± .00 | 2.05 ± .03 | 2.05 ± .02 | 1.69 ± .03 | 2.06 ± .01 | 1.65 ± .01 | 2.44 ± .20 | 2.76 ± .00 | 5.02 ± .03 | 1.61 ± .01 | **1.58** ± .01 |
| PR | 0.91 ± .00 | 0.75 ± .00 | 1.16 ± .00 | 0.61 ± .00 | 1.94 ± .00 | 0.47 ± .00 | 1.48 ± .12 | 0.54 ± .01 | 0.94 ± .00 | 0.43 ± .01 | **0.42** ± .00 |
| SP | 0.23 ± .00 | 0.24 ± .00 | 0.35 ± .01 | 0.23 ± .01 | 0.17 ± .00 | 0.20 ± .00 | 0.17 ± .01 | 0.21 ± .00 | **0.16** ± .00 | 0.17 ± .00 | 0.16 ± .00 |
| LE | 1.28 ± .00 | 1.47 ± .00 | 1.58 ± .00 | 1.19 ± .01 | 2.95 ± .00 | 1.12 ± .02 | 1.61 ± .07 | 1.31 ± .02 | 1.32 ± .01 | **0.94** ± .01 | **0.94** ± .01 |
| AB | 2.53 ± .01 | 2.61 ± .03 | 3.31 ± .07 | 2.51 ± .06 | 2.96 ± .01 | 2.31 ± .29 | 3.21 ± .15 | 3.63 ± .00 | 2.69 ± .01 | 2.36 ± .02 | **2.24** ± .01 |
| A4 | 1.06 ± .00 | 1.27 ± .04 | 1.53 ± .01 | 1.02 ± .01 | 2.30 ± .00 | 0.99 ± .01 | 2.31 ± .03 | 1.04 ± .02 | 1.22 ± .01 | 0.96 ± .00 | **0.92** ± .00 |
| CM | 2.38 ± .00 | 2.68 ± .06 | 3.11 ± .04 | 2.37 ± .04 | 4.24 ± .01 | **1.99** ± .03 | 4.06 ± .07 | 2.40 ± .01 | 2.40 ± .01 | 2.17 ± .00 | 2.13 ± .02 |
| GE | 2.53 ± .01 | 2.61 ± .03 | 3.31 ± .07 | 2.51 ± .06 | 2.96 ± .01 | 2.31 ± .29 | 3.21 ± .15 | 3.63 ± .00 | 2.69 ± .01 | 2.35 ± .01 | **2.22** ± .02 |
| ST | 1.80 ± .00 | 1.75 ± .01 | 1.85 ± .02 | 1.56 ± .08 | 1.88 ± .00 | 1.14 ± .00 | 2.42 ± .16 | 1.50 ± .00 | 1.70 ± .01 | 0.93 ± .01 | **0.88** ± .01 |
| LI | 1.82 ± .00 | 0.74 ± .00 | 0.50 ± .01 | 0.71 ± .01 | 0.55 ± .01 | 0.33 ± .01 | 3.84 ± .27 | 4.90 ± .00 | 2.05 ± .01 | 0.22 ± .01 | **0.21** ± .01 |
| CH | 1.13 ± .00 | 1.27 ± .00 | 1.61 ± .00 | 0.99 ± .00 | 2.02 ± .00 | 0.94 ± .01 | 2.27 ± .02 | 0.98 ± .02 | 1.16 ± .00 | 0.89 ± .00 | **0.88** ± .00 |

Table 20: MAE scores under the **MAR** setting across different levels of missingness on the extra 17 datasets. Please refer to Table 9 for dataset names.

| Dataset | Mean | Knn | Svd | Mice | Spectral | HI | Gain | Miracle | Miwae | Grape | M³-Impute |
|---|---|---|---|---|---|---|---|---|---|---|---|
| **Missing 10%** | | | | | | | | | | | |
| AI | 2.40 ± .19 | 1.87 ± .24 | 2.37 ± .34 | 1.85 ± .18 | 2.01 ± .28 | **0.56 ± .16** | 2.17 ± .09 | 1.22 ± .05 | 2.57 ± .05 | 0.75 ± .02 | 0.68 ± .02 |
| BL | 1.12 ± .43 | 0.55 ± .30 | 1.05 ± .35 | 0.63 ± .52 | 0.79 ± .31 | 0.59 ± .18 | 1.07 ± .10 | 1.14 ± .00 | 1.99 ± .10 | 0.21 ± .00 | **0.20 ± .00** |
| WW | 0.74 ± .12 | **0.36 ± .01** | 0.86 ± .04 | 0.45 ± .06 | 0.71 ± .11 | 0.39 ± .01 | 0.60 ± .02 | 0.44 ± .01 | 0.69 ± .01 | 0.38 ± .00 | 0.37 ± .00 |
| IO | 2.11 ± .02 | 1.19 ± .04 | 1.41 ± .01 | 1.44 ± .14 | 1.41 ± .06 | 1.32 ± .03 | 1.36 ± .02 | 5.41 ± .00 | 6.09 ± .09 | 1.19 ± .03 | 1.04 ± .02 |
| BR | 1.18 ± .03 | 0.53 ± .04 | 0.59 ± .02 | **0.29 ± .03** | 0.34 ± .01 | 0.30 ± .01 | 0.50 ± .03 | 1.40 ± .00 | 2.18 ± .03 | 0.34 ± .01 | 0.34 ± .00 |
| IR | 2.33 ± .56 | 0.97 ± .10 | 1.31 ± .18 | 0.88 ± .14 | 0.99 ± .29 | 0.89 ± .06 | 1.23 ± .14 | 2.19 ± .00 | 3.41 ± .36 | 0.87 ± .01 | **0.86 ± .01** |
| DI | 1.78 ± .38 | 1.11 ± .03 | 1.84 ± .42 | 1.26 ± .40 | 1.69 ± .41 | 0.93 ± .02 | 1.16 ± .03 | 2.41 ± .00 | 4.20 ± .09 | **0.87 ± .01** | 0.87 ± .02 |
| PR | 0.92 ± .09 | 0.23 ± .04 | 1.47 ± .15 | 0.30 ± .08 | 0.59 ± .32 | 0.19 ± .01 | 0.58 ± .13 | **0.17 ± .02** | 0.99 ± .05 | 0.20 ± .01 | 0.19 ± .00 |
| SP | 0.23 ± .01 | **0.14 ± .02** | 0.50 ± .15 | 0.20 ± .01 | 0.18 ± .00 | 0.15 ± .01 | 0.22 ± .00 | 0.18 ± .00 | 0.15 ± .01 | 0.14 ± .00 | 0.14 ± .00 |
| LE | 1.35 ± .08 | **0.31 ± .03** | 1.36 ± .09 | 0.88 ± .12 | 1.43 ± .20 | 0.45 ± .07 | 0.96 ± .06 | 0.66 ± .03 | 1.56 ± .01 | 0.44 ± .01 | 0.40 ± .00 |
| AB | 2.63 ± .28 | 2.21 ± .24 | 2.37 ± .24 | 1.93 ± .13 | 2.45 ± .08 | 2.16 ± .29 | 2.06 ± .06 | 3.09 ± .00 | 2.54 ± .04 | 1.91 ± .02 | **1.79 ± .02** |
| A4 | 1.23 ± .20 | 0.61 ± .23 | 1.38 ± .15 | 0.93 ± .39 | 1.12 ± .34 | **0.48 ± .42** | 1.27 ± .22 | 0.40 ± .02 | 1.53 ± .06 | 1.14 ± .01 | 1.10 ± .01 |
| CM | 2.17 ± .07 | 2.33 ± .11 | 2.37 ± .30 | 2.19 ± .14 | 2.97 ± .09 | 1.64 ± .19 | 2.04 ± .15 | 1.84 ± .00 | 1.97 ± .07 | 1.63 ± .01 | 1.56 ± .01 |
| GE | 2.63 ± .28 | 2.21 ± .24 | 2.37 ± .24 | 1.93 ± .13 | 2.45 ± .08 | 2.16 ± .29 | 2.06 ± .06 | 3.09 ± .00 | 2.54 ± .04 | 1.92 ± .00 | **1.83 ± .01** |
| ST | 1.76 ± .18 | 0.52 ± .07 | 1.30 ± .16 | 0.64 ± .10 | 1.07 ± .17 | 0.25 ± .06 | 0.76 ± .03 | 0.60 ± .03 | 1.48 ± .02 | 0.28 ± .01 | **0.21 ± .00** |
| LI | 1.88 ± .02 | 0.26 ± .00 | 0.36 ± .01 | **0.04 ± .00** | 0.15 ± .00 | 0.10 ± .00 | 0.34 ± .01 | 5.88 ± .00 | 2.26 ± .04 | 0.11 ± .00 | 0.10 ± .01 |
| CH | 1.18 ± .41 | 0.53 ± .13 | 1.53 ± .26 | 0.67 ± .12 | 1.21 ± .12 | 0.40 ± .09 | 1.11 ± .23 | 0.37 ± .01 | 1.02 ± .04 | **0.27 ± .00** | **0.27 ± .00** |
| **Missing 30%** | | | | | | | | | | | |
| AI | 2.33 ± .14 | 1.59 ± .70 | 2.99 ± .83 | 2.16 ± .28 | 2.01 ± .60 | **1.21 ± .21** | 2.29 ± .09 | 2.23 ± .00 | 2.56 ± .01 | 1.57 ± .02 | 1.54 ± .02 |
| BL | 0.91 ± .02 | 0.90 ± .25 | 0.91 ± .07 | 1.00 ± .40 | 1.22 ± .36 | 0.88 ± .33 | 1.01 ± .15 | 1.14 ± .00 | 2.03 ± .03 | 0.29 ± .01 | **0.28 ± .00** |
| WW | 0.87 ± .01 | 0.53 ± .02 | 0.78 ± .05 | 0.63 ± .04 | 0.99 ± .23 | 0.57 ± .08 | 0.65 ± .11 | 0.60 ± .05 | 0.69 ± .01 | 0.48 ± .00 | 0.48 ± .00 |
| IO | 2.02 ± .08 | 1.09 ± .03 | 1.40 ± .08 | 1.43 ± .08 | 1.50 ± .02 | 1.30 ± .03 | 1.71 ± .10 | 5.28 ± .00 | 6.10 ± .04 | 1.17 ± .03 | 1.07 ± .01 |
| BR | 1.13 ± .03 | 0.53 ± .03 | 0.61 ± .03 | 0.32 ± .07 | 0.46 ± .04 | 0.34 ± .02 | 0.69 ± .05 | 1.33 ± .00 | 2.17 ± .03 | 0.39 ± .00 | 0.37 ± .01 |
| IR | 1.99 ± .25 | 0.91 ± .08 | 1.85 ± .42 | 0.85 ± .09 | 1.62 ± .13 | 1.05 ± .11 | 1.25 ± .04 | 2.38 ± .00 | 3.46 ± .13 | 0.86 ± .02 | **0.82 ± .03** |
| DI | 1.74 ± .33 | 1.43 ± .23 | 2.09 ± .02 | 1.33 ± .23 | 1.32 ± .20 | 1.46 ± .10 | 1.34 ± .04 | 2.56 ± .00 | 4.26 ± .06 | 1.12 ± .01 | **1.07 ± .00** |
| PR | 0.91 ± .03 | 0.22 ± .02 | 1.32 ± .20 | 0.35 ± .06 | 1.77 ± .26 | 0.23 ± .01 | 0.88 ± .02 | 0.27 ± .04 | 0.97 ± .01 | 0.20 ± .01 | **0.20 ± .00** |
| SP | 0.23 ± .01 | 0.16 ± .01 | 0.40 ± .06 | 0.24 ± .01 | 0.19 ± .02 | 0.18 ± .00 | 0.20 ± .00 | 0.20 ± .00 | 0.15 ± .00 | 0.15 ± .01 | **0.14 ± .00** |
| LE | 1.36 ± .02 | **0.42 ± .05** | 1.56 ± .29 | 1.07 ± .02 | 2.42 ± .21 | 0.58 ± .11 | 1.28 ± .13 | 0.97 ± .04 | 1.56 ± .01 | 0.57 ± .01 | 0.54 ± .00 |
| AB | 2.53 ± .08 | 2.43 ± .19 | 2.65 ± .16 | 2.26 ± .27 | 2.72 ± .35 | 2.21 ± .10 | 2.21 ± .08 | 3.06 ± .00 | 2.59 ± .02 | 2.08 ± .03 | **1.91 ± .02** |
| A4 | 1.35 ± .19 | 0.79 ± .04 | 1.12 ± .19 | 0.94 ± .20 | 1.72 ± .76 | 0.88 ± .27 | **0.77 ± .12** | 0.90 ± .05 | 1.44 ± .02 | 1.11 ± .00 | 1.08 ± .00 |
| CM | 2.41 ± .21 | 2.11 ± .09 | 2.60 ± .10 | 2.02 ± .21 | 3.71 ± .74 | 2.22 ± .03 | 2.18 ± .11 | 2.21 ± .00 | 2.08 ± .02 | 1.83 ± .01 | **1.74 ± .01** |
| GE | 2.53 ± .08 | 2.43 ± .19 | 2.65 ± .16 | 2.26 ± .27 | 2.72 ± .35 | 2.25 ± .10 | 2.21 ± .08 | 3.07 ± .00 | 2.59 ± .02 | 2.06 ± .02 | **1.91 ± .02** |
| ST | 1.79 ± .17 | 0.77 ± .10 | 1.58 ± .42 | 0.91 ± .12 | 1.32 ± .35 | 0.48 ± .03 | 1.25 ± .09 | 0.84 ± .02 | 1.48 ± .01 | 0.45 ± .01 | **0.42 ± .02** |
| LI | 1.91 ± .01 | 0.48 ± .01 | 0.40 ± .00 | **0.07 ± .00** | 0.30 ± .01 | 0.15 ± .00 | 0.96 ± .01 | 5.07 ± .00 | 2.24 ± .03 | 0.15 ± .01 | 0.11 ± .01 |
| CH | 1.04 ± .23 | 0.83 ± .07 | 1.65 ± .20 | 0.71 ± .25 | 2.61 ± .24 | 0.47 ± .10 | 1.29 ± .15 | 0.42 ± .03 | 1.02 ± .03 | **0.32 ± .00** | **0.32 ± .00** |
| **Missing 50%** | | | | | | | | | | | |
| AI | 2.47 ± .17 | 2.13 ± .79 | 3.42 ± .64 | 2.18 ± .14 | 2.52 ± .34 | 2.03 ± .11 | 2.18 ± .16 | 2.16 ± .00 | 2.27 ± .01 | **1.86 ± .01** | **1.86 ± .01** |
| BL | 1.22 ± .20 | 1.26 ± .18 | 0.78 ± .05 | 0.80 ± .12 | 1.11 ± .45 | 0.96 ± .08 | 1.24 ± .05 | 1.39 ± .00 | 2.31 ± .04 | **0.73 ± .00** | **0.73 ± .00** |
| WW | 0.77 ± .04 | 0.68 ± .09 | 0.88 ± .08 | 0.69 ± .00 | 1.08 ± .15 | 0.75 ± .03 | 1.40 ± .05 | 0.74 ± .00 | 0.82 ± .02 | 0.65 ± .00 | **0.65 ± .00** |
| IO | 2.04 ± .03 | **1.06 ± .01** | 1.55 ± .09 | 1.70 ± .10 | 1.79 ± .08 | 1.43 ± .04 | 2.54 ± .10 | 5.43 ± .00 | 5.31 ± .03 | 1.20 ± .00 | 1.13 ± .02 |
| BR | 1.13 ± .03 | 0.62 ± .01 | 0.65 ± .04 | **0.42 ± .02** | 0.61 ± .08 | 0.43 ± .02 | 0.93 ± .11 | 1.37 ± .00 | 2.09 ± .01 | 0.42 ± .00 | 0.42 ± .00 |
| IR | 2.50 ± .23 | 1.62 ± .54 | 2.34 ± .53 | 1.53 ± .28 | 2.62 ± .27 | 1.36 ± .35 | 1.74 ± .29 | 3.07 ± .00 | 4.03 ± .30 | **0.84 ± .01** | **0.84 ± .01** |
| DI | 1.82 ± .25 | 1.76 ± .22 | 1.82 ± .17 | 1.53 ± .24 | 1.84 ± .21 | **1.37 ± .16** | 2.20 ± .05 | 2.88 ± .00 | 5.79 ± .02 | 1.65 ± .01 | 1.60 ± .02 |
| PR | 1.03 ± .06 | 0.38 ± .03 | 1.11 ± .07 | 0.41 ± .04 | 1.72 ± .31 | 0.27 ± .02 | 1.09 ± .18 | 0.38 ± .00 | 0.95 ± .01 | **0.23 ± .00** | **0.23 ± .00** |
| SP | 0.23 ± .01 | 0.17 ± .00 | 0.37 ± .02 | 0.22 ± .01 | 0.16 ± .02 | 0.19 ± .01 | **0.16 ± .00** | 0.20 ± .00 | 0.16 ± .00 | 0.16 ± .01 | 0.15 ± .00 |
| LE | 1.28 ± .07 | 0.86 ± .03 | 1.61 ± .06 | 1.05 ± .04 | 2.61 ± .11 | 0.92 ± .03 | 1.90 ± .10 | 1.27 ± .00 | 1.36 ± .02 | 0.71 ± .02 | **0.69 ± .01** |
| AB | 2.39 ± .16 | 2.44 ± .26 | 3.19 ± .30 | 2.51 ± .04 | 2.83 ± .15 | 2.27 ± .14 | 2.66 ± .23 | 3.65 ± .00 | 2.58 ± .04 | 2.19 ± .02 | **2.04 ± .01** |
| A4 | 1.17 ± .11 | 1.11 ± .37 | 1.29 ± .02 | **0.87 ± .29** | 1.90 ± .34 | 0.87 ± .04 | 1.50 ± .16 | 1.10 ± .04 | 1.17 ± .01 | 0.87 ± .01 | 0.84 ± .00 |
| CM | 2.37 ± .06 | 2.28 ± .18 | 2.67 ± .11 | 2.38 ± .05 | 4.27 ± .47 | **1.86 ± .08** | 3.39 ± .01 | 2.54 ± .00 | 2.43 ± .01 | 2.17 ± .01 | 2.04 ± .02 |
| GE | 2.39 ± .16 | 2.44 ± .26 | 3.19 ± .30 | 2.51 ± .04 | 2.83 ± .15 | 2.27 ± .14 | 2.66 ± .23 | 3.65 ± .00 | 2.58 ± .04 | 2.23 ± .03 | **2.05 ± .03** |
| ST | 1.72 ± .11 | 0.83 ± .03 | 1.89 ± .23 | 1.18 ± .14 | 1.59 ± .21 | 0.83 ± .05 | 2.04 ± .24 | 1.13 ± .00 | 1.52 ± .02 | 0.64 ± .01 | **0.57 ± .01** |
| LI | 1.89 ± .02 | 0.58 ± .00 | 0.44 ± .01 | **0.16 ± .04** | 0.43 ± .01 | 0.21 ± .01 | 2.11 ± .08 | 4.25 ± .00 | 2.31 ± .04 | 0.22 ± .02 | 0.13 ± .01 |
| CH | 1.09 ± .18 | 0.88 ± .12 | 1.51 ± .02 | 1.03 ± .13 | 1.78 ± .42 | 0.81 ± .08 | 1.74 ± .19 | 0.78 ± .00 | 1.06 ± .02 | 0.67 ± .00 | **0.66 ± .00** |
| **Missing 70%** | | | | | | | | | | | |
| AI | 2.50 ± .19 | 2.31 ± .55 | 3.21 ± .96 | 2.24 ± .16 | 2.47 ± .22 | 2.06 ± .11 | 2.32 ± .22 | 2.25 ± .00 | 2.26 ± .01 | 1.96 ± .00 | **1.95 ± .00** |
| BL | 1.27 ± .31 | 1.13 ± .17 | 1.00 ± .29 | 0.93 ± .15 | 0.94 ± .51 | 1.01 ± .08 | 1.09 ± .08 | 1.40 ± .00 | 2.33 ± .03 | **0.80 ± .01** | **0.80 ± .01** |
| WW | 0.81 ± .02 | 0.88 ± .02 | 0.92 ± .01 | **0.71 ± .03** | 1.35 ± .21 | 0.85 ± .02 | 1.56 ± .06 | 0.81 ± .00 | 0.84 ± .00 | 0.72 ± .00 | 0.71 ± .00 |
| IO | 2.08 ± .03 | 1.48 ± .08 | 2.03 ± .10 | 1.90 ± .05 | 2.37 ± .13 | 1.60 ± .01 | 3.52 ± .32 | 5.44 ± .00 | 5.28 ± .05 | **1.29 ± .00** | **1.29 ± .01** |
| BR | 1.13 ± .04 | 0.68 ± .01 | 0.73 ± .02 | 0.74 ± .07 | 0.73 ± .03 | 0.54 ± .02 | 1.13 ± .07 | 1.40 ± .00 | 2.00 ± .01 | 0.55 ± .00 | 0.54 ± .01 |
| IR | 2.48 ± .13 | 1.56 ± .13 | 1.89 ± .46 | 1.89 ± .60 | 2.94 ± .52 | 1.39 ± .20 | 2.86 ± .24 | 3.07 ± .00 | 4.03 ± .29 | 0.85 ± .01 | **0.85 ± .01** |
| DI | 1.91 ± .04 | 1.96 ± .07 | 2.01 ± .09 | 1.77 ± .05 | 2.26 ± .08 | **1.52 ± .10** | 2.90 ± .13 | 2.96 ± .00 | 5.03 ± .05 | 1.68 ± .00 | 1.68 ± .01 |
| PR | 1.05 ± .05 | 0.53 ± .02 | 1.18 ± .07 | 0.57 ± .17 | 2.02 ± .18 | 0.44 ± .00 | 1.32 ± .27 | 0.60 ± .08 | 1.01 ± .01 | **0.40 ± .00** | **0.40 ± .00** |
| SP | 0.23 ± .01 | 0.20 ± .01 | 0.38 ± .01 | 0.24 ± .00 | **0.17 ± .00** | 0.21 ± .00 | 0.16 ± .01 | 0.22 ± .00 | 0.16 ± .01 | 0.16 ± .01 | 0.16 ± .01 |
| LE | 1.30 ± .08 | 1.30 ± .14 | 1.72 ± .13 | 1.25 ± .04 | 3.16 ± .10 | **1.21 ± .03** | 1.67 ± .14 | 1.30 ± .02 | 1.32 ± .05 | 0.98 ± .01 | **0.98 ± .01** |
| AB | 2.59 ± .09 | 2.47 ± .06 | 3.53 ± .23 | 2.63 ± .05 | 3.12 ± .17 | 2.93 ± .16 | 3.08 ± .07 | 3.54 ± .00 | 2.68 ± .01 | 2.46 ± .01 | **2.33 ± .03** |
| A4 | 1.14 ± .02 | 1.13 ± .11 | 1.17 ± .02 | 0.98 ± .11 | 2.45 ± .29 | **0.86 ± .04** | 1.67 ± .18 | 1.16 ± .00 | 1.21 ± .09 | 0.94 ± .01 | 0.90 ± .01 |
| CM | 2.43 ± .05 | 2.46 ± .20 | 3.13 ± .10 | 2.53 ± .06 | 4.32 ± .41 | | 4.24 ± .29 | 2.58 ± .00 | 2.40 ± .02 | 2.10 ± .00 | **2.05 ± .03** |
| GE | 2.59 ± .09 | 2.47 ± .06 | 3.53 ± .23 | 2.63 ± .05 | 3.12 ± .17 | 2.93 ± .16 | 3.08 ± .07 | 3.54 ± .00 | 2.68 ± .01 | 2.43 ± .00 | **2.34 ± .03** |
| ST | 1.75 ± .12 | 1.07 ± .05 | 1.88 ± .08 | 1.61 ± .10 | 2.10 ± .12 | 1.12 ± .04 | 2.36 ± .06 | 1.31 ± .00 | 1.55 ± .03 | 0.82 ± .01 | **0.76 ± .00** |
| LI | 1.90 ± .01 | 0.71 ± .03 | 0.57 ± .01 | 0.55 ± .02 | 0.89 ± .02 | 0.36 ± .01 | 3.18 ± .09 | 4.92 ± .00 | 2.10 ± .01 | 0.28 ± .01 | **0.23 ± .01** |
| CH | 1.12 ± .06 | 1.23 ± .06 | 1.75 ± .08 | 1.01 ± .08 | 1.74 ± .08 | 0.91 ± .06 | 2.18 ± .07 | 1.02 ± .01 | 1.21 ± .00 | 0.88 ± .00 | **0.87 ± .00** |

Table 21: MAE scores under the **MNAR** setting across different levels of missingness on the extra 17 datasets. Please refer to Table 9 for dataset names.

| Dataset | Mean | Knn | Svd | Mice | Spectral | HI | Gain | Miracle | Miwae | Grape | M³-Impute |
|---|---|---|---|---|---|---|---|---|---|---|---|
| **Missing 10%** | | | | | | | | | | | |
| AI | 2.46 ± .12 | 1.93 ± .10 | 2.81 ± .19 | 1.83 ± .13 | 2.39 ± .33 | **0.68** ± .03 | 2.24 ± .03 | 1.70 ± .04 | 2.38 ± .03 | 0.75 ± .01 | 0.69 ± .02 |
| BL | 1.05 ± .05 | 1.07 ± .13 | 1.01 ± .18 | 0.62 ± .12 | 1.01 ± .12 | 0.56 ± .09 | 1.22 ± .05 | 1.34 ± .00 | 1.98 ± .06 | **0.39** ± .00 | **0.39** ± .00 |
| WW | 0.78 ± .04 | **0.41** ± .01 | 0.88 ± .04 | 0.51 ± .02 | 0.67 ± .05 | 0.45 ± .01 | 0.68 ± .00 | 0.49 ± .00 | 0.77 ± .00 | 0.44 ± .00 | 0.44 ± .00 |
| IO | 2.07 ± .05 | 1.10 ± .02 | 1.29 ± .04 | 1.41 ± .09 | 1.38 ± .01 | 1.24 ± .04 | 1.46 ± .04 | 5.45 ± .00 | 5.49 ± .12 | 1.17 ± .01 | **1.08** ± .00 |
| BR | 1.10 ± .02 | 0.50 ± .01 | 0.58 ± .01 | **0.26** ± .01 | 0.33 ± .01 | 0.28 ± .01 | 0.46 ± .01 | 1.34 ± .00 | 1.92 ± .01 | 0.31 ± .00 | 0.31 ± .00 |
| IR | 2.20 ± .12 | 1.21 ± .16 | 1.67 ± .16 | 0.89 ± .04 | 1.32 ± .03 | 0.91 ± .02 | 1.30 ± .05 | 2.79 ± .00 | 4.18 ± .09 | 0.75 ± .01 | **0.73** ± .01 |
| DI | 1.79 ± .13 | 1.51 ± .05 | 1.72 ± .12 | 1.13 ± .16 | 1.44 ± .09 | **1.02** ± .07 | 1.39 ± .02 | 2.74 ± .00 | 4.59 ± .09 | 1.20 ± .03 | 1.18 ± .01 |
| PR | 0.91 ± .03 | 0.32 ± .02 | 1.03 ± .09 | 0.27 ± .01 | 0.80 ± .08 | 0.21 ± .02 | 0.57 ± .02 | 0.25 ± .01 | 0.99 ± .04 | **0.19** ± .00 | **0.19** ± .00 |
| SP | 0.23 ± .01 | 0.15 ± .00 | 0.37 ± .02 | 0.20 ± .01 | 0.16 ± .00 | 0.18 ± .00 | 0.21 ± .01 | 0.18 ± .00 | 0.16 ± .00 | 0.16 ± .00 | **0.15** ± .00 |
| LE | 1.31 ± .02 | **0.34** ± .01 | 1.32 ± .02 | 0.90 ± .03 | 1.29 ± .04 | 0.50 ± .01 | 1.07 ± .01 | 0.74 ± .00 | 1.40 ± .01 | 0.44 ± .00 | 0.43 ± .00 |
| AB | 2.53 ± .12 | 2.06 ± .12 | 2.63 ± .11 | 2.04 ± .12 | 2.31 ± .03 | 1.95 ± .09 | 2.23 ± .05 | 3.43 ± .00 | 2.56 ± .03 | 1.83 ± .00 | **1.66** ± .02 |
| A4 | 1.07 ± .08 | 1.05 ± .07 | 1.21 ± .09 | 0.84 ± .15 | 1.23 ± .15 | **0.65** ± .11 | 2.49 ± .04 | 0.79 ± .02 | 1.24 ± .02 | 0.86 ± .00 | 0.84 ± .00 |
| CM | 2.45 ± .08 | 2.19 ± .12 | 2.49 ± .06 | 2.06 ± .04 | 2.68 ± .10 | **1.74** ± .11 | 2.49 ± .04 | 2.07 ± .04 | 2.38 ± .00 | 1.91 ± .00 | 1.83 ± .00 |
| GE | 2.53 ± .12 | 2.06 ± .12 | 2.63 ± .11 | 2.04 ± .12 | 2.31 ± .03 | 1.95 ± .09 | 2.23 ± .05 | 3.43 ± .00 | 2.56 ± .03 | 1.84 ± .02 | **1.67** ± .00 |
| ST | 1.71 ± .05 | 0.54 ± .03 | 1.30 ± .02 | 0.71 ± .02 | 0.90 ± .00 | 0.60 ± .03 | 0.93 ± .04 | 0.71 ± .02 | 1.60 ± .01 | 0.35 ± .01 | **0.30** ± .01 |
| LI | 1.84 ± .01 | 0.20 ± .00 | 0.37 ± .00 | **0.05** ± .00 | 0.14 ± .00 | 0.06 ± .00 | 0.38 ± .00 | 5.62 ± .00 | 2.16 ± .02 | 0.11 ± .01 | 0.09 ± .00 |
| CH | 1.07 ± .04 | 0.87 ± .06 | 1.24 ± .05 | 0.60 ± .05 | 1.42 ± .04 | 0.45 ± .04 | 1.22 ± .16 | 0.49 ± .01 | 1.09 ± .01 | **0.37** ± .00 | **0.37** ± .00 |
| **Missing 30%** | | | | | | | | | | | |
| AI | 2.36 ± .11 | 2.11 ± .27 | 2.98 ± .52 | 2.07 ± .14 | 2.64 ± .18 | **1.23** ± .04 | 2.21 ± .05 | 1.72 ± .00 | 2.47 ± .03 | 1.46 ± .03 | 1.46 ± .01 |
| BL | 0.98 ± .05 | 1.04 ± .12 | 0.98 ± .09 | 0.76 ± .17 | 1.40 ± .18 | 0.82 ± .18 | 1.09 ± .06 | 1.24 ± .00 | 1.99 ± .04 | 0.42 ± .00 | **0.41** ± .00 |
| WW | 0.82 ± .01 | 0.60 ± .02 | 0.82 ± .04 | 0.62 ± .02 | 0.88 ± .13 | 0.58 ± .05 | 0.69 ± .01 | 0.59 ± .01 | 0.72 ± .00 | 0.49 ± .00 | 0.49 ± .00 |
| IO | 2.04 ± .06 | 1.12 ± .03 | 1.36 ± .07 | 1.44 ± .07 | 1.46 ± .02 | 1.28 ± .02 | 1.55 ± .03 | 5.39 ± .00 | 5.66 ± .02 | 1.15 ± .01 | **1.06** ± .02 |
| BR | 1.11 ± .02 | 0.55 ± .02 | 0.60 ± .03 | **0.33** ± .02 | 0.41 ± .03 | 0.36 ± .03 | 0.62 ± .02 | 1.35 ± .00 | 2.05 ± .00 | 0.38 ± .00 | 0.36 ± .01 |
| IR | 2.06 ± .09 | 1.53 ± .52 | 1.66 ± .20 | 0.99 ± .11 | 1.35 ± .11 | 1.07 ± .07 | 1.26 ± .04 | 2.67 ± .00 | 3.98 ± .32 | 0.89 ± .02 | **0.87** ± .00 |
| DI | 1.77 ± .20 | 1.60 ± .17 | 1.93 ± .02 | 1.27 ± .16 | 1.51 ± .13 | 1.30 ± .19 | 1.43 ± .06 | 2.60 ± .00 | 4.62 ± .08 | 1.21 ± .01 | **1.19** ± .00 |
| PR | 0.93 ± .02 | 0.55 ± .05 | 1.17 ± .08 | 0.38 ± .05 | 1.66 ± .12 | 0.23 ± .01 | 0.60 ± .19 | 0.28 ± .06 | 0.96 ± .01 | **0.21** ± .01 | **0.21** ± .00 |
| SP | 0.23 ± .01 | 0.18 ± .01 | 0.40 ± .07 | 0.26 ± .02 | 0.16 ± .01 | 0.17 ± .00 | 0.18 ± .00 | 0.19 ± .00 | 0.15 ± .00 | 0.15 ± .00 | **0.14** ± .00 |
| LE | 1.32 ± .08 | 0.59 ± .05 | 1.56 ± .29 | 0.94 ± .01 | 1.95 ± .14 | 0.58 ± .07 | 1.30 ± .08 | 0.96 ± .01 | 1.48 ± .01 | 0.54 ± .01 | **0.52** ± .00 |
| AB | 2.49 ± .14 | 2.46 ± .11 | 2.67 ± .07 | 2.25 ± .06 | 2.55 ± .10 | 2.36 ± .13 | 2.37 ± .03 | 3.27 ± .00 | 2.60 ± .04 | 2.03 ± .01 | **1.89** ± .03 |
| A4 | 1.01 ± .26 | 1.27 ± .13 | 0.99 ± .18 | 0.88 ± .15 | 1.98 ± .03 | 0.84 ± .22 | **0.81** ± .07 | 1.12 ± .06 | 1.35 ± .03 | 1.04 ± .00 | 1.02 ± .01 |
| CM | 2.37 ± .10 | 2.23 ± .03 | 2.54 ± .07 | 2.36 ± .13 | 2.95 ± .05 | 1.91 ± .07 | 2.13 ± .03 | 2.06 ± .00 | 2.15 ± .05 | 1.82 ± .01 | **1.71** ± .02 |
| GE | 2.49 ± .14 | 2.46 ± .11 | 2.67 ± .07 | 2.25 ± .06 | 2.55 ± .10 | 2.36 ± .13 | 2.37 ± .03 | 3.27 ± .00 | 2.60 ± .04 | 2.04 ± .02 | **1.88** ± .02 |
| ST | 1.75 ± .08 | 0.79 ± .13 | 1.58 ± .14 | 0.91 ± .11 | 1.24 ± .11 | 0.69 ± .04 | 1.27 ± .09 | 0.89 ± .04 | 1.56 ± .01 | 0.42 ± .00 | **0.39** ± .01 |
| LI | 1.85 ± .03 | 0.37 ± .00 | 0.38 ± .01 | **0.09** ± .00 | 0.22 ± .00 | 0.12 ± .01 | 0.62 ± .03 | 5.29 ± .00 | 2.21 ± .02 | 0.12 ± .01 | 0.12 ± .01 |
| CH | 0.95 ± .10 | 1.19 ± .14 | 1.52 ± .15 | 0.62 ± .14 | 1.69 ± .42 | **0.53** ± .06 | 1.09 ± .17 | 0.54 ± .02 | 1.06 ± .01 | 0.42 ± .00 | 0.42 ± .00 |
| **Missing 50%** | | | | | | | | | | | |
| AI | 2.39 ± .11 | 2.37 ± .68 | 3.04 ± .23 | 2.35 ± .09 | 2.82 ± .08 | 2.02 ± .02 | 2.13 ± .11 | 2.17 ± .00 | 2.27 ± .03 | **1.86** ± .00 | **1.86** ± .00 |
| BL | 1.22 ± .23 | 1.24 ± .09 | 1.06 ± .31 | 0.87 ± .13 | 1.19 ± .41 | 0.94 ± .07 | 1.24 ± .04 | 1.39 ± .00 | 2.26 ± .03 | **0.77** ± .00 | **0.77** ± .00 |
| WW | 0.70 ± .07 | 0.78 ± .04 | 0.96 ± .08 | 0.72 ± .04 | 1.13 ± .10 | 0.74 ± .03 | 1.24 ± .04 | 0.73 ± .00 | 0.83 ± .02 | **0.65** ± .00 | **0.65** ± .00 |
| IO | 2.06 ± .03 | 1.20 ± .02 | 1.54 ± .02 | 1.61 ± .07 | 1.74 ± .04 | 1.43 ± .07 | 2.59 ± .15 | 5.43 ± .00 | 5.27 ± .01 | 1.18 ± .00 | **1.15** ± .00 |
| BR | 1.12 ± .03 | 0.66 ± .01 | 0.67 ± .03 | 0.46 ± .02 | 0.55 ± .03 | 0.43 ± .01 | 0.87 ± .09 | 1.40 ± .00 | 2.05 ± .02 | 0.44 ± .01 | **0.43** ± .01 |
| IR | 2.42 ± .09 | 1.58 ± .10 | 2.19 ± .26 | **1.30** ± .17 | 2.27 ± .05 | 1.38 ± .32 | 1.61 ± .23 | 3.10 ± .00 | 4.16 ± .30 | 0.96 ± .01 | 0.96 ± .01 |
| DI | 1.99 ± .04 | 1.70 ± .03 | 1.92 ± .07 | 1.69 ± .07 | 1.80 ± .19 | **1.39** ± .15 | 1.99 ± .02 | 2.86 ± .00 | 5.64 ± .00 | 1.60 ± .02 | 1.60 ± .00 |
| PR | 1.04 ± .10 | 0.66 ± .09 | 1.06 ± .03 | 0.45 ± .07 | 1.75 ± .17 | 0.30 ± .01 | 1.26 ± .15 | 0.40 ± .00 | 0.95 ± .01 | **0.25** ± .00 | **0.25** ± .00 |
| SP | 0.23 ± .00 | 0.19 ± .00 | 0.38 ± .03 | 0.23 ± .00 | 0.16 ± .02 | 0.19 ± .00 | 0.19 ± .01 | 0.20 ± .00 | 0.16 ± .00 | 0.17 ± .00 | **0.15** ± .00 |
| LE | 1.32 ± .02 | 1.26 ± .11 | 1.61 ± .04 | 1.08 ± .03 | 2.62 ± .10 | 0.92 ± .03 | 1.62 ± .15 | 1.24 ± .00 | 1.34 ± .02 | 0.72 ± .00 | **0.70** ± .01 |
| AB | 2.41 ± .06 | 2.47 ± .04 | 3.14 ± .23 | 2.34 ± .11 | 2.65 ± .19 | 2.37 ± .13 | 2.92 ± .22 | 3.63 ± .00 | 2.60 ± .01 | 2.23 ± .03 | **2.03** ± .02 |
| A4 | 1.17 ± .09 | 1.25 ± .22 | 1.37 ± .26 | 0.85 ± .11 | 2.44 ± .19 | 0.75 ± .01 | 1.46 ± .10 | 1.17 ± .00 | 1.17 ± .02 | **0.88** ± .00 | **0.84** ± .00 |
| CM | 2.40 ± .07 | 2.34 ± .07 | 2.82 ± .22 | 2.30 ± .10 | 4.13 ± .24 | **1.90** ± .09 | 3.29 ± .27 | 2.51 ± .00 | 2.43 ± .02 | 2.10 ± .00 | 1.98 ± .03 |
| GE | 2.41 ± .06 | 2.47 ± .04 | 3.14 ± .23 | 2.34 ± .11 | 2.65 ± .19 | 2.37 ± .13 | 2.92 ± .22 | 3.63 ± .00 | 2.60 ± .01 | 2.19 ± .03 | **2.01** ± .05 |
| ST | 1.76 ± .09 | 0.96 ± .06 | 1.71 ± .10 | 1.27 ± .09 | 1.67 ± .07 | 0.85 ± .01 | 1.88 ± .07 | 1.17 ± .00 | 1.56 ± .02 | 0.63 ± .01 | **0.59** ± .01 |
| LI | 1.87 ± .01 | 0.52 ± .01 | 0.44 ± .01 | **0.18** ± .01 | 0.40 ± .01 | 0.06 ± .00 | 1.68 ± .08 | 4.39 ± .00 | 2.26 ± .02 | 0.20 ± .02 | 0.14 ± .01 |
| CH | 1.12 ± .12 | 1.11 ± .11 | 1.66 ± .11 | 0.88 ± .13 | 1.82 ± .23 | 0.80 ± .07 | 1.94 ± .26 | 0.78 ± .00 | 1.08 ± .02 | 0.68 ± .00 | **0.67** ± .00 |
| **Missing 70%** | | | | | | | | | | | |
| AI | 2.36 ± .18 | 2.33 ± .43 | 3.26 ± .67 | 2.33 ± .09 | 2.84 ± .13 | 2.11 ± .12 | 2.25 ± .14 | 2.29 ± .00 | 2.28 ± .01 | 2.01 ± .02 | **1.99** ± .02 |
| BL | 1.22 ± .19 | 1.05 ± .23 | 1.25 ± .29 | 0.93 ± .13 | 1.36 ± .19 | 1.01 ± .08 | 1.16 ± .05 | 1.40 ± .00 | 2.29 ± .03 | **0.82** ± .01 | **0.82** ± .00 |
| WW | 0.79 ± .01 | 0.90 ± .02 | 0.96 ± .00 | 0.75 ± .01 | 1.47 ± .12 | 0.87 ± .02 | 1.53 ± .11 | 0.80 ± .00 | 0.84 ± .00 | 0.72 ± .00 | **0.71** ± .00 |
| IO | 2.07 ± .01 | 1.67 ± .08 | 2.11 ± .01 | 1.92 ± .05 | 2.29 ± .07 | 1.57 ± .02 | 4.02 ± .46 | 5.43 ± .00 | 5.27 ± .04 | 1.31 ± .01 | **1.28** ± .02 |
| BR | 1.15 ± .03 | 0.93 ± .04 | 0.79 ± .06 | 0.83 ± .09 | 0.75 ± .05 | **0.55** ± .01 | 1.16 ± .06 | 1.40 ± .00 | 1.97 ± .01 | 0.55 ± .00 | 0.55 ± .01 |
| IR | 2.46 ± .10 | 1.57 ± .18 | 2.21 ± .30 | 2.58 ± .99 | 2.78 ± .52 | 1.28 ± .16 | 1.69 ± .13 | 3.07 ± .00 | 4.13 ± .29 | 1.01 ± .01 | 1.01 ± .01 |
| DI | 1.86 ± .03 | 1.96 ± .13 | 2.06 ± .05 | **1.67** ± .17 | 2.45 ± .05 | 1.58 ± .10 | 2.41 ± .19 | 2.95 ± .00 | 5.01 ± .03 | 1.67 ± .02 | 1.66 ± .02 |
| PR | 1.04 ± .05 | 0.63 ± .02 | 1.19 ± .06 | 0.59 ± .17 | 1.17 ± .09 | 0.46 ± .01 | 1.44 ± .14 | 0.62 ± .05 | 1.00 ± .01 | **0.41** ± .00 | **0.41** ± .00 |
| SP | 0.23 ± .00 | 0.23 ± .00 | 0.38 ± .00 | 0.23 ± .00 | 0.17 ± .00 | 0.21 ± .00 | 0.19 ± .03 | 0.22 ± .00 | 0.16 ± .00 | 0.17 ± .01 | **0.16** ± .00 |
| LE | 1.33 ± .04 | 1.48 ± .03 | 1.75 ± .11 | 1.25 ± .01 | 3.05 ± .01 | 1.21 ± .02 | 1.63 ± .08 | 1.24 ± .04 | 1.34 ± .02 | **0.98** ± .00 | **0.98** ± .00 |
| AB | 2.61 ± .07 | 2.51 ± .07 | 3.52 ± .17 | 2.56 ± .04 | 2.99 ± .10 | 2.36 ± .35 | 3.13 ± .06 | 3.55 ± .00 | 2.66 ± .02 | 2.47 ± .02 | **2.34** ± .03 |
| A4 | 1.13 ± .01 | 1.20 ± .10 | 1.53 ± .30 | 1.04 ± .13 | 2.55 ± .09 | 1.09 ± .13 | 1.60 ± .19 | 1.18 ± .00 | 1.22 ± .09 | 0.95 ± .00 | **0.91** ± .01 |
| CM | 2.38 ± .13 | 2.52 ± .07 | 3.16 ± .17 | 2.38 ± .11 | 4.62 ± .24 | 2.15 ± .06 | 4.00 ± .23 | 2.55 ± .00 | 2.40 ± .02 | 2.12 ± .04 | **2.02** ± .02 |
| GE | 2.61 ± .07 | 2.51 ± .07 | 3.52 ± .17 | 2.56 ± .04 | 2.99 ± .10 | 2.36 ± .35 | 3.13 ± .06 | 3.55 ± .00 | 2.66 ± .02 | 2.46 ± .02 | **2.33** ± .05 |
| ST | 1.74 ± .11 | 1.45 ± .03 | 2.05 ± .12 | 1.55 ± .12 | 2.16 ± .05 | 1.25 ± .09 | 2.35 ± .15 | 1.31 ± .00 | 1.58 ± .03 | 0.83 ± .01 | **0.78** ± .01 |
| LI | 1.89 ± .01 | 0.71 ± .01 | 0.60 ± .05 | 0.62 ± .03 | 0.92 ± .01 | 0.37 ± .01 | 3.67 ± .29 | 4.93 ± .00 | 2.09 ± .01 | 0.27 ± .02 | **0.24** ± .00 |
| CH | 1.22 ± .05 | 1.27 ± .07 | 1.81 ± .06 | 0.99 ± .06 | 1.95 ± .11 | 0.91 ± .03 | 2.02 ± .10 | 1.02 ± .03 | 1.20 ± .00 | 0.89 ± .00 | **0.88** ± .00 |

1350
1351
1352
1353
1354
1355
1356
1357
1358
1359
1360
1361
1362
1363
1364
1365
1366
1367
1368
1369
1370
1371
1372
1373
1374
1375
1376
1377
1378
1379
1380
1381
1382
1383
1384
1385
1386
1387
1388
1389
1390
1391
1392
1393
1394
1395
1396
1397
1398
1399
1400
1401
1402
1403

# B  ADDITIONAL EXPERIMENTS DURING THE REBUTTAL PERIOD

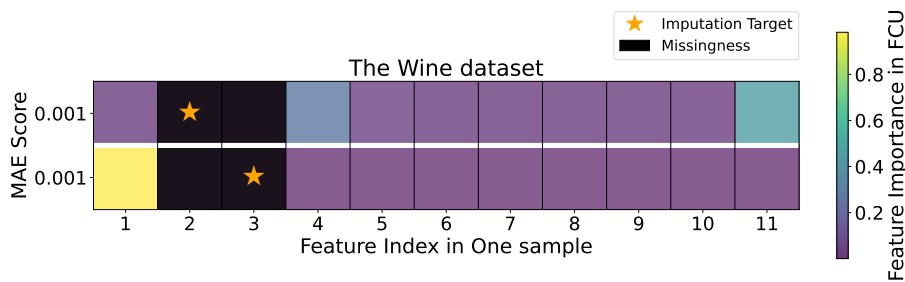

Figure 6: MAE scores and feature importance scores calculated from FCU when imputing two missing entries in a sample. FCU dynamically adjusts the importance of observed features within the sample when imputing different feature values.

In this section, we provide further clarification on the working mechanism and effectiveness of the Feature Correlation Unit (FCU) and the Sample Correlation Unit (SCU) in M³-Impute, supported by illustrative experiment results. In Section B.1, we show that the FCU dynamically adjusts the feature correlations by considering both the imputation targets and the missingness patterns. In Section B.2, we highlight that the SCU introduces an improved measure of sample correlations which results in enhanced imputation performance compared to the one with the standard cosine similarity measure.

## B.1  FCU ADJUSTS FEATURE IMPORTANCE WHEN IMPUTING DIFFERENT TARGETS

The FCU in M³-Impute is designed to fully exploit feature correlations when imputing a missing entry by adjusting the importance of observed features. In the following experiments, we demonstrate how the FCU adaptively weighs observable features based on both the imputation targets and the missingness patterns across different samples. Specifically, we first show that when imputing different missing values within a sample, the FCU adjusts the importance of observable features according to the specific imputation targets. In addition, we demonstrate that when imputing missing values in the same feature across different samples, the FCU adaptively learns the importance of observable features by considering the unique missingness patterns of each sample.

Suppose we are to impute the missing value of feature $f$ for a sample $s$. We compute the feature-wise similarities between $f$ and the observed features in $s$, which are given by $(\mathbf{H}_F^\top \mathbf{h}_f) \odot \mathbf{m}'_s$, as in Equation 3 . We then take the absolute values of the feature-wise similarities and normalize the values such that their sum becomes one. We here use the resulting normalized scores to represent the importance of each observed feature in $s$ when imputing the feature $f$.

In Figure 6, we show a heatmap on how the FCU dynamically adjusts the importance of observed features when imputing different missing feature values in a sample from the Wine dataset. The sample contains two missing values in Feature 2 and Feature 3. Missingness is represented by black cells, while the target feature value to impute at each step is marked with a star. Each row of the heatmap represents the importance scores of observable features when imputing the corresponding target feature value on the same row, which are derived from FCU. In the first row, when imputing the second feature (*volatile_acidity*), the FCU identifies the fourth feature (*residual_sugar*) and the last feature (*alcohol*) as the most important features, while assigning low importance to other observable features such as fixed_acidty (feature 1) and density (feature 8). This makes sense as fixed acids such as tartaric acid, malic acid, and citric acid are non-volatile and do not easily evaporate. In addition, density is primarily determined by sugar and alcohol levels, with no direct correlation to volatile acidity. Consequently, these two features offer limited information in imputing the missing feature volatile_acidity. In the second row, when imputing the third feature (*citric_acid*), the FCU considers the first feature (*fixed_acidity*) to be more important than the other features such as free_sulfur_dioxide (Feature 6) and sulphates (Feature 10). This again demonstrates that the FCU can effectively adjust the importance of observed features within the same sample when imputing different missing values.

In addition, we fix the target imputation feature and analyze how the FCU adjusts feature importance based on the missingness patterns across different samples. As shown in Figure 7, each row represents

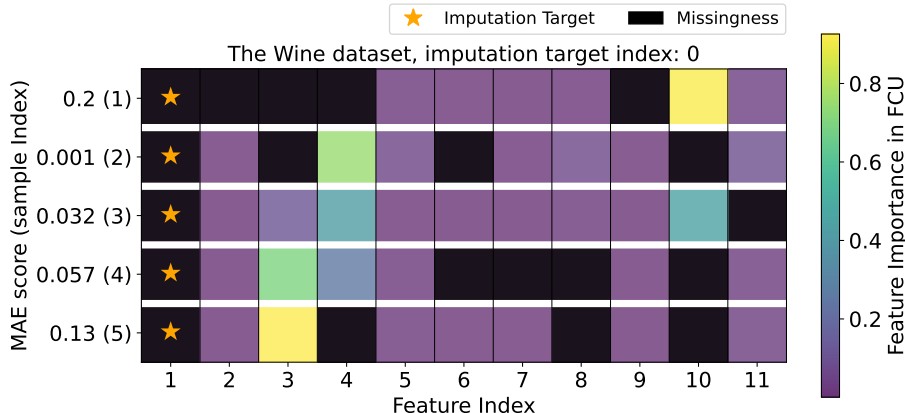

Figure 7: MAE scores and feature importance for the same imputation target across different samples. The number in parentheses indicates the sample index. FCU dynamically adjusts the importance of observed features under varying missingness patterns.

a different sample, and the missingness patterns vary in different samples. For the first sample (first row), when imputing the missing value in the first feature (*fixed_acidity*), the FCU assigns the highest importance to the tenth feature (*sulphates*). In the second sample, where the tenth feature is missing, the FCU considers the fourth feature (*residual_sugar*) as the most important feature for imputation. For the third sample, both the fourth and tenth features are observed. Thus, they have similar importance, followed by the third feature (*citric_acid*). In the fourth sample, when the tenth feature is missing, the FCU considers the third and fourth features as the most important features. Finally, for the fifth sample, where both the fourth and tenth features are missing, the FCU assigns the highest importance to the third feature (*citric_acid*) when imputing the missing value in the first feature (*fixed_acidity*). This example clearly demonstrates that the FCU is effective in dynamically assigning feature importance under varying missingness patterns across different samples.

### B.2 SCU IS SUPERIOR AT CAPTURING SAMPLE CORRELATIONS

A common approach to compute sample correlations would be to use the dot product or cosine similarity between their embedding vectors. This approach, however, fails to take into account the missingness pattern in a sample. It also does not consider the fact that different observed features are of different importance to the target feature to impute when it comes to measuring the similarities. To address these limitations, in SCU, we introduce the mutual sample masking mechanism and integrate it with the FCU to jointly consider the commonly observed features between samples and their importance in imputing different targets. While the details of SCU are explained in Section 3.4, the key computation of pairwise similarity in SCU is given by Equation 6.

We first introduce a variant of M$^3$-Impute, denoted as "**SCU (cos sim)**", in which we change Equation 6 with cosine similarity while keeping the remaining computations in M$^3$-Impute untouched. The updated equation is now defined as:

$$\text{sim}(s, p \mid f) = \cos(\mathbf{h_s}, \ \mathbf{h_p}), \tag{13}$$

where $s$ and $p$ represent two samples, and $\mathbf{h_s}$ and $\mathbf{h_p}$ denote their respective sample embeddings. For imputation, SCU (cos sim) computes the similarities between the target sample and a subset of peers, as done in SCU. We below evaluate the impact of the peer similarities by SCU and SCU (cos sim) on the imputation performance, where the averaged absolute similarity score is used per imputation.

In Figure 8, we summarize the MAE scores of M$^3$-Impute with our SCU and SCU (cos sim) under different peer similarity scores when imputing all the missing values in the Wine dataset. The MAE scores of both methods decrease as peer similarity increases. This is intuitive, as similar peers can provide more relevant information for imputation, resulting in lower imputation errors. However, M$^3$-Impute consistently achieves lower mean MAE scores, demonstrating the effectiveness of our novel similarity measure.

To better quantify the improvements achieved by the enhanced similarity measure in SCU, we present comprehensive results in terms of MAE scores for M$^3$-Impute and SCU (cos sim) in Table 22.

Table 22: MAE scores of SCU (cos sim) and SCU in M³-Impute under the MCAR setting with 30% missingness.

|  | Steel | Naval | Breast | Airfoil | Concrete | Ionosphere | Yacht | Abalone |
|---|---|---|---|---|---|---|---|---|
| SCU (cos sim) | 0.46 | 0.07 | 0.38 | 1.16 | 0.75 | 1.06 | 1.37 | 1.88 |
| M³-Impute | 0.39 | 0.06 | 0.36 | 1.09 | 0.71 | 1.01 | 1.33 | 1.84 |
| Improv. Ratio | 14.47% | 10.04% | 6.25% | 6.03% | 5.71% | 4.72% | 2.92% | 2.13% |
|  | Wine | CMC | Diabetes | Housing | Energy | Power | German | Kin8nm |
| SCU (cos sim) | 0.61 | 1.84 | 1.31 | 0.60 | 1.32 | 1.00 | 1.88 | 2.50 |
| M³-Impute | 0.60 | 1.81 | 1.29 | 0.59 | 1.31 | 0.99 | 1.87 | 2.50 |
| Improv. Ratio | 2.12% | 1.63% | 1.53% | 1.01% | 0.76% | 0.70% | 0.53% | 0.00% |

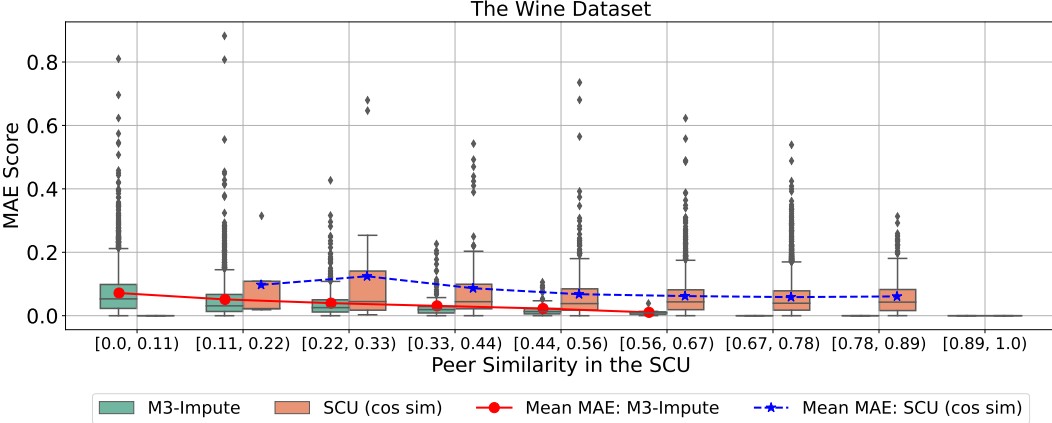

Figure 8: MAE scores and peer similarity scores measured by two methods. Data are missing under the MCAR setting with 30% missingness.

The results show that M³-Impute consistently outperforms SCU (cos sim), with up to 14.47% improvement on the steel dataset. These findings highlight the effectiveness of the proposed SCU, which incorporates the mutual sample masking mechanism and the explicit consideration of feature importance through FCU to measure peer similarity more effectively.

