# OpenReview forum: "M$^3$-Impute: Mask-guided Representation Learning for Missing Value Imputation"
_ICLR.cc/2025/Conference — Submitted to ICLR 2025_

### Official Review · Reviewer_pTfx · 2024-10-27

**Soundness:** 2
**Presentation:** 3
**Contribution:** 2
**Rating:** 3
**Confidence:** 5

**Summary:**

Missing values challenge data analysis and machine learning, requiring effective imputation to enhance dataset quality. The authors claim that current methods often ignore missingness information and correlations among features and samples, reducing accuracy. M3-Impute addresses this by modeling data as a bipartite graph and using a graph neural network that incorporates missingness in initialization. Its feature and sample correlation units (FCU and SCU) capture essential correlations for improved imputation. Testing on 25 benchmark datasets under three missingness settings shows M3-Impute’s superior performance, achieving top MAE scores in most cases.

**Strengths:**

Originality: M3-Impute introduces a novel approach to missing value imputation by leveraging a mask-guided representation learning method. Unlike conventional methods, it incorporates both sample-feature relationships and missingness information in embedding initialization. The development of the feature correlation unit (FCU) and sample correlation unit (SCU) further distinguishes this method, providing unique mechanisms to explicitly model feature- and sample-wise correlations.

Quality: The proposed methodology demonstrates a well-structured integration of graph representation learning with innovative imputation techniques. M3-Impute’s use of bipartite graph modeling, along with the FCU and SCU units, reflects a robust design that balances theory and practical application, enhancing the overall quality and rigor of the imputation process.

Clarity: The paper clearly outlines each component of M3-Impute, detailing the step-by-step process of embedding initialization, feature- and sample-wise correlation modeling, and final imputation integration. This level of clarity provides readers with a clear understanding of the method’s architecture and functionality.

Significance: By directly addressing gaps in current imputation methods, M3-Impute holds significant potential to advance the field. Its effective incorporation of missingness information and explicit correlation modeling not only improves imputation accuracy but also sets a foundation for more sophisticated data analysis in complex datasets, impacting fields reliant on high-quality data imputation.

**Weaknesses:**

There are several major limitations.

First, the discussion of missing data mechanisms is incomplete. The literature on different missing data mechanisms—such as Missing Completely at Random (MCAR), Missing at Random (MAR), and Missing Not at Random (MNAR)—is extensive. Please refer to:

Ibrahim, J. G., Chen, M. H., Lipsitz, S. R., & Herring, A. H. (2005). Missing-data methods for generalized linear models: A comparative review. Journal of the American Statistical Association, 100(469), 332-346.
Kim, J.K. and Shao, J. (2021). Statistical methods for handling incomplete data. Chapman and Hall/CRC.
This paper, however, is limited to addressing only MCAR.

Second, the graph-based missing data method proposed by You et al. (2020) faces significant issues. In particular, it does not scale well to large datasets, as its efficiency depends heavily on both sample size and the number of variables.

Third, some concepts similar to the feature correlation unit (FCU) and sample correlation unit (SCU) have been previously explored in the following references:

Zhong, J., Gui, N., & Ye, W. (2023). "Data Imputation with Iterative Graph Reconstruction." Proceedings of the AAAI Conference on Artificial Intelligence, 37(9), 11399-11407.
 Jarrett, Daniel, et al. "HyperImpute: Generalized Iterative Imputation with Automatic Model Selection" ICML 2022.
 Gupta, Shubham, et al. "GRAFENNE: learning on graphs with heterogeneous and dynamic feature sets." International Conference on Machine Learning. PMLR, 2023.
 Qiao, Lishan, et al. "Estimating functional brain networks by incorporating a modularity prior." Neuroimage 141 (2016): 399-407.
 Um, Daeho, et al. "Confidence-Based Feature Imputation for Graphs with Partially Known Features." The Eleventh International Conference on Learning Representations.

**Questions:**

While Graph Neural Networks (GNNs) are valuable for capturing the intrinsic structure of feature and sample correlations, the M3 method, along with similar approaches, falls short in modeling the underlying missing data mechanisms. Additionally, most machine learning tasks increasingly emphasize causal inference to ensure robust and interpretable outcomes. However, your current method overlooks causal inference considerations, which could limit its application to settings where understanding causal relationships is critical.

---

> ### Author Response · Authors · 2024-11-16
>
> We appreciate the reviewer for the comments. Below we respond to the main concerns raised.
>
> **Q(1) First, the discussion of missing data mechanisms is incomplete.**
>
> - Thanks for the comment. We would like to point out that we indeed included the evaluation of M$^3$-Impute under all MCAR, MAR, and MNAR settings in the submitted manuscript. Please refer to Table 1, Table 16, and Table 19 for MCAR results, Table 7, Table 17, and Table 20 for MAR results, and Table 8, Table 18, and Table 21 for MNAR results.
>
>
> **Q(2) Second, the graph-based missing data method proposed by You et al. (2020) faces significant issues. In particular, it does not scale well to large datasets, as its efficiency depends heavily on both sample size and the number of variables.**
>
> - Thanks for the comment. With all due respect, we would like to clarify that graph-based imputation methods are highly scalable to large datasets. Regarding the training and inference efficiency on large datasets, this can be effectively managed using the graph mini-batch training/inference and the distributed computing techniques [1, 2]. Additionally, the number of model parameters remain **independent** of both the number of features and the number of samples in the dataset, making graph-based imputation methods efficient in terms of model size as well. In particular, we also demonstrated that M$^3$-Impute runs efficiently for inference on large datasets, such as Protein, Spam, Letter, and California-Housing, with a running time of less than one second to impute all the missing values in each dataset with 30\% missingness when using a GPU, as shown in Table 10 of the submitted manuscript. We will soon also share the training time during this discussion period.
>
>
> **Q(3) Third, some concepts similar to the feature correlation unit (FCU) and sample correlation unit (SCU) have been previously explored in the following references [3-7]**
>
> - Thanks for the comment. Although there may be some relevance between our work and the previous studies [3,4,5,8] in capturing feature and sample correlations, our FCU and SCU are still substantially different from them. The primary advantage of FCU lies in its ability to dynamically adjust the importance of observable features when imputing different missing features for a sample (see Equation (5)). Similarly, the key benefit of SCU is its capability to adjust both the information passed from peer samples and the importance of observable features when imputing missing values across different samples. This ability to adaptively learn the importance of peers and observable features improves imputation accuracy, as shown in Table 2 for the ablation study in the submitted manuscript. None of the existing graph-based imputation methods and other previous studies [3-7] incorporate such dynamic capabilities introduced by FCU and SCU. Furthermore, we below differentiate our work from the previous studies [3-7] in more detail.
>
> 1. IGRM [3]: IGRM enables direct message passing between samples through extra edge creation. However, it incurred out-of-memory (OOM) errors for some of the datasets that we used with their official implementation. Thus we did not include it in our submitted manuscript.
>
> 2. HyperImpute [4]: Hyperimpute is an ensemble method that demonstrates highly competitive imputation accuracy and even outperforms the recent imputation methods [9, 10]. It selects a method (e.g., kNN, LR, NN) from its model pool to impute missing values. While these methods can model feature and sample correlations, they do not incorporate the dynamic adjustments introduced by the FCU and SCU. For example, the kNN method imputes missing values by identifying the k nearest neighbors and averaging their values. However, it fails to consider the importance or the weights of different features when calculating their similarities. In contrast, the FCU and SCU can adaptively learn the importance of each feature for imputing different values in different samples. We would also like to point out that M$^3$-Impute consistently outperforms HyperImpute for most of the 25 datasets. Please kindly refer to Table 1, Table 7, and Table 8 (as well as other tables and figures) of the submitted manuscript for more details.
>
> 3. GRAFENNE [5]: There are several key distinctions between M$^3$-Impute and GRAFENNE: (1) GRAFENNE does not impute missing values; it targets at completing specific downstream tasks only. (2) GRAFENNE requires labeled data while M$^3$-Impute does not, as M$^3$-Impute adopts a self-supervised learning approach. (3) M$^3$-Impute adaptively learns feature and sample correlations when imputing different values of different samples, whereas GRAFENNE cannot.
>
> 4. Lowrank [6]: It introduces prior information in estimating functional brain networks. There seems to be no connection between this paper and missing data imputation. We are not sure why it is mentioned here.

---

> > ### Author Response · Authors · 2024-11-16
> >
> > (Cont.)
> >
> > 5. PCFI [7]: There are several key distinctions between M$^3$-Impute and PCFI: (1) PCFI uses a homogeneous graph as input, where all nodes are sample nodes, and each node is associated with an $F$-dimensional feature vector that includes missingness. In contrast, M$^3$-Impute models the tabular data as a bipartite graph, where each row corresponds to a sample node, and each column corresponds to a feature node. Missing values are naturally captured by the absence of edges between corresponding sample nodes and feature nodes. (2) PCFI method only captures the "sample-sample" relationship. In contrast, the aggregation operation in M$^3$-Impute captures multiple relationships, such as "sample-feature," "feature-feature," and "sample-sample." (3) PCFI uses information from pre-defined k-hop neighbors for imputation, which may not be similar to the target samples. In contrast, the SCU in M$^3$-Impute adaptively learns the information from the entire dataset in imputing missing values across different samples.
> >
> >
> > **Q(4) While Graph Neural Networks (GNNs) are valuable for capturing the intrinsic structure of feature and sample correlations, the M3 method, along with similar approaches, falls short in modeling the underlying missing data mechanisms.**
> >
> > - Thanks for the comment. We would like to clarify that M$^3$-Impute improves over existing GNN-based imputation methods, and can potentially identify the underlying relationships between missing values and observed ones. More specifically, when data is missing under MAR and MNAR conditions, the missing values depend on the observed ones. The FCU in M$^3$-Impute is designed to explicitly capture the relationships between observed and missing features. By modeling these relationships directly, M$^3$-Impute can model the missing data patterns, leading to enhanced imputation accuracy. This capability may explain why the improvement of M$^3$-Impute over baselines is more substantial under the MAR and MNAR settings, compared to that in the MCAR setting, where missingness is introduced completely at random.
> >
> > **Q(5) Additionally, most machine learning tasks increasingly emphasize causal inference to ensure robust and interpretable outcomes. However, your current method overlooks causal inference considerations, which could limit its application to settings where understanding causal relationships is critical.**
> >
> > - Thanks for the comment. Causal relationships are, however, not always predictable in real-world datasets. In such cases, attempting to model these relationships may reduce imputation accuracy instead. For example, **some baseline methods**, such as MIRACLE, considered in our manuscript aims to incorporate causal relationships between samples and features for imputation. However, they do not perform well in most experiments. A similar trend is also observed in [4]. In contrast, M$^3$-Impute outperforms **all the baseline methods**, including the causal based ones, highlighting its effectiveness even without explicitly modeling causal relationships. We would like to point out that, in addition to the results shown in the main manuscript, our method consistently outperforms **those methods** on 17 additional datasets under 3 missingness patterns, as shown in Tables 19–21 in the Appendix.
> >
> > [1] Zheng, Da, et al. "DistDGL: Distributed graph neural network training for billion-scale graphs." 2020 IEEE/ACM 10th Workshop on Irregular Applications: Architectures and Algorithms (IA3). IEEE, 2020.
> >
> > [2] Huang, Xin, et al. "CATGNN: Cost-Efficient and Scalable Distributed Training for Graph Neural Networks." arXiv preprint arXiv:2404.02300 (2024).
> >
> > [3] Zhong, J, et al. "Data Imputation with Iterative Graph Reconstruction." Proceedings of the AAAI Conference on Artificial Intelligence.
> >
> > [4] Jarrett, Daniel, et al. "Hyperimpute: Generalized iterative imputation with automatic model selection." International Conference on Machine Learning. PMLR, 2022.
> >
> > [5] Gupta, Shubham, et al. "GRAFENNE: learning on graphs with heterogeneous and dynamic feature sets." International Conference on Machine Learning. PMLR, 2023
> >
> > [6] Qiao, Lishan, et al. "Estimating functional brain networks by incorporating a modularity prior." Neuroimage 141 (2016).
> >
> > [7] Um, Daeho, et al. "Confidence-Based Feature Imputation for Graphs with Partially Known Features." The Eleventh International Conference on Learning Representations.
> >
> > [8] You, Jiaxuan, et al. "Handling missing data with graph representation learning." Advances in Neural Information Processing Systems (2020).
> >
> > [9] Du, T, et al. "ReMasker: Imputing Tabular Data with Masked Autoencoding" ICLR 2024
> >
> > [10] Zhang, H, et al. "Unleashing the Potential of Diffusion Models for Incomplete Data Imputation" arXiv preprint (2024).

---

> > > ### Comment · Reviewer_pTfx · 2024-11-16
> > > **modeling the underlying missing data mechanisms.**
> > >
> > > MNAR is a highly challenging scenario, and it is unclear how your method can be extended to address such complexities. You need to thoroughly understand these challenges and propose viable solutions. As mentioned previously, you significantly underestimate these difficulties and overstate your contributions.

---

> > > > ### Author Response · Authors · 2024-11-21
> > > >
> > > > **Q(2) MNAR is a highly challenging scenario, and it is unclear how your method can be extended to address such complexities. You need to thoroughly understand these challenges and propose viable solutions. As mentioned previously, you significantly underestimate these difficulties and overstate your contributions.**
> > > >
> > > > - Thanks for the comment. We indeed understand the challenges. In fact, we carefully designed our method to tackle such challenges. In Table 8, Table 18, and Table 21 of the submitted manuscript, we show that the proposed M$^3$-Impute method consistently achieves lower imputation error than SOTA imputation methods under *MNAR missingness*. This superior performance is due to M$^3$-Impute's unique approach. Rather than assuming the data follows MCAR, MAR, or MNAR missingness patterns from the outset, we designed M$^3$-Impute to leverage the missingness information directly, enabling it to learn feature-wise and sample-wise correlations. Specifically, the FCU leverages the missingness information with a learnable soft mask to capture the feature-feature correlation between the missing features and observable features. The SCU integrates the results from FCU with another learnable soft mask to better capture the sample-feature and sample-sample correlations. To the best of our knowledge, this direct and effective encoding of missingness information using learnable masks in the training process has not been proposed in the literature. Since feature and sample correlations exist regardless of the cause of missingness, M$^3$-Impute is naturally adaptive and robust across all three missingness settings.
> > > >
> > > > - In addition to the extensive experiments with the 10 baselines in the manuscript, we have done new experiments for the comparison with not-MIWAE [1], an imputation framework specially designed for imputing missingness under MNAR settings. The results, presented in Table 1 below, show that our method outperforms not-MIWAE substantially, highlighting again the robustness of M$^3$-Impute across different missingness patterns.
> > > >
> > > >   &nbsp;
> > > >    ### Table 1: Imputation accuracy in MAE under MNAR setting with 30\% missingness
> > > >
> > > >    |            | Yacht | Wine | Concrete | Housing | Energy | Naval | Kin8nm | Power |
> > > >    |------------|-------|------|----------|---------|--------|-------|--------|-------|
> > > >    | not-MIWAE  | 3.08  | 1.43 | 2.14     | 1.80    | 3.87   | 2.27  | 2.50   | 2.46  |
> > > >    | M$^3$-Impute  | **1.15** | **0.60** | **0.68** | **0.54** | **1.09** | **0.08** | **2.46** | **1.00** |
> > > >
> > > > [1] Ipsen, N.B., et al. "not-MIWAE: Deep generative modelling with missing not at random data." International Conference on Learning Representation. 2021.

---

> > > > > ### Comment · Reviewer_pTfx · 2024-11-24
> > > > > **imputing missingness under MNAR settings.**
> > > > >
> > > > > Please provide the formulation for generating the MNAR data in your tables, as it has not been included. Additionally, you have not discussed why your method is capable of handling all kinds of MNAR in a rational and justifiable manner.

---

> > > > > > ### Author Response · Authors · 2024-11-25
> > > > > >
> > > > > > **Q(1) Please provide the formulation for generating the MNAR data in your tables, as it has not been included. Additionally, you have not discussed why your method is capable of handling all kinds of MNAR in a rational and justifiable manner.**
> > > > > >
> > > > > > Thanks for the comment.
> > > > > > 1. We have discussed the process to generate data under the setting of MNAR missingness in our response to your comment: "The discussion of missing data mechanisms is incomplete". This process is the same as the one used in [1,2,3,4]. To clarify further, the process has two parts. The first part uses the MAR mechanism, and the second part uses the MCAR mechanism. In the first part, in order to use the MAR mechanism, we need to identify a fraction of fully observable features and use them as an input to a logistic model, such that missingness information can be generated on the rest of the features. In the second part, MCAR is applied on the fully observable features to mask out feature values randomly such that a target missingness ratio can be reached. For instance, to generate 30\% missingness under the MNAR setting, in the MAR step, we randomly select half (50\%) of the features to remain fully observable and mask 40\% of the remaining half by using a logistic model that takes the fully observable features as an input. This results in $50\% \times 40\% = 20\%$ of the values being masked. We then mask 20\% additional feature values in the set of fully observable features (50\%) using the MCAR mechanism, to generate an additional $50\% \times 20\% = 10\%$ missing values, thereby leading to a total missingness ratio of $20\% + 10\% = 30\%$.
> > > > > >
> > > > > > 2. We would like to point out that in our response to your comment on "modeling the underlying missing data mechanisms," we have already provided an explanation of why M$^3$-Impute is well-suited for handling MNAR missingness settings. Additionally, we have demonstrated, through extensive experiments involving 10 baseline methods included in the manuscript, as well as not-MIWAE (a method specifically optimized for the MNAR setting), that M$^3$-Impute consistently outperforms these baselines under MNAR scenarios.
> > > > > >
> > > > > > If there is still a concern, could you kindly provide additional references that include a theoretical study on the formulation of the MNAR setting for missing value imputation so that we can compare our M$^3$-Impute with their methods further?
> > > > > >
> > > > > > [1] Jarrett, Daniel, et al. "Hyperimpute: Generalized iterative imputation with automatic model selection." International Conference on Machine Learning. PMLR, 2022.
> > > > > >
> > > > > > [2] You, Jiaxuan, et al. "Handling missing data with graph representation learning." Advances in Neural Information Processing Systems (2020).
> > > > > >
> > > > > > [3] Muzellec, Boris, et al. "Missing data imputation using optimal transport." International Conference on Machine Learning. PMLR, 2020.
> > > > > >
> > > > > > [4] Zhong, J, et al. "Data Imputation with Iterative Graph Reconstruction." Proceedings of the AAAI Conference on Artificial Intelligence

---

> > > > > > > ### Comment · Reviewer_pTfx · 2024-11-26
> > > > > > > **the setting of MNAR missingness**
> > > > > > >
> > > > > > > The main issue is that MNAR encompasses a broad range of missing data mechanisms, yet you rely on a single specific simulation setting to promote your methods. However, it is always possible to design simulation studies that favor certain results. This is why your arguments are not sensible.
> > > > > > >
> > > > > > > https://academic.oup.com/aje/article/193/7/1019/7612961
> > > > > > >
> > > > > > > https://academic.oup.com/biomet/article-abstract/111/4/1413/7633920
> > > > > > >
> > > > > > > https://cameronpatrick.com/post/2023/06/untangling-mar-mcar-mnar/?twit
> > > > > > >
> > > > > > > https://academic.oup.com/biostatistics/advance-article/doi/10.1093/biostatistics/kxae044/7902044
> > > > > > >
> > > > > > > https://proceedings.mlr.press/v89/tu19a/tu19a.pdf

---

> > > > > > > > ### Author Response · Authors · 2024-11-28
> > > > > > > >
> > > > > > > > **The main issue is that MNAR encompasses a broad range of missing data mechanisms, yet you rely on a single specific simulation setting to promote your methods. However, it is always possible to design simulation studies that favor certain results. This is why your arguments are not sensible.**
> > > > > > > >
> > > > > > > > Thanks for the comment.
> > > > > > > >
> > > > > > > > 1. We agree that "MNAR encompasses a broad range of missing data mechanisms." However, we would like to point out that it is impossible to consider all possible MNAR settings exhaustively. We would also like to emphasize that M$^3$-Impute is not designed to work for specific MNAR settings, but it is designed in a way that is agnostic to the underlying missingness patterns. In other words, M$^3$-Impute does not use any prior information on the missingness patterns but leverage where missing values appear and how they are related to observable features. Therefore, we have evaluated and compared its performance against SOTA methods under not only the MNAR setting but also the MAR and MCAR settings. As for the MNAR setting, we did not arbitrarily generate missing values that could favor M$^3$-Impute. Instead, we followed the standard approach to generate missing values under the MNAR setting, as used in the literature [1-4] for missing value imputation.
> > > > > > > >
> > > > > > > > 2. Again, we would like to emphasize that our focus is not on leveraging causal relationships for missing value imputation, but on how to effectively and systemically learn correlations between features and samples so as to go beyond the performance of SOTA methods for missing value imputation. We do admit that causal analysis is important for missing value imputation. However, it does not mean that it is the only way to tackle the problem of missing value imputation or advance the field.
> > > > > > > >
> > > > > > > > [1] Jarrett, Daniel, et al. "HyperImpute: Generalized iterative imputation with automatic model selection." International Conference on Machine Learning. PMLR, 2022.
> > > > > > > >
> > > > > > > > [2] You, Jiaxuan, et al. "Handling missing data with graph representation learning." Advances in Neural Information Processing Systems (2020).
> > > > > > > >
> > > > > > > > [3] Muzellec, Boris, et al. "Missing data imputation using optimal transport." International Conference on Machine Learning. PMLR, 2020.
> > > > > > > >
> > > > > > > > [4] Zhong, J, et al. "Data Imputation with Iterative Graph Reconstruction." Proceedings of the AAAI Conference on Artificial Intelligence

---

> > ### Comment · Reviewer_pTfx · 2024-11-16
> > **the discussion of missing data mechanisms is incomplete**
> >
> > **********Thanks for the comment. We would like to point out that we indeed included the evaluation of M
> > -Impute under all MCAR, MAR, and MNAR settings in the submitted manuscript. Please refer to Table 1, Table 16, and Table 19 for MCAR results, Table 7, Table 17, and Table 20 for MAR results, and Table 8, Table 18, and Table 21 for MNAR results.
> >
> > Res.   You need to clearly describe how the MCAR, MAR, and MNAR settings are generated in the paper. This is currently missing. As a result, your claim regarding MNAR is inaccurate.

---

> > ### Comment · Reviewer_pTfx · 2024-11-16
> > **Third, some concepts similar to the feature correlation unit (FCU) and sample correlation unit (SCU) have been previously explored in the following references [3-7]**
> >
> > Both FCU and SCU are straightforward concepts, and your contributions are overstated compared with the existing references.

---

> > > ### Author Response · Authors · 2024-11-21
> > >
> > > **Q(3) Both FCU and SCU are straightforward concepts, and your contributions are overstated compared with the existing references**
> > >
> > > - Thanks for the comment. As explained above, our FCU and SCU are not straightforward concepts, but carefully designed based on learnable soft masks which directly encode the missing information across features and samples. They are effective in improving imputation accuracy, as demonstrated by the results in Table 2 for the ablation study in our submitted manuscript. If there are still concerns, could you kindly provide extra references that have similar designs other than the ones that we have already differentiated in the submitted manuscript and the rebuttal? We can differentiate further. Thanks again.

---

> > > > ### Comment · Reviewer_pTfx · 2024-11-24
> > > > **Both FCU and SCU are straightforward concepts,**
> > > >
> > > > As demonstrated in all the papers I cited from the literature, they essentially employed similar ideas to leverage comparable information. Therefore, I cannot be positive on your contributions.

---

> > > > > ### Author Response · Authors · 2024-11-25
> > > > >
> > > > > **Q(2) As demonstrated in all the papers I cited from the literature, they essentially employed similar ideas to leverage comparable information. Therefore, I cannot be positive on your contributions.**
> > > > >
> > > > > Thanks for the comment.
> > > > > 1. We would like to point out that in our previous response to your comment: "Third, some concepts similar to the feature correlation unit (FCU) and sample correlation unit (SCU) have been previously explored in the following references [3-7]",  we have already differentiated the FCU and SCU in M$^3$-Impute from the listed references. Specifically, none of the existing graph-based imputation methods or the cited studies have a similar design to our FCU and SCU. The FCU dynamically adjusts the feature correlations by considering both the imputation targets and the missingness patterns, and the SCU introduces an improved measure of sample correlations which results in enhanced imputation performance. To the best of our knowledge, this direct and effective encoding of missingness information introduced by the FCU and SCU has not been proposed in the literature.
> > > > >
> > > > > 2. In addition, we incorporate an extra section (Section B) in the appendix of our revised manuscript to demonstrate the capabilities of FCU and SCU. In the newly added Section B.1, we show that FCU utilizes the sample missingness information and dynamically adjusts the correlations between the target missing feature and observed features within each sample. In Section B.2, we demonstrate how SCU effectively measures sample correlations, resulting in improved imputation performance. To this end, we first show the limitations of using the popular cosine similarity measure to compute sample correlations, where it fails to consider the missingness pattern in a sample and the importance of different observed features to the target feature to impute. In contrast, SCU employs a novel mutual sample masking mechanism, integrating it with the FCU to jointly account for commonly observed features across samples and their importance for different imputation targets. This approach results in more robust modeling of sample correlations and significantly enhances imputation performance. Please kindly refer to Section B of the appendix (Pages 26–28) in the revised manuscript.

---

> ### Author Response · Authors · 2024-11-21
>
> **Q(1) You need to clearly describe how the MCAR, MAR, and MNAR settings are generated in the paper. This is currently missing. As a result, your claim regarding MNAR is inaccurate.**
>
> Thanks for the comment. We would like to point out that the steps for generating missingness under MCAR, MAR, and MNAR settings are detailed in Section A.2 of the manuscript, and this process is the same as the one in [1, 2].
>
> 1. For the MCAR setting, given a dataset containing $n$ samples and $m$ features, we generate an $n \times m$ matrix as a mask. Each entry value of the matrix is sampled from a uniform distribution. Entries with values no greater than the missingness ratio are treated as missing, i.e., masked out, and the remaining entries remain observable.
> 2. For the MAR setting, a subset of features is randomly selected to be fully observed. Thus, only the remaining features can have missing values. The values for the remaining features are removed according to a logistic model with random weights, using the fully observed feature values as input. The desired missingness ratio is achieved by adjusting the bias term.
> 3. For the MNAR setting, we first apply the MAR mechanism. Once its output is generated, we also remove additional feature values according to the MCAR mechanism.
>
>
> [1] You, Jiaxuan, et al. "Handling missing data with graph representation learning." Advances in Neural Information Processing Systems (2020).
>
> [2] Jarrett, Daniel, et al. "Hyperimpute: Generalized iterative imputation with automatic model selection." International Conference on Machine Learning. PMLR, 2022.

---

> ### Author Response · Authors · 2024-11-29
>
> Dear reviewer pTfx,
>
> &nbsp;
>
> We hope our responses have addressed your concerns. If there are any remaining questions, we would be happy to provide further clarification. May we kindly ask if our responses have resolved your concerns?
>
> &nbsp;
>
> Sincerely,
>
> Paper 5469 authors.

---

### Official Review · Reviewer_ykDf · 2024-10-31

**Soundness:** 3
**Presentation:** 3
**Contribution:** 2
**Rating:** 5
**Confidence:** 3

**Summary:**

The paper introduces $M^3$-Impute, a novel method for missing value imputation that leverages mask-guided representation learning to address the challenge of accurately imputing missing data. M3-Impute incorporates missingness information and captures feature and sample correlations through innovative masking schemes, offering a significant advancement in the field. The method's key contributions include a refined embedding initialization process that integrates missingness information, the introduction of feature correlation unit (FCU) and sample correlation unit (SCU) to effectively model feature-wise and sample-wise correlations, and extensive experiments on 25 datasets demonstrating M3-Impute's superior performance over existing methods under various missing data patterns. By jointly modeling feature and sample correlations with missingness information, M3-Impute leads to improved accuracy in imputing missing values.

**Strengths:**

1.$M^3$-Impute demonstrates a high level of originality, with significant contributions including an innovative embedding initialization process that incorporates missingness information, which is uncommon in previous studies and offers a new perspective for the field of missing data imputation.

2.The introduction of feature and sample correlation unit (FCU and SCU) is innovative as they explicitly model feature and sample correlations through a soft masking mechanism, marking a novel approach in existing missing data imputation methods.

3.Extensive experiments on 25 datasets demonstrating $M^3$-Impute's superior performance over existing methods under different missing data patterns.

**Weaknesses:**

1.To strengthen the paper, it would benefit from comparing $M^3$-Impute against the most recent state-of-the-art methods in missing value imputation, highlighting its unique advantages over the latest techniques.

2.The paper should include a detailed examination of $M^3$-Impute's performance with large-scale datasets to assess its scalability and efficiency as data volumes expand.

3.Adding a section on the model's interpretability would enhance the paper, as it is vital to understand the model's predictions for trust and application in critical fields like healthcare and finance.

4.The paper should delve into $M^3$-Impute's computational efficiency, including memory usage and optimization, to address resource limitations in practical applications.

**Questions:**

1.The experiments, although extensive, are primarily conducted on a set of datasets that may not fully represent the diversity of real-world data. It would be beneficial to include more datasets with varying characteristics, such as different domains and levels of missingness (eg: a certain type of severe deficiency), to more comprehensively evaluate the robustness and generalizability of $M^3$-Impute.

2.The paper notes that $M^3$-Impute struggles with datasets where data points have low dependencies, as FCU and SCU rely on data similarity. This can lead to sparse or inadequate graph formation and poor model performance. Can we employ strategies such as enhancing data representation and refining graph construction techniques to tackle these challenges?

3.Reference [1] also engages in graph construction and employs graph representation learning for missing value imputation. However, the approach to graph construction varies between the two methods. Has the author conducted a comparative analysis? Is there a correlation between the efficiency of missing value imputation and the method of graph construction? Which approach proves to be more effective?

[1] Jiaxuan You, Xiaobai Ma, Daisy Yi Ding, Mykel J. Kochenderfer, and Jure Leskovec. Handling missing data with graph representation learning. Advances in Neural Information Processing Systems 33: Annual Conference on Neural Information Processing Systems 2020, NeurIPS 2020.

---

> ### Author Response · Authors · 2024-11-25
>
> **Q(1). To strengthen the paper, it would benefit from comparing M$^3$-Impute against the most recent state-of-the-art methods in missing value imputation, highlighting its unique advantages over the latest techniques.**
> - Thanks for the comment. In our manuscript, we included two highly competitive imputation methods. One is Hyperimpute, which is an ensemble method that demonstrates highly competitive imputation accuracy and even outperforms the state-of-the-art imputation methods [1, 2] in 2024. The other is Grape, a slightly older method that still delivers impressive results and occasionally outperforms HyperImpute. We believe the current baseline selection is fair and comprehensive, with a wide range of experiments with 10 baseline methods on 25 open datasets. We would also like to point out that M$^3$-Impute consistently outperforms both HyperImpute and Grape for most of the 25 datasets under all three missingness patterns. Please kindly refer to Table 1, Table 7, and Table 8 (as well as other tables and figures) of the submitted manuscript for more details.
>
>
> **Q(2). The paper should include a detailed examination of M$^3$-Impute's performance with large-scale datasets to assess its scalability and efficiency as data volumes expand.**
> - Thanks for the comment. We would like to point out that we indeed included some large-scale datasets, such as Protein, Spam, Letter, and California-Housing. M$^3$-Impute runs efficiently for inference on these large datasets and consistently achieves better imputation accuracy compared to baseline methods. The inference time on various datasets is presented in Table 10 of the submitted manuscript. Results show that M$^3$-impute takes less than one second to impute all the missing values in each dataset with 30\% missingness when using a GPU. Please also kindly refer to Table 9 of the submitted manuscript for statistics of the large datasets and Table 19, Table 20, and Table 21 for imputation accuracy under various missingness conditions.
>
>
> **Q(3). Adding a section on the model's interpretability would enhance the paper, as it is vital to understand the model's predictions for trust and application in critical fields like healthcare and finance.**
> - Thanks for the comment. We appreciate your suggestion to add a section on model interpretability and incorporate an extra section (Section B) in the appendix of our revised manuscript. In the newly added Section B.1, we show that FCU utilizes the sample missingness information and dynamically adjusts the correlations between the target missing feature and observed features within each sample. In Section B.2, we demonstrate how SCU effectively measures sample correlations, resulting in improved imputation performance. To this end, we first show the limitations of using the popular cosine similarity measure to compute sample correlations, where it fails to consider the missingness pattern in a sample and the importance of different observed features to the target feature to impute. In contrast, SCU employs a novel mutual sample masking mechanism, integrating it with the FCU to jointly account for commonly observed features across samples and their importance for different imputation targets. This approach results in more robust modeling of sample correlations and significantly enhances imputation performance. Please kindly refer to Section B of the appendix (Pages 26–28) in the revised manuscript. Thanks again.
>
> [1] Du, T, et al. "ReMasker: Imputing Tabular Data with Masked Autoencoding" ICLR 2024
>
> [2] Zhang, H, et al. "Unleashing the Potential of Diffusion Models for Incomplete Data Imputation" arXiv preprint (2024).

---

> ### Author Response · Authors · 2024-11-25
>
> **Q(4). The paper should delve into M$^3$-Impute's computational efficiency, including memory usage and optimization, to address resource limitations in practical applications.**
> - Thanks for the comment. M$^3$-Impute is efficient, with reasonable running time and memory usage. In Tables 4 and 10 of the submitted manuscript, we present a comparison of inference times between M$^3$-Impute and baseline methods. The results highlight that M$^3$-Impute is computationally efficient during inference, requiring less than one second to impute all missing values in each dataset with 30\% missingness when using a GPU.
>
> - In addition, the tables below provide a detailed analysis of M$^3$-Impute's computational complexity during the training phase. Table 1 reports the GPU runtime of M$^3$-Impute during training. As shown, M$^3$-Impute demonstrates high efficiency, requiring less than one second to complete the forward and backward passes for a single epoch across all examined datasets. Furthermore, Table 2 summarizes the GPU memory usage of M$^3$-Impute when employing a mini-batch training mechanism with a batch size of 256. The results highlight that M$^3$-Impute maintains a moderate memory footprint while demonstrating excellent scalability, even when applied to large-scale datasets such as Naval, Power, and Kin8nm.
>
>
>    ### Table 1: GPU running time per epoch (in seconds) at the training stage.
>    | Method         | Yacht | Wine | Concrete | Housing | Energy | Naval | Kin8nm | Power |
>    |-----------------------|-------|------|----------|---------|--------|-------|--------|-------|
>    | Init                 | 0.02  | 0.02 | 0.02     | 0.02    | 0.02   | 0.08  | 0.04   | 0.03  |
>    | Init + FCU           | 0.02  | 0.02 | 0.02     | 0.02    | 0.02   | 0.10  | 0.05   | 0.04  |
>    | Init + SCU (Peer=5)  | 0.03  | 0.06 | 0.03     | 0.03    | 0.03   | 0.54  | 0.15   | 0.12  |
>    | M3-Impute (Peer=5)   | 0.03  | 0.06 | 0.04     | 0.04    | 0.04   | 0.54  | 0.16   | 0.14  |
>
>
>    ### Table 2: Memory Usage at the training stage (batch_size=256).
>    |                Method             | Yacht  | Wine   | Concrete | Housing | Energy | Naval  | Kin8nm | Power  |
>    |-----------------------------------|--------|--------|----------|---------|--------|--------|--------|--------|
>    | Init                              | 1279MiB| 1425MiB| 1332MiB  | 1310MiB | 1315MiB| 3007MiB| 1943MiB| 1698MiB|
>    | Init + FCU                        | 1281MiB| 1433MiB| 1341MiB  | 1317MiB | 1321MiB| 3017MiB| 1953MiB| 1703MiB|
>    | Init + SCU (peer=5)               | 1313MiB| 1453MiB| 1373MiB  | 1351MiB | 1353MiB| 3053MiB| 1987MiB| 1737MiB|
>    | M3-Impute (Peer=5)                | 1317MiB| 1455MiB| 1379MiB  | 1359MiB | 1353MiB| 3063MiB| 1992MiB| 1743MiB|
>
>
> **Q(5). The experiments, although extensive, are primarily conducted on a set of datasets that may not fully represent the diversity of real-world data. It would be beneficial to include more datasets with varying characteristics, such as different domains and levels of missingness (eg: a certain type of severe deficiency), to more comprehensively evaluate the robustness and generalizability of M$^3$-Impute.**
>
> - Thanks for the comment. We would like to point out that the 25 datasets used in the paper cover a wide range of real-world scenarios, including civil engineering (CONCRETE, ENERGY), physics and chemistry (YACHT), thermal dynamics (NAVAL), healthcare (BREAST cancer, DIABETES), and finance (California-Housing). These datasets also possess mixed data types with both continuous and discrete values. In addition, we would like to clarify that we indeed evaluated M$^3$-Impute against baseline methods across 4 different levels of missingness, ranging from missing 10\% to missing 70\%, and under three distinct missingness patterns, i.e, MCAR, MAR, and MNAR. Detailed results can be found in the submitted manuscript, where Table 16 and Table 19 present the MCAR results, Table 17 and Table 20 present the MAR results, and Table 18 and Table 21 present the MNAR results.

---

> ### Author Response · Authors · 2024-11-25
>
> **Q(6). The paper notes that  M$^3$-Impute struggles with datasets where data points have low dependencies, as FCU and SCU rely on data similarity. This can lead to sparse or inadequate graph formation and poor model performance. Can we employ strategies such as enhancing data representation and refining graph construction techniques to tackle these challenges?**
>
> - Thanks for the comment. During graph formation, low data-point dependencies *do not* make the graph sparse and inadequate. Instead, the connectivity of the constructed bipartite graph is determined by the level of missingness in the tabular data. When missingness level is high, the bipartite graph exhibits low connectivity, as missing data is modeled as non-existent edges. We evaluated the robustness of M$^3$-Impute across different levels of missingness (up to 70\%). As presented in Table 16 to Table 21. The results show that M$^3$-Impute is robust across different levels of missingness and consistently outperforms other methods. We appreciate the suggested strategies and will consider them as a future work. Thanks.
>
> **Q(7). Grape also engages in graph construction and employs graph representation learning for missing value imputation. However, the approach to graph construction varies between the two methods. Has the author conducted a comparative analysis? Is there a correlation between the efficiency of missing value imputation and the method of graph construction? Which approach proves to be more effective?**
>
> - Thanks for the comment. We would like to clarify that the graph construction in M$^3$-Impute is the same as the one in Grape. However, M$^3$-Impute improves the initialization of sample-node and feature-node embeddings in Grape. More specifically, Grape initializes all sample node embeddings as all-one vectors and feature node embeddings as one-hot vectors, which have a value 1 in the positions representing their respective features and 0's elsewhere. We observe, however, that such an initialization does not effectively utilize the information from the masked data matrix, which leads to inferior imputation accuracy. Thus, in M$^3$-Impute, we propose to initialize each sample node embedding based on its associated (initial) feature node embeddings instead of initializing them separately. While the feature node embeddings are randomly initialized as a learnable vector, the sample node embeddings are initialized in a way that reflects the embeddings of the features whose values are available in their corresponding samples, detailed in Section 3.2 of the submitted manuscript.
>
> - Finally, we would like to point out that all the novel units proposed in M3-impute (including the initialization of node embeddings, the FCU, and the SCU) are *independent* of the bipartite graph modeling. Thus, all the novel units can be easily adapted to any graph that would be constructed with an improved modeling technique and can benefit from the improved graph model.

---

> ### Author Response · Authors · 2024-11-29
>
> Dear reviewer ykDf,
>
> &nbsp;
>
> We hope our responses have addressed your concerns. If there are any remaining questions, we would be happy to provide further clarification. May we kindly ask if our responses have resolved your concerns?
>
> &nbsp;
>
> Sincerely,
>
> Paper 5469 authors.

---

### Official Review · Reviewer_7xNf · 2024-11-04

**Soundness:** 2
**Presentation:** 2
**Contribution:** 3
**Rating:** 6
**Confidence:** 4

**Summary:**

The paper proposes a novel grpah neural network-based approach to address the problem of missing value imputation. The proposed method explicitly uses the missingness information with correlations between features and samples. With 25 benchmark datasets, authors demonstrated its superiority compared to recent missing value imputation methods.

**Strengths:**

- Clear statement and approach: This paper clearly states the limitations of existing methods and thereby why the proposed method is necessary.

- Novel approach: Combination of feature correlation, sample correlation, and missingness information is fair separately and concurrently.

- Comprehensive experiments: This paper conducted comprehensive experiments with 25 benchmark datasets and 3 missingness patterns. This is the most broad experimental result among studies on missing value imputation.

- Detailed description: Authors provided detailed explanation for each module in the proposed method.

**Weaknesses:**

Complexity issue: The proposed method is much more complex than other methods, including its predecessor GRAPE. Although the authors already provided the computing time at test phase, but I am curious about how long the training phase takes.

Limitations in comparative analysis: While the authors conducted comprehensive experiments, but the improvements are mostly marginal, making it difficult to assess whether the improvements are significant or not. The authors should consider ways to present in a more compelling manner to highlight their significance.

**Questions:**

1. Is there any additional analysis regarding the computational complexity during the training phase?

2. Is there any analysis when the proposed method significantly outperforms and when does not. In practicality, it is helpful to know when the proposed method is powerful since it entails more complexity. If the improvement is marginal, GRAPE can be a good alternative.

3. How did you tune the proposed method? Information about baselines are attached in Appendix but I cannot see about the proposed method. Also, some ablation study only presents marginal changes, so it is difficult to get insights from them. Is there any different results about them?

---

> ### Author Response · Authors · 2024-11-21
>
> **Q(1) Complexity issue: The proposed method is much more complex than other methods, including its predecessor GRAPE. Although the authors already provided the computing time at test phase, but I am curious about how long the training phase takes.**
>
> - Thanks for the comment. M$^3$-Impute is also efficient during training. Table 1 presents the GPU running time of M$^3$-Impute components during training. The time taken to impute all the missing values in a training epoch for any dataset we tested is less than one second under the setting of MCAR with 30\% missingness.
>
>    &nbsp;
>    ### Table 1: GPU running time per epoch (in seconds) at the training stage.
>    | Method         | Yacht | Wine | Concrete | Housing | Energy | Naval | Kin8nm | Power |
>    |-----------------------|-------|------|----------|---------|--------|-------|--------|-------|
>    | Init                 | 0.02  | 0.02 | 0.02     | 0.02    | 0.02   | 0.08  | 0.04   | 0.03  |
>    | Init + FCU           | 0.02  | 0.02 | 0.02     | 0.02    | 0.02   | 0.10  | 0.05   | 0.04  |
>    | Init + SCU (Peer=5)  | 0.03  | 0.06 | 0.03     | 0.03    | 0.03   | 0.54  | 0.15   | 0.12  |
>    | M3-Impute (Peer=5)   | 0.03  | 0.06 | 0.04     | 0.04    | 0.04   | 0.54  | 0.16   | 0.14  |

---

> ### Author Response · Authors · 2024-11-21
>
> **Q(2) Limitations in comparative analysis: While the authors conducted comprehensive experiments, but the improvements are mostly marginal, making it difficult to assess whether the improvements are significant or not. The authors should consider ways to present in a more compelling manner to highlight their significance.**
>
> - Thanks for the comment. To provide a clearer understanding of the conditions under which M$^3$-Impute significantly outperforms, we categorize real-world datasets based on the level of feature correlations (Pearson correlations) in each dataset.
>
>
> 1. **Low feature correlations**: This refers to the datasets where each feature is *weakly correlated or even independent* of most or all the other features. Imputation in such datasets is particularly challenging, since knowing any of the non-missing features offers little help in imputing the missing ones due to their independence. An example dataset in our submitted manuscript is "Kin8nm", where most features are independent of each other. Nonetheless, as shown in Table 1, M$^3$-Impute achieves the second-best result and does remain competitive, with a difference of less than 0.02 in MAE compared to the top performer.
>
> 2. **High feature correlations:** This refers to the datasets where each feature is *highly correlated* with most or even all the rest features. For such datasets, imputation is relatively easy, since knowing any of the non-missing features makes the inference of missing values easier. Such datasets in our manuscript are "Naval", "Power", "Breast", "Diabetes", and "Protein". As shown in the Table 2 below, the improvements brought by M$^3$-Impute over SOTA method Grape are marginal (for the comprehensive results, please refer to Table 1, 16-21 in the submitted manuscript). Nonetheless, M$^3$-Impute consistently outperforms GRAPE.
>
>     &nbsp;
>     ### Table 2: Results for datasets with high feature correlations. MAE scores under MCAR setting with 30\% missingness are reported.
>     | Model          | Naval  | Power | Breast | Diabetes | Protein |
>     |-----------------|--------|-------|--------|----------|---------|
>     | Grape           | 0.07   | 1     | 0.37   | 1.31     | 0.25    |
>     | M$^3$-Impute    | 0.06   | 0.99  | 0.36   | 1.29     | 0.24    |
>     | Improv. Ratio   | 14.29% | 1.00% | 2.70%  | 1.53%    | 4.00%   |
>
> 3. **Medium level of feature correlations:** This refers to the rest of the datasets (other than the ones exhibiting low and high feature correlations), which is the most common case in real-world applications. Specifically, a dataset has a medium level of feature correlations when each feature is highly correlated with a small subset of features and weakly correlated or nearly independent of the others. For such datasets, imputation is also challenging because we need to consider feature-feature, sample-sample, and sample-feature correlations. Given a subset of observable features and a target feature value to impute in a sample, an ideal model should identify which features are highly or weakly correlated with the target, i.e., sample-feature and feature-feature correlations. In addition, the model should also consider which similar samples to use in imputation, i.e., sample-sample correlations altogether to achieve good imputation results. Our proposed feature correlation unit (FCU) and sample correlation unit (SCU) are designed to capture these three types of correlations. In our submitted manuscript, datasets having a medium level of feature correlations are: "Yacht", "Concrete", "Energy", "Ionosphere", "Abalone", "German", and "Steel". As shown in the Table 3 below, M$^3$-Impute outperforms GRAPE substantially.
>
>     &nbsp;
>     ### Table 3: Results for datasets with medium level of feature correlations. MAE scores under MCAR setting with 30\% missingness are reported.
>     | Model           | Yacht  | Concrete | Energy | Ionosphere | Abalone | German | Steel |
>     |-----------------|--------|----------|--------|------------|---------|--------|-------|
>     | Grape           | 1.46   | 0.75     | 1.36   | 1.08       | 2.01    | 2.02   | 0.45  |
>     | M$^3$-Impute    | 1.33   | 0.71     | 1.31   | 1.01       | 1.84    | 1.87   | 0.39  |
>     | Improv. Ratio   | 8.90%  | 5.33%    | 3.68%  | 6.48%      | 8.46%   | 7.43%  | 13.33% |

---

> ### Author Response · Authors · 2024-11-21
>
> **Q(3) Is there any additional analysis regarding the computational complexity during the training phase?**
> - Thanks for the comment. Please kindly refer to the answer for Q(1).
>
>
> **Q(4) Is there any analysis when the proposed method significantly outperforms and when does not. In practicality, it is helpful to know when the proposed method is powerful since it entails more complexity. If the improvement is marginal, GRAPE can be a good alternative.**
> - Thanks for the comment. Please kindly refer to the answer for Q(2).
>
> **Q(5) How did you tune the proposed method? Information about baselines are attached in Appendix but I cannot see about the proposed method. Also, some ablation study only presents marginal changes, so it is difficult to get insights from them. Is there any different results about them?**
>
> - Thanks for the comment. (1) The settings of M$^3$-Impute can be found in Section 4.2 of the submitted manuscript. We also included the results of hyperparameter analysis in M$^3$-Impute in Table 12, Table 13, and Table 14 of the submitted manuscript. (2) We include additional ablation results on datasets with medium level of feature correlations in Table 4 below. The results show that M$^3$-Impute again outperforms GRAPE substantially. More importantly, all the three novel units in M$^3$-Impute contribute to the improvement.
>
>    &nbsp;
>    ### Table 4: Additional ablation studies. MAE scores under MCAR setting with 30% missingness are reported.
>    | Model          | Ionosphere | Abalone | German | Steel |
>    |-----------------|------------|---------|--------|-------|
>    | Grape           | 1.08 $\pm$ .01       | 2.01  $\pm$ .02  | 2.02 $\pm$ .01  | 0.45 $\pm$ .00 |
>    | M$^3$-Impute    | 1.01 $\pm$ .01       | 1.84  $\pm$ .02  | 1.87 $\pm$ .02  | 0.39 $\pm$ .00 |
>    | Improv. Ratio   | 6.48%      | 8.46%   | 6.97%  | 13.33% |
>    | Architecture    | | | | |
>    | Init Only       | 1.08 $\pm$ .01       | 2.00  $\pm$ .02  | 2.00 $\pm$ .01  | 0.45 $\pm$ .00 |
>    | Init + FCU      | 1.01 $\pm$ .03       | 1.89  $\pm$ .02  | 1.87 $\pm$ .02  | 0.42 $\pm$ .00 |
>    | Init + SCU      | 1.02 $\pm$ .00       | 1.84  $\pm$ .02  | 1.86 $\pm$ .03  | 0.42 $\pm$ .01 |

---

> ### Author Response · Authors · 2024-11-29
>
> Dear reviewer 7xNf,
>
> &nbsp;
>
> We hope our responses have addressed your concerns. If there are any remaining questions, we would be happy to provide further clarification. May we kindly ask if our responses have resolved your concerns?
>
> &nbsp;
>
> Sincerely,
>
> Paper 5469 authors.

---

### Author Response · Authors · 2024-11-25

Dear Reviewers,

&nbsp;

**We have uploaded a revised version of our manuscript.**

In this revision, we have added a section (Pages 26 to 28 in the revised manuscript) at the end of the Appendix to provide further clarification on the working mechanism and effectiveness of the Feature Correlation Unit (FCU) and the Sample Correlation Unit (SCU) in M$^3$-Impute.

1. **In Section B.1**, we show that the FCU dynamically adjusts the feature correlations by considering both the imputation targets and the missingness patterns.

2. **In Section B.2**, we highlight that the SCU introduces an improved measure of sample correlations which results in enhanced imputation performance compared to the one with the standard cosine similarity measure.

&nbsp;

Thank you very much.

&nbsp;

Best Regards,\
Paper 5469 authors.

---

### Meta-Review · Area_Chair_UoEw · 2024-12-18

**Metareview:**

The current paper proposed a new graph neural network for missing value imputation, with novel modules explicitly considering missing patterns. All reviewers acknowledged that the empirical study is extensive and the paper is well written. After the rebuttal, the reviewers have remaining concerns on how the method tackles different missing mechanisms, including the highly challenging MNAR, which should be analyzed in more detail, and modest claims should be made as only one specific MNAR setting is tested. The method's scalability needs more clarification on dataset with a large number features and on the computational overhead of FCU and SCU modules. The effectiveness analysis of the imputation should incorporate both feature correlations and sample correlations.

**Additional Comments On Reviewer Discussion:**

There are two borderline ratings and one strong rejection. All reviewers replied to the authors' rebuttal.

This is a highly practical paper and its scalability and effectiveness needs more testing based on the reviewer's post-rebuttal comments.

I communicated with the reviewer pTfx (who put a strong rejection) through email, who confirmed their comments are not sufficiently addressed. The authors raised some concerns on this reviewer. However the other two reviewers did not provide a strong case of acceptance.

Additionally, I looked at the code in the supplementary material, the overall quality is not satisfactory.

---

### Decision · Program_Chairs · 2025-01-22

Reject